# On the Statistical Consistency of Risk-Sensitive Bayesian Decision-Making

**Prateek Jaiswal**
Department of Statistics
Texas A&M University
College Station, TX 77843
jaiswalp@stat.tamu.edu

**Harsha Honnappa**
School of Industrial Engineering
Purdue University
West Lafayette, IN, 47906
honnappa@purdue.edu

**Vinayak A. Rao**
Department of Statistics
Purdue University
West Lafayette, IN, 47907
varao@purdue.edu

## Abstract

We study data-driven decision-making problems in the Bayesian framework, where the expectation in the Bayes risk is replaced by a risk-sensitive entropic risk measure with respect to the posterior distribution. We focus on problems where calculating the posterior distribution is intractable, a typical situation in modern applications with large datasets and complex data generating models. We leverage a dual representation of the entropic risk measure to introduce a novel risk-sensitive variational Bayesian (RSVB) framework for jointly computing a risk-sensitive posterior approximation and the corresponding decision rule. Our general framework includes *loss-calibrated* VB [16] as a special case. We also study the impact of these computational approximations on the predictive performance of the inferred decision rules. We compute the convergence rates of the RSVB approximate posterior and the corresponding optimal value. We illustrate our theoretical findings in parametric and nonparametric settings with the help of three examples.

## 1 Introduction

This paper focuses on *risk-sensitive* Bayesian decision-making, considering objective functions of the form

$$\min_{a \in \mathcal{A}} \varrho^{\gamma}_{\Pi_n}(R(a, \theta)) := \gamma^{-1} \log \mathbb{E}_{\Pi_n}[\exp(\gamma R(a, \theta))]. \tag{SO}$$

Here $\mathcal{A}$ is the decision/action space, $\theta$ is a random model parameter lying in an arbitrary measurable space $(\Theta, \mathcal{T})$, and $R(a, \theta) : \mathcal{A} \times \Theta \mapsto \mathbb{R}$ is a problem-specific model risk function. For any $B \subseteq \Theta$, $\Pi_n(B) := \Pi(B|\tilde{X}_n)$ is the Bayesian posterior distribution over the parameters given observations $\tilde{X}_n$ from the true model $P^n_{\theta_0}(\equiv P^n_0)$ with parameter $\theta_0 \in \Theta$. The scalar $\gamma \in \mathbb{R}$ is user-specified and characterizes the sensitivity of the decision-maker (DM) to the distribution $\Pi_n$. Recall that the posterior distribution is obtained by updating a prior probability distribution $\Pi(\cdot)$, capturing subjective beliefs of the decision maker over $\Theta$, according to the Bayes rule

$$d\Pi(\theta|\tilde{X}_n) \propto d\Pi(\theta)dP^n_\theta(\tilde{X}_n), \tag{1}$$

where $dP^n_\theta(\tilde{X}_n)$ is the likelihood of observing $\tilde{X}_n$. We denote the corresponding densities (if they exist) for the model, prior, and posterior as $p^n_\theta(\cdot)$, $\pi(\cdot)$, and $\pi(\cdot|\tilde{X}_n)$ respectively.

The functional $\varrho^{\gamma}$ in (SO) is also known as the *entropic risk measure*, and models a range of risk-averse or risk-seeking behaviors in a succinct manner through the parameter $\gamma$. Consider only strictly positive $\gamma$, and observe that $\lim_{\gamma \downarrow 0} \frac{1}{\gamma} \log \mathbb{E}_{\Pi_n}[\exp(\gamma R(a, \theta))] = \mathbb{E}_{\Pi_n}(R(a, \theta))$; that is, there is no sensitivity to potential risks due to large tail effects and the decision-maker is risk neutral.

37th Conference on Neural Information Processing Systems (NeurIPS 2023).

On the other hand, $\lim_{\gamma \to +\infty} \varrho_{\Pi_n}^{\gamma}(R(a,\theta)) = \operatorname{ess\,sup}_{\Pi_n}(R(a,\theta))$, the essential supremum of the model risk $R(a,\theta)$. In other words, a decision maker is completely risk averse and anticipates the worst possible realization ($\Pi_n$-almost surely). Observe that (SO) strictly generalizes the standard Bayesian decision-theoretic formulation of a decision-making problem, where the goal is to solve $\min_{a \in \mathcal{A}} \mathbb{E}_{\Pi_n}[R(a,\theta)]$.

The risk-sensitive formulation (SO) is very general and can be used to model a wide variety of decision-making problems in machine learning [16, 15], operations research/management science [24, 7, 20], simulation optimization [6, 32], and finance [17, 1, 4]. However, solving (SO) to compute an optimal decision over $\mathcal{A}$ is challenging. The difficulty mainly stems from the fact that, with the exception of conjugate models, the posterior distribution in (1) is an intractable quantity. Canonically, posterior intractability is addressed using either a sampling- or optimization-based approach. Sampling-based approaches, such as Markov chain Monte Carlo (MCMC), offer a tractable way to compute the integrals and theoretical guarantees of exact inference in the large computational budget limit.

Optimization-based methods such as variational Bayes (VB) or variational inference (VI) have emerged as a popular alternative [3]. The VB approximation of the true posterior is a tractable distribution chosen from a 'simpler' family of distributions known as a variational family by minimizing the discrepancy between the true posterior and members of that family. Kullback-Liebler (KL) divergence is the most often used measure of the approximation discrepancy, although other divergences (such as the $\alpha$-Rényi divergence [19, 28, 14]) have been used. The minimizing member, termed the VB approximate posterior, can be used as a proxy for the true posterior. Empirical studies have shown that VB methods are computationally faster and far more scalable to higher-dimensional problems and large datasets. Theoretical guarantees, such as large sample statistical inference, have been a topic of recent interest in the theoretical statistics community, with asymptotic properties such as convergence rate and asymptotic normality of the VB approximate posterior recently established in [34, 21] and [31, 14] respectively.

Our ultimate goal is not to merely approximate the posterior distribution but also to make decisions when that posterior is intractable. A *näive* approach would be to plug in the VB approximation in place of the true posterior in (SO) and compute the optimal decision. However, it has been noted in [16] that such a naive loss-unaware approach can be suboptimal. In particular, [16] demonstrated, through an example, that a naive posterior approximation only captures the most dominant mode of the true posterior, which may not be relevant from a decision-making perspective. Consequently, they proposed a loss-calibrated variational Bayesian (LCVB) algorithm for solving Bayesian decision-making problems where the underlying risk function is discrete. [15] extended their approach to continuous risk functions. Despite these algorithmic advances in developing decision-centric VB methods, their statistical properties, such as asymptotic consistency and convergence rates of the loss-aware posterior approximation and the associated decision rule are not well understood. With an aim to address these gaps, we summarize our contribution in this paper below:

1. In Section 2, we introduce a minimax optimization framework titled 'risk-sensitive variational Bayes' (RSVB), extracted from the dual representation of (SO) using the so-called Donsker-Varadhan variational free-energy principle [9]. The decision-maker computes a risk-sensitive approximation to the true posterior (termed as RSVB posterior) and the decision rule simultaneously by solving a minimax optimization problem. Moreover, we recover the LCVB approach [16] as a special case of RSVB (with $\gamma = 1$).

2. In Section 3, we identify verifiable regularity conditions on the prior, likelihood model and the risk function under which the RSVB posterior enjoys the same rate of convergence as the true posterior to a Dirac delta distribution concentrated at the true model parameter $\theta_0$, as the sample size increases. Using this result, we also prove the rate of convergence of the RSVB decision rule when the decision space $\mathcal{A}$ is compact. Our theoretical results also imply the asymptotic properties of the LCVB posterior and the associated decision rule.

3. In Section 4, we demonstrate our theoretical results with the help of three widely studied decision-making problems, including Gaussian process classification and a newsvendor problem. For each example, we show that the rate of convergence of the respective RSVB approximate posterior matches that of the corresponding true posterior.

4. In Section 4, we also present some simulation results with the single product (1-d) newsvendor problem, which is summarized here in Figure 1. The figures demonstrate the effect of changing $\gamma$ on the optimality gap in values (see Definition 2.1) and the variance of the

RSVB posterior for a given $n$. In particular, we observe that for smaller $n$, increasing $\gamma$ (after a certain value) results in a significantly more risk-averse decision; however, the effect of increasing $\gamma$ on risk-averse decision-making reduces as $n$ increases. This observation demonstrates the fact that when there is enough certainty in the parameter estimation, the need to be risk-sensitive to parametric uncertainty diminishes. The Figure 1 also compares the optimality gap in values evaluated at RSVB decision rules and variance of the RSVB posterior for various values of $\gamma$ against the näive VB approach and against the true posterior when the conjugate prior is used.

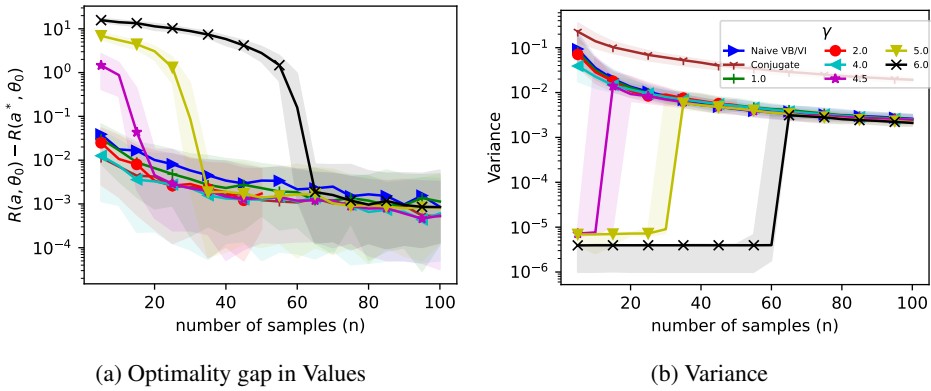

(a) Optimality gap in Values                    (b) Variance

Figure 1: (a) Optimality gap in values and (b) variance of the RSVB posterior (mean over 100 sample paths) against the number of samples ($n$) for various values of $\gamma$.

## 2  Risk-Sensitive Variational Bayes

Our approach exploits the dual representation of the log-exponential risk measure in (SO), which is convex (or extended coherent) risk measure [23, 10]. From the Donsker-Varadhan variational free energy principle [9] observe that,

$$\varrho_{\Pi_n}^{\gamma}(a) = \max_{Q \in \mathcal{M}} \left\{ \mathbb{E}_Q[R(a,\theta)] - \gamma^{-1}\mathrm{KL}(Q\|\Pi_n) \right\} \qquad \gamma > 0, \qquad \text{(DV)}$$

where $\mathcal{M}$ is the set of all distribution functions that are absolutely continuous with respect to the posterior distribution $\Pi_n$ and 'KL' represents the Kullback-Leibler divergence. Formally, for any two distributions $P$ and $Q$ defined on measurable space $(\Theta, \mathcal{T})$, the KL divergence is defined as $\mathrm{KL}(Q\|P) = \int_\Theta dQ(\theta)\log\frac{dQ(\theta)}{dP(\theta)}$, when measure $Q$ is absolutely continuous with respect to $P$ and $\infty$ otherwise. Notice that this dual formulation exposes the reason we choose to use the log-exponential risk – the 'free-energy' objective on the RHS provides a combined assessment of the risk associated with model estimation (the KL divergence $\mathrm{KL}(Q\|\Pi_n)$) and the decision risk under the estimated posterior $Q$ ($\mathbb{E}_Q[R(a,\theta)]$).

As stated above, the reformulation presented in (DV) offers no computational gains. However, restricting ourselves to an appropriately chosen subset $\mathcal{Q} \subset \mathcal{M}$, that consists of distributions where the integral $\mathbb{E}_q[R(a,\theta)]$ and $\mathrm{KL}(Q\|\Pi_n)$ can be tractably computed and optimized, we immediately obtain a *risk-sensitive variational Bayesian* (RSVB) formulation from (DV):

$$\gamma^{-1}\log\mathbb{E}_{\Pi_n}\left[e^{\gamma R(a,\theta)}\right] \geq \max_{Q \in \mathcal{Q}} \left\{ \mathbb{E}_Q[R(a,\theta)] - \gamma^{-1}\mathrm{KL}(Q\|\Pi_n) \right\} =: \mathcal{F}(a; Q(\cdot), \tilde{X}_n, \gamma). \quad \text{(RSVB)}$$

RSVB is our framework for data-driven risk-sensitive decision-making. The family of distributions $\mathcal{Q}$ is popularly known as the *variational family*. Our analysis in Section 3.1 reveals general guidelines on how to choose $\mathcal{Q}$ that ensures a small optimality gap (defined below) with high probability.

With an appropriate choice of $\mathcal{Q}$, the optimization on the RHS can yield a good approximation to the log-exponential risk measurement on the left hand side (LHS). For brevity, for a given $a \in \mathcal{A}$ we define the RSVB approximation to the true posterior $\Pi(\theta|\tilde{X}_n)$ as

$$Q_{a,\gamma}^*(\theta|\tilde{X}_n) := \mathrm{argmax}\{Q \in \mathcal{Q} : \mathcal{F}(a; Q(\cdot), \tilde{X}_n, \gamma)\}$$

and the RSVB optimal decision as

$$\mathtt{a}^*_{\mathtt{RS}} := \operatorname{argmin}_{a \in \mathcal{A}} \mathcal{F}(a; Q^*_{a,\gamma}(\theta|\tilde{X}_n), \tilde{X}_n, \gamma) = \operatorname{argmin}_{a \in \mathcal{A}} \max_{Q \in \mathcal{Q}} \mathcal{F}(a; Q(\cdot), \tilde{X}_n, \gamma).$$

Observe that $Q^*_{a,\gamma}(\theta|\tilde{X}_n)$ and $\mathtt{a}^*_{\mathtt{RS}}$ are random quantities, conditional on the data $\tilde{X}_n$.

Examples of $\mathcal{Q}$ include the family of Gaussian distributions, delta functions, or the family of factorized 'mean-field' distributions that discard correlations between components of $\theta$. The choice of $\mathcal{Q}$ is decisive in determining the performance of the algorithm. Part of the analysis in this paper is to articulate sufficient conditions on $\mathcal{Q}$ that ensure small optimality gap (defined below) for the optimal decision, $\mathtt{a}^*_{\mathtt{RS}}$. This establishes the statistical "goodness" of the procedure as number of samples increase. In this paper, we analyze the efficacy of the decision rules obtained using the RSVB approximation, by providing high-probability bounds on the optimality gap. We define the *optimality gap* for any $\mathtt{a} \in \mathcal{A}$ with value $V = R(\mathtt{a}, \theta_0)$ as,

**Definition 2.1** (Optimality Gap). *Let $V^*_0 := \min_{a \in \mathcal{A}} R(a, \theta_0)$ be the optimal value for the true model parameter $\theta_0$. Then, the optimality gap in the value is the difference $V - V^*_0$.*

A similar performance measure was used in [15], to measure the effectiveness of loss-calibrated VB (LCVB) approach, which can be obtained by setting $\gamma = 1$. We provide definitions of some standard terminologies that we use in the subsequent sections, such as covering number, test functions, $\Gamma$-convergence, and primal feasibility in the Appendix A for ready reference. In the following section, we lay down important assumptions used throughout the paper to establish our theoretical results.

## 2.1 Assumptions

In order to bound the optimality gap, we require some control over how quickly the posterior distribution concentrates at the true parameter $\theta_0$. Our next assumption in terms of a verifiable test condition on the model (sub-)space is one of the conditions required to quantify this rate. Let $L_n : \Theta \times \Theta \mapsto [0, \infty)$ be a distance metric on the model space.

**Assumption 2.1** (Model indentifiability). *Fix $n \geq 1$. Then, for any $\epsilon > \epsilon_n$ such that $\epsilon_n \to 0$ as $n \to \infty$ and $n\epsilon_n^2 \geq 1$, there exists a measurable sequence of test functions $\phi_{n,\epsilon} : \tilde{X}_n \mapsto [0, 1]$ and sieve set $\Theta_n(\epsilon) \subseteq \Theta$ such that (i) $\mathbb{E}_{P^n_0}[\phi_{n,\epsilon}] \leq C_0 \exp(-Cn\epsilon^2)$, and (ii) $\sup_{\{\theta \in \Theta_n(\epsilon): L_n(\theta, \theta_0) \geq C_1 n\epsilon^2\}} \mathbb{E}_{P^n_\theta}[1 - \phi_{n,\epsilon}] \leq \exp(-Cn\epsilon^2).$*

Observe that Assumption 2.1$(i)$ quantifies the rate at which a Type I error diminishes with the sample size, while the condition in Assumption 2.1$(ii)$ quantifies that of a Type II error. Notice that both of these are stated through test functions; indeed, what is required are consistent test functions. [11, Theorem 7.1] (stated in Appendix as Lemma C.7 for completeness) roughly implies that an appropriately bounded model subspace $\{P^n_\theta, \theta \in \Theta\}$ guarantees the existence of *consistent* test functions, to test the null hypothesis that the true parameter is $\theta_0$ against an alternate hypothesis – the alternate being defined using the distance function $L_n$. Subsequently, we will use a specific distance function to obtain finite sample bounds for the optimality gap in decisions and values. Note that in some cases, it is also possible to construct consistent test functions directly without recourse to Lemma C.7. We demonstrate this in Section 4.1 below. Next, we assume a condition on the prior distribution that ensures that it provides sufficient mass to the sieve set $\Theta_n(\epsilon) \subseteq \Theta$, as defined above in Assumption 2.1.

**Assumption 2.2.** *Fix $n \geq 1$. Then, for any $\epsilon > \epsilon_n$ such that $\epsilon_n \to 0$ as $n \to \infty$ and $n\epsilon_n^2 \geq 1$, the prior distribution satisfies $\Pi(\Theta^c_n(\epsilon)) \leq \exp(-Cn\epsilon^2)$.*

Notice that Assumption 2.2 is trivially satisfied if $\Theta_n(\epsilon) = \Theta$. The next assumption ensures that the prior distribution places sufficient mass around a neighborhood of the distribution $P^n_0$.

**Assumption 2.3** (Prior thickness). *Fix $n \geq 1$ and a constant $\lambda > 0$. Let $A_n := \{\theta \in \Theta : D_{1+\lambda} (P^n_0 \| P^n_\theta) \leq C_3 n\epsilon_n^2\}$, where $D_{1+\lambda} (P^n_0 \| P^n_\theta) := \frac{1}{\lambda} \log \int \left(\frac{dP^n_0}{dP^n_\theta}\right)^\lambda dP^n_0$ is the Rényi divergence between $P^n_0$ and $P^n_\theta$, assuming $P^n_0$ is absolutely continuous with respect to $P^n_\theta$. The prior distribution satisfies $\Pi(A_n) \geq \exp(-nC_2\epsilon_n^2)$.*

This assumption guarantees that the prior distribution covers the neighborhood with positive mass. The above three assumptions are adopted from [11] and has also been used in [34] to prove convergence

rates of variational posteriors. Interested readers may refer to [11] and [34] to read more about the above assumptions.

It is apparent by the first term in (RSVB) that in addition to Assumption 2.1, 2.2, and 2.3, we also require regularity conditions on the risk function $R(a, \cdot)$. The next assumption restricts the prior distribution with respect to $R(a, \theta)$.

**Assumption 2.4.** *Fix $n \geq 1$ and $\gamma > 0$. For any $\epsilon > \epsilon_n$, $a \in \mathcal{A}$, $\mathbb{E}_\Pi[\mathbb{1}_{\{\gamma R(a,\theta) > C_4(\gamma)n\epsilon^2\}} e^{\gamma R(a,\theta)}] \leq \exp(-C_5(\gamma)n\epsilon^2)$, where $C_4(\gamma)$ and $C_5(\gamma)$ are scalar positive functions of $\gamma$.*

Note that the set $\{\gamma R(a, \theta) > C_4(\gamma)n\epsilon^2\}$ represents the subset of the model space where the risk $R(a, \theta)$ (for a fixed decision $a$) is large, and the prior is assumed to place sufficiently small mass over such sets. Moreover, using the Cauchy-Schwarz inequality observe that $\mathbb{E}_\Pi[\mathbb{1}_{\{\gamma R(a,\theta) > C_4(\gamma)n\epsilon^2\}} e^{\gamma R(a,\theta)}] \leq \left(\mathbb{E}_\Pi[\mathbb{1}_{\{\gamma R(a,\theta) > C_4(\gamma)n\epsilon^2\}}]\right)^{1/2} \left(\mathbb{E}_\Pi[e^{2\gamma R(a,\theta)}]\right)^{1/2} \leq e^{-C_4(\gamma)n\epsilon^2}\mathbb{E}_\Pi[e^{2\gamma R(a,\theta)}]$, which implies that if the risk function is bounded in $(a, \theta)$, then the assumption is trivially satisfied. Finally, we also require the following condition lower bounding the risk function $R$.

**Assumption 2.5.** $R(a, \theta)$ *is assumed to satisfy* $W := \inf_{\theta \in \Theta} \inf_{a \in \mathcal{A}} e^{R(a,\theta)} > 0$.

Note that any risk function which is bounded from below in both the arguments satisfies this condition. In the next section, we establish high-probability bounds on the optimality gap in values and decision rules computed using RSVB approach for sufficiently large $n$.

## 3  Asymptotic Analysis of the Optimality Gaps

Our first result, establishes an upper bound on the expected deviation from the true model $P_0^n$, measured using distance function $L_n(\cdot, \theta_0)$, under the RSVB approximate posterior. We also note that the following result generalizes Theorem 2.1 of [34], which is exclusively for the case when $\gamma \to 0^+$. Our proof techniques follows that of Theorem 2.1 in [34].

**Theorem 3.1.** *Fix $a' \in \mathcal{A}$ and $\gamma > 0$. For any $L_n(\theta, \theta_0) \geq 0$, under Assumptions 2.1, 2.2, 2.3, 2.4, and 2.5, and for $\min(C, C_4(\gamma) + C_5(\gamma)) > C_2 + C_3 + C_4(\gamma) + 2$ and $\eta_n^R(\gamma) := \frac{1}{n}\inf_{Q \in \mathcal{Q}} \mathbb{E}_{P_0^n}\left[\mathrm{KL}(Q(\theta)\|\Pi(\theta|\tilde{X}_n)) - \gamma\inf_{a \in \mathcal{A}} \mathbb{E}_Q[R(a,\theta)]\right]$, for sufficiently large $n$ the RSVB approximator of the true posterior $Q_{a',\gamma}^*(\theta|\tilde{X}_n)$ satisfies,*

$$\mathbb{E}_{P_0^n}\left[\int_\Theta L_n(\theta, \theta_0)dQ_{a',\gamma}^*(\theta|\tilde{X}_n)\right] \leq n\left(M(\gamma)\epsilon_n^2 + M\eta_n^R(\gamma)\right), \tag{2}$$

*for a positive number $M(\gamma) = 2\left(C_1 + MC_4(\gamma)\right)$, where $M = \frac{2C_1}{\min(C,\lambda,1)}$.*

*Proof sketch.* For brevity we denote the likelihood ratio as $\mathcal{LR}_n(\theta, \theta_0) = \frac{p(\tilde{X}_n|\theta)}{p(\tilde{X}_n|\theta_0)}$. We prove this result using a series of lemma. The first Lemma C.1 separates the term $\eta_n^R(\gamma)$, which is later analyzed using Assumption 3.1 in Section 3.1. Lemma C.1 establish that

$$\zeta\mathbb{E}_{P_0^n}\left[\int_\Theta L_n(\theta, \theta_0)\, dQ_{a',\gamma}^*(\theta|\tilde{X}_n)\right] \leq \log\mathbb{E}_{P_0^n}\left[\int_\Theta e^{\zeta L_n(\theta,\theta_0)} \frac{e^{\gamma R(a',\theta)}\, \mathcal{LR}_n(\theta,\theta_0)d\Pi(\theta)}{\int_\Theta e^{\gamma R(a',\theta)}\, \mathcal{LR}_n(\theta,\theta_0)d\Pi(\theta)}\right]$$
$$+ \log\mathbb{E}_{P_0^n}\left[\int_\Theta e^{\gamma R(a',\theta)} \frac{\mathcal{LR}_n(\theta,\theta_0)d\Pi(\theta)}{\int_\Theta \mathcal{LR}_n(\theta,\theta_0)d\Pi(\theta)}\right] + n\eta_n^R(\gamma). \tag{3}$$

The proof of Lemma C.1 uses simple arguments that follow easily from Jensen's inequality and the definition of the posterior distribution.

To analyze the first term in the above display, we define the set $K_n := \{\theta \in \Theta : L_n(\theta, \theta_0) > C_1 n\epsilon^2\}$, with an aim to control the exponential moment of $L_n(\theta, \theta_0)$ by characterizing its tails. Also, notice that set $K_n$ is the set of alternate hypotheses as defined in Assumption 2.1 through Lemma C.2. For brevity, define $\Pi_R(K_n|\tilde{X}_n) := \frac{\int_{K_n} e^{\gamma R(a',\theta)}\, \mathcal{LR}_n(\theta,\theta_0)d\Pi(\theta)}{\int_\Theta e^{\gamma R(a',\theta)}\, \mathcal{LR}_n(\theta,\theta_0)d\Pi(\theta)}$. Next, we divide the expected calibrated posterior probability of the set $K_n$ as follows:

$$\mathbb{E}_{P_0^n}\left[\Pi_R(K_n|\tilde{X}_n)\right] \leq \mathbb{E}_{P_0^n}\phi_{n,\epsilon} + \mathbb{E}_{P_0^n}\left[\mathbb{1}_{B_n^C}\right] + \mathbb{E}_{P_0^n}\left[(1 - \phi_{n,\epsilon})\mathbb{1}_{B_n}\Pi_R(K_n|\tilde{X}_n)\right], \tag{4}$$

where recall that $\{\phi_{n,\epsilon}\}$ is the sequence of test function from Assumption 2.1 and set $B_n = \left\{ \tilde{X}_n : \int_\Theta \mathcal{LR}_n(\theta, \theta_0) d\Pi(\theta) \geq e^{-(1+C_3)n\epsilon^2} \Pi(A_n) \right\}$ with set $A_n$ is defined in Assumption 2.3. Set $B_n$ is introduced to separately control (by lower bounding) the denominator in the definition of the posterior distribution in the last term of (4). In addition, the Assumption 2.3 is used to show that $P_0^n(B_n^C) \leq e^{-\lambda n\epsilon^2}$. The first term can be controlled by Assumption 2.1 (i). Now, it remains to analyze the last term in (4). Using Assumption 2.3 and 2.5 observe that on set $B_n$, $\int_\Theta e^{\gamma R(a', \theta)} \mathcal{LR}_n(\theta, \theta_0) d\Pi(\theta) \geq W^\gamma e^{-(1+C_2+C_3)n\epsilon^2}$. This observation, with the application of Fubini's theorem, enables us to bifurcate further the last term in (4) on the set $S_\gamma(\theta) := \{ e^{\gamma R(a',\theta)} > e^{C_4(\gamma)n\epsilon^2} \}$ as

$$\mathbb{E}_{P_0^n} \left[ (1 - \phi_{n,\epsilon}) \mathbb{1}_{B_n} \Pi_R(K_n | \tilde{X}_n) \right] \leq W^{-\gamma} e^{\tilde{C} n\epsilon^2} \left[ \int_{K_n \cap S_\gamma(\theta)} \mathbb{E}_{P_\theta^n} \left[ (1 - \phi_{n,\epsilon}) \right] d\Pi(\theta) \right.$$
$$\left. + e^{-C_4(\gamma)n\epsilon^2} \int_{K_n \cap S_\gamma(\theta)} e^{\gamma R(a',\theta)} d\Pi(\theta) \right], \quad (5)$$

$\tilde{C} = 1 + C_2 + C_3 + C_4(\gamma)$. Observe that the last term above can be controlled using Assumption 2.4. The first integral in (5) can be further bounded by $\int_{K_n \cap \Theta_n(\epsilon)} \mathbb{E}_{P_\theta^n} \left[ (1 - \phi_{n,\epsilon}) \right] d\Pi(\theta) + \Pi(\Theta_n(\epsilon)^c)$, where $\Theta_n(\epsilon)$ is defined in Assumption 2.1 and both the terms can be exponentially bounded using Assumption 2.1 (ii) and 2.2 respectively. The last term in (3) can be bounded using a similar set of arguments and techniques. $\qquad \square$

*Remark:* We note that while deriving the bound in Lemma C.1, we can interchange the order of expectation and infimum and compute a tighter bound. However, this bring us back to the RSVB optimizer, the very objective of the whole analysis. Taking expectation before infimum enables us to derive a more interpretable and meaningful bound. This is mainly because it is easier to control the expectation of the RSVB objective than its infimum through Assumption 3.1, as presented in Section 3.1.

The detailed proof of Theorem 3.1 is provided in the Appendix C.2. Now recall that $\epsilon_n$ is the convergence rate of the true posterior [11, Theorem 7.3]. Notice that the additional term $\eta_n^R(\gamma)$ emerges from the posterior approximation and depends on the choice of the variational family $\mathcal{Q}$, risk function $R(\cdot, \cdot)$, and the parameter $\gamma$. The appearance of $\eta_n^R(\gamma)$ in the bound also signifies that to minimize the expected gap (under the RSVB posterior) between the true model and any other model (defined using $n^{-1} L_n(\theta, \theta_0)$) the expected RSVB objective has to be maximized. Later in this section, we specify the conditions that ensure $\eta_n^R(\gamma) \to 0$ as $n \to \infty$. Moreover, we also identify mild regularity conditions on $\mathcal{Q}$ to show that $\eta_n^R(\gamma)$ is $O(\epsilon_n^2)$ and show that as $\gamma$ increases $\eta_n^R(\gamma)$ decreases. We discuss this result and the bound therein later in the next subsection. Before that, we establish our main result (the bounds on the optimality gap) using the theorem above. We now fix

$$L_n(\theta, \theta_0) = n \left( \sup_{a \in \mathcal{A}} |R(a, \theta) - R(a, \theta_0)| \right)^2. \quad (6)$$

Notice that for a given $\theta$, $n^{-1/2} \sqrt{L_n(\theta, \theta_0)}$ is the uniform distance between the $R(a, \theta)$ and $R(a, \theta_0)$. Intuitively, Theorem 3.1 implies that the expected uniform difference $\frac{1}{n} L_n(\theta, \theta_0)$ with respect to the RSVB approximate posterior is $O(M(\gamma)\epsilon_n^2 + M\eta_n^R(\gamma))$, and if $M(\gamma)\epsilon_n^2 + M\eta_n^R(\gamma) \to 0$ as $n \to \infty$ then it converges to zero at that rate.

Also, note that in order to use (6) we must demonstrate that it satisfies Assumption 2.1. This can be achieved by constructing bespoke test functions for a given $R(a, \theta)$. We demonstrate this approach by an example in Section B.2. We also provide sufficient conditions for the existence of the test functions in the appendix. These conditions are typically easy to verify when the loss function $R(\cdot, \cdot)$ is bounded, for instance. Next, we bound the optimality gap between $R(a_{\mathsf{RS}}^*, \theta_0)$ and $V_0^*$.

**Theorem 3.2.** *Fix $\gamma > 0$. Suppose the set $\mathcal{A}$ is compact. Then, under Assumptions 2.1, 2.2, 2.3, 2.4, and 2.5, for $\min(C, C_4(\gamma) + C_5(\gamma)) > C_2 + C_3 + C_4(\gamma) + 2$ and any $\tau > 0$, the $P_0^n-$ probability of $\left\{ \tilde{X}_n : R(a_{\mathsf{RS}}^*, \theta_0) - \inf_{a \in \mathcal{A}} R(a, \theta_0) \leq 2\tau \left[ M(\gamma)\epsilon_n^2 + M\eta_n^R(\gamma) \right]^{\frac{1}{2}} \right\}$ is at least $1 - \tau^{-1}$, for a positive mapping $M(\gamma) = 2 \left( C_1 + MC_4(\gamma) \right)$, where $M = \frac{2C_1}{\min(C, \lambda, 1)}$ for sufficiently large $n$.*

## 3.1 Properties of $\eta_n^R(\gamma)$

Evidently, the bounds obtained in both the results that we have proved so far depend on $\eta_n^R(\gamma)$. Consequently, we establish some important properties of $\eta_n^R(\gamma)$ with respect to $n$ and $\gamma$, under additional regularity conditions. In order to characterize $\eta_n^R(\gamma)$, we specify conditions on variational family $\mathcal{Q}$ such that $\eta_n^R(\gamma) = O(\epsilon_n'^2)$, for some $\epsilon_n' \geq \frac{1}{\sqrt{n}}$ and $\epsilon_n' \to 0$. We impose the following condition on the variational family $\mathcal{Q}$ that lets us obtain a bound on $\eta_n^R(\gamma)$ in terms of $n$ and $\gamma$.

**Assumption 3.1.** *There exists a sequence of distributions $\{q_n(\cdot)\}$ in the variational family $\mathcal{Q}$ such that for a positive constant $C_9$,*

$$\frac{1}{n}\left[ \text{KL}\left(Q_n(\theta)\|\Pi(\theta)\right) + \mathbb{E}_{Q_n(\theta)}\left[ \text{KL}\left( dP_0^n(\tilde{X}_n)\|dP_\theta^n(\tilde{X}_n)\right)\right]\right] \leq C_9\epsilon_n'^2. \qquad (7)$$

If the observations in $\tilde{X}_n$ are i.i.d, then observe that $\frac{1}{n}\mathbb{E}_{Q_n(\theta)}\left[ \text{KL}\left( dP_0^n(\tilde{X}_n)\|dP_\theta^n(\tilde{X}_n)\right)\right] = \mathbb{E}_{Q_n(\theta)}\left[ \text{KL}\left( dP_0\|dP_\theta\right)\right]$. Intuitively, this assumption implies that the variational family must contain a sequence of distributions weakly convergent to a Dirac delta distribution concentrated at the true parameter $\theta_0$ otherwise the second term in the LHS of (7) will be non-zero. Also, note that the above assumption does not imply that the minimizing sequence $Q_{a',\gamma}^*(\theta|\tilde{X}_n)$ (automatically) converges weakly to a Dirac-delta distribution at the true parameter $\theta_0$. Furthermore, unlike Theorem 2.3 of [34], our condition on $\mathcal{Q}$ in Assumption 3.1, to obtain a bound on $\eta_n^R(\gamma)$, does not require the support of the distributions in $\mathcal{Q}$ to shrink to the true parameter $\theta_0$ at some appropriate rate, as the numbers of samples increases.

The condition that the variational family contains Dirac delta distributions at each point in the parameter space is a mild and reasonable requirement for consistency. Further, Assumption 3.1 requires that $\mathcal{Q}$ contains sequences of distributions that weakly converge to each Dirac delta distribution at a certain rate. This is easily satisfied if $\mathcal{Q}$ has no "holes", e.g. if it is the family of Gaussians with all means and variances, then we can always construct sequences converging to any Dirac delta at any rate. A similar assumption has also been made in [31], and is true for most exponential family distributions. For instance, in the newsvendor application, we fix the variational family to a class of shifted-Gamma distributions and choose a sequence of distributions with parameter sequence $\alpha = n$ and $\beta = n/\theta_0$. This implies that the sequence of distributions has mean $\theta_0$ and variance $\theta_0^2/n$, and, therefore, it converges to a Dirac delta distribution at $\theta_0$. Next, we show that

**Proposition 3.1.** *Under Assumption 3.1 and for a constant $C_8 = -\inf_{Q \in \mathcal{Q}} \inf_{a \in \mathcal{A}} \mathbb{E}_Q[R(a, \theta)]$ and $C_9 > 0$, $\eta_n^R(\gamma) \leq \gamma n^{-1}C_8 + C_9\epsilon_n'^2$.*

In Section 4, we present an example where the likelihood is exponentially distributed, the prior is inverse-gamma (non-conjugate), and the variational family is the class of gamma distributions, where we construct a sequence of distributions in the variational family that satisfies Assumption 3.1. We also provide another example where the likelihood is multivariate Gaussian with unknown mean and variational family is uncorrelated Gaussian restricted to a compact subset of $\mathbb{R}^d$ with a uniform prior on the same compact set satisfy Assumption 3.1.

By definition $\epsilon_n^2 \to 0$ and $\epsilon_n' \to 0$ as $n \to \infty$, and therefore it follows from Proposition 3.1 that $M(\gamma)\epsilon_n^2 + M\eta_n^R(\gamma) \to 0$. However, the bound obtained in the last proposition might be loose with respect to $\gamma$, when $C_8 < 0$. To see this, we prove the following result.

**Proposition 3.2.** *If the solution to the optimization problem in $\eta_n^R(\gamma)$ is primal feasible then $\eta_n^R(\gamma)$ decreases as $\gamma$ increases.*

## 4 Applications

In the examples below, we use three examples to study the interplay between sample size $n$ and the risk parameter $\gamma$, and their effect on the optimality gap in values. Additionally, we consider a multi-product newsvendor example in the Appendix B.2.

### 4.1 Single-product Newsvendor Model

We start with a canonical data-driven decision-making problem with a 'well-behaved' risk function $R(a, \theta)$, the data-driven newsvendor model. This problem has received extensive study in the literature and remains a cornerstone of inventory management [25, 2, 18]. Recall that the newsvendor

loss function is defined as $\ell(a, \xi) := h(a - \xi)^+ + b(\xi - a)^+$ where $h$ (underage cost) and $b$ (overage cost) are given positive constants, $\xi \in [0, \infty)$ the random demand, and $a$ the inventory or decision variable, typically assumed to take values in a compact decision space $\mathcal{A}$ with $\underline{a} := \min\{a : a \in \mathcal{A}\}$ and $\bar{a} := \max\{a : a \in \mathcal{A}\}$, and $\underline{a} > 0$. The distribution over the random demand, $P_\theta$ is assumed to be exponential with unknown rate parameter $\theta \in (0, \infty)$. The model risk $R(a, \theta) := \mathbb{E}_{P_\theta}[\ell(a, \xi)] = ha - \frac{h}{\theta} + (b+h)\frac{e^{-a\theta}}{\theta}$, which is convex in $a$. We assume that $\tilde{X}_n := \{\xi_1, \xi_2 \ldots \xi_n\}$ be $n$ observations of the random demand, assumed to be i.i.d random samples drawn from $P_0$.

We fix the model space $\Theta = [T, \infty)$ for some $T > 0$ and assume that $\theta_0$ lies in the interior of $\Theta$. We now assume a non-conjugate truncated inverse-gamma (Inv $-\Gamma$) prior distribution restricted to $\Theta$, with shape and rate parameter $\alpha$ and $\beta$ respectively, that is for a set $A \subseteq \Theta$, we define $\Pi(A) = $ Inv$-\Gamma_\Theta(A; \alpha, \beta) = $ Inv$-\Gamma(A \cap \Theta; \alpha, \beta)/$Inv$-\Gamma(\Theta; \alpha, \beta)$. We verify Assumptions 2.2, 2.1, 2.3, 2.5 and 2.4 (in that order) in this newsvendor setting and provide the proofs in the Appendix B.1. Next, we bound the optimality gap in values for the single product newsvendor model risk.

**Theorem 4.1.** *Fix $\gamma > 0$. Suppose that the set $\mathcal{A}$ is compact. Then, for the newsvendor model with exponentially distributed demand with rate $\theta \in \Theta = [T, \infty)$, prior distribution $\Pi(\cdot) = $ Inv$-\Gamma_\Theta(\cdot; \alpha, \beta) = $ Inv$-\Gamma(A \cap \Theta; \alpha, \beta)/$Inv$-\Gamma(\Theta; \alpha, \beta)$, and the variational family fixed to shifted (by $T > 0$) gamma distributions, and for any $\tau > 0$, the $P_0^n-$ probability of the following event*
$$\left\{ \tilde{X}_n : R(\mathtt{a}_{\mathtt{RS}}^*, \theta_0) - \inf_{z \in \mathcal{A}} R(z, \theta_0) \leq 2\tau M'(\gamma) \left( \frac{\log n}{n} \right)^{1/2} \right\} \text{ is at least } 1 - \tau^{-1} \text{ for sufficiently}$$
*large $n$ and for some known mapping $M' : \mathbb{R}^+ \to \mathbb{R}^+$, where $R(\cdot, \theta)$ is the newsvendor model risk.*

The proof of the theorem above is a direct consequence of Theorem 3.2 and Proposition 3.2, and Lemmas B.1, B.2, B.3, B.4, B.5 stated in the Appendix. We also extend the analysis above to a multi-product newsvendor problem. The details are provided in Appendix B.2.

Next, we demonstrate the effect of varying the risk-sensitivity parameter $\gamma$. We fix $\theta_0 = 0.1$, $b = 1$, $h = 5$, $\alpha = 1$, and $\beta = 4.1$. We run RSVB algorithm with $\gamma \in \{0(\text{ naive }), 1, 2, 4.5, 5, 6\}$ and repeat the experiment over 100 sample paths. We plot the results in Figure 1. In Figure 1(a) we plot the optimality gap in values that is $R(\mathtt{a}_{\mathtt{RS}}^*(\gamma), \theta_0) - R(a_0^*, \theta_0)$ for various values of $\gamma$. We observe that the gap decreases when $n$ increases. This observation supports our results in Propositions 3.1 and 3.2 that establishes the properties of $\eta_n^R(\gamma)$ as $n$ increases. Lastly, in Figure 1(b), we plot the variance of the RSVB posterior as $n$ increases for various values of $\gamma$; as anticipated, the variance reduces as $n$ increases. To observe the effect of $\gamma$, first recall that as $\gamma$ increases the decision maker becomes more risk averse, and so is our algorithmic framework RSVB. Indeed, from the rightmost variance plot in Figure 1, it is evident that for a larger value of $\gamma$ ($> 4$), the RSVB posterior is more concentrated on the subset of $\Theta$, where risk is more and consequently we observe large optimality gaps in values. Moreover, as $n$ increases, the effect of larger $\gamma$ reduces, since as $n$ increases, the incentive to deviate from the posterior reduces (due to increased KL divergence dominance for larger $n$ in RSVB). We also observe that the decision rule learned using the conjugate prior (Gamma distribution for exponential models) has a similar performance as the naïve approach. However, the variance of the true conjugate posterior is higher than those computed through the RSVB and VI approach, corresponding to the well known fact that the variational approximations underestimate the true posterior variance ([28, 19]).

## 4.2 Gaussian process classification

Consider a problem of classifying an input pattern or features $Y$ lying in measure space $([0, 1]^d, \mathcal{Y}, \nu)$ (with sigma algebra $\mathcal{Y}$ and probability measure $\nu$) into one of two classes $\{-1, 1\}$. Let $Y \mapsto \xi(Y) \in \{-1, 1\}$ denote the class of $Y$. For a given $Y$, we model the classifier using a Bernoulli distribution $p(\xi|Y, \theta) = \Psi_\xi(\theta(Y))$, where $\theta : [0, 1]^d \to \mathbb{R}$ is a non-parametric model parameter in a separable Banach space $(\Theta, \|\cdot\|)$ and measurable functions $\Psi_1(x) = (1 + e^{-x})^{-1}$ and $\Psi_{-1}(x) = 1 - \Psi_1(x)$. Note that $\Psi_1(\cdot)$ is a logistic function. We denote $\psi(\cdot)$ as the derivative of $\Psi_1(\cdot)$. Assume that the features $Y$ are generated independently of $\xi$. Thus the sequence of independent observations $\{\tilde{Y}_n, \tilde{X}_n\} = \{(Y_1, \xi_1), (Y_2, \xi_2), \ldots, (Y_n, \xi_n)\}$ are assumed to be generated from the model $dP_\theta(a, y) = P_\theta(\xi = a, Y \in dy) = p(\xi = a|Y = y, \theta)\nu(dy)$. In the above binary classification problem, the objective is to estimate $\theta(\cdot)$ using the observation vector $\{\tilde{Y}_n, \tilde{X}_n\}$.

We posit a Gaussian process (GP) prior $\Pi(\cdot)$ on $\theta(\cdot) \in \Theta$ defined as $W(\cdot) = \sum_{j=1}^{\bar{J}_\alpha} \sum_{k=1}^{2^{jd}} \mu_j Z_{j,k} \vartheta_{j,k}(\cdot)$, where $\{\mu_j\}$ is a sequence that decreases with $j$, $\{Z_{i,j}\}$ are i.i.d. standard

Gaussian random variables and $\{\vartheta_{j,k}\}$ form a double-indexed orthonormal basis (with respect to measure $\nu$), that is $\mathbb{E}_\nu[\vartheta_{j,k}\vartheta_{l,m}] = \mathbb{1}_{\{j=l,k=m\}}$). $\bar{J}_\alpha$ is the smallest integer satisfying $2^{\bar{J}_\alpha d} = n^{d/(2\alpha+d)}$ for a given $\alpha > 0$. This prior construction is motivated from the work in [30]. We also assume that $\nu(\cdot)$ is known, and we do not place any prior on it. The posterior distribution over $\theta(\cdot)$ given observations $\{\tilde{Y}_n, \tilde{X}_n\}$ can be defined as $d\Pi(\theta|\{\tilde{Y}_n, \tilde{X}_n\}) = \frac{d\Pi(\theta)\prod_{i=1}^n \Psi_{\xi_i}(\theta(Y_i))\nu(Y_i)}{\int \prod_{i=1}^n \Psi_{\xi_i}(\theta(Y_i))\nu(Y_i)d\Pi(\theta)} = \frac{d\Pi(\theta)\prod_{i=1}^n \Psi_{\xi_i}(\theta(Y_i))}{\int \prod_{i=1}^n \Psi_{\xi_i}(\theta(Y_i))d\Pi(\theta)}$. Consider the loss function $\ell(a, \xi) = \{0, \text{ if } a = \xi; \quad c_+, \text{if } a = +1, \xi = -1; \quad c_-, \text{if } a = -1, \xi = +1\}$, where $c_+$ and $c_-$ are known positive constants. For a given feature $Y$, the model risk is given by

$$R(a, \theta) = \mathbb{E}_{P_\theta}[\ell(a, \xi)] = \begin{cases} c_+\mathbb{E}_\nu[\Psi_{-1}(\theta(Y))], & a = +1, \\ c_-\mathbb{E}_\nu[\Psi_1(\theta(Y))], & a = -1. \end{cases} \tag{8}$$

We fix the variational family $\mathcal{Q}_{GP}$ is a class of Gaussian distributions on $\Theta$, defined as $\mathcal{N}(m_q, \mathcal{C}_q)$, $m_q$ belongs to $\Theta$ and $\mathcal{C}_q$ is the covariance operator defined as $\mathcal{C}_q = \mathcal{C}^{1/2}(I - S)\mathcal{C}^{1/2}$, for any $S$ which is a symmetric and Hilbert-Schmidt (HS) operator on $\Theta$ (eigenvalues of HS operator are square summable). Note that $S$ and $m_q$ span the distributions in $\mathcal{Q}_{GP}$. We can show, using the technical lemmas derived in Section B.3 in Appendix, that the optimality gap in values of the binary GP classification problem converges to zero at a minimax optimal rate (upto logarithmic factors).

**Theorem 4.2.** *Fix $\gamma > 0$ and for a given $J \in \mathbb{N}$. For the binary GP classification problem with GP prior induced by $W = \sum_{j=1}^J \sum_{k=1}^{2^{jd}} \mu_j Z_{j,k}\vartheta_{j,k}$, where $\mu_j = 2^{-jd/2-j\mathsf{a}}$ for some $\mathsf{a} > 0$, $\{Z_{i,j}\}$ are i.i.d. standard Gaussian random variables and $\{\vartheta_{j,k}\}$ form a double-indexed orthonormal basis (with respect to measure $\nu$), and $\|\theta_0\|_{\beta;\infty,\infty} < \infty$, and $\theta_0^J(y)$ lie in the Cameron-Martin space $Im(\mathcal{C}^{1/2})$, the variational family $\mathcal{Q}_{GP}$, and for any $\tau > 0$, the $P_0^n-$ probability of $\{\tilde{X}_n : R(\mathsf{a}_{RS}^*, \theta_0) - \inf_{z \in \mathcal{A}} R(z, \theta_0) \leq 2\tau M'(\gamma)\epsilon_n\}$ is at least $1 - \tau^{-1}$ for sufficiently large $n$ and for some mapping $M' : \mathbb{R}^+ \to \mathbb{R}^+$, where $R(\cdot, \theta)$ is defined in (8) and*

$$\epsilon_n = \begin{cases} n^{-\frac{\beta}{(2\alpha+d)}}\log n \quad \forall \beta \in [\mathsf{a}, \alpha]; \quad n^{-\frac{\alpha}{(2\alpha+d)}}\log n \quad \forall \alpha \in [\mathsf{a}, \beta]; \\ n^{-\frac{\mathsf{a}}{(2\mathsf{a}+d)}}(\log n)^{\frac{d}{(2\mathsf{a}+d)}}, \quad \forall \mathsf{a} \in [\alpha, \beta]; \quad n^{-\frac{\beta}{(2\mathsf{a}+d)}}(\log n)^{\frac{d}{(2\mathsf{a}+d)}} \quad \forall \beta \in [\alpha, \mathsf{a}]. \end{cases} \tag{9}$$

A similar Gaussian process classification problem was studied empirically in [16].

### 4.3 Eight-schools model

We consider the eight-schools problem [33, 15], where the objective is to study the effectiveness of the special coaching program for SAT exams, offered in 8 schools. Each school reported the treatment effect $y_i$ and its standard deviation $\sigma_i$, where $i \in \{1, 2, 3, \ldots, 8\}$. The observations $\{y_i, \sigma_i\}_{i=1}^8$ are modeled using the following hierarchical Bayesian model : 1) Prior distributions: $\mu \sim \mathcal{N}(\cdot|0, 5)$, $\tau \sim$ half-Cauchy$(\cdot|0, 5)$, 2) $\theta_i \sim \mathcal{N}(\cdot|\mu, \tau^2)$ and $y_i \sim \mathcal{N}(\cdot|\theta_i, \sigma_i^2)$ for each $i \in \{1, 2, 3, \ldots, 8\}$, where $\{\sigma_i\}_{i=1}^8$ are assumed to be a known sequence of covariates. The posterior distribution is defined as $\pi_8(\mu, \tau, \{\theta_i\}_{i=1}^8|\{y_i, \sigma_i\}_{i=1}^8) \propto \mathcal{N}(\cdot|0, 5)$half-Cauchy$(\cdot|0, 5)\prod_{i=1}^8 \mathcal{N}(\theta_i|\mu, \tau^2)\mathcal{N}(y_i|\theta_i, \sigma_i^2)$. For a given treatment effect for eight schools $y \in \mathbb{R}^8$, the loss function is defined as:

$$\ell(y, a) = \sum_{i=1}^8 l(y_i, a_i), \text{ where } l(y_i, a_i) = \begin{cases} 0.2\,|a_i - y_i|, \; y_i \geq a_i \\ (1 - 0.2)\,|a_i - y_i|, \; y_i < a_i, \end{cases} \tag{10}$$

where the decision variable $a = \{a_1, a_2, \ldots, a_8\}$ denotes the effectiveness level. We define $R(a, \theta) = \log \int p(y|\theta, \{\sigma_i\}_{i=1}^8)\ell(y, a)dy$, $\theta \in \mathbb{R}^8$ considering $\{\sigma_i\}_{i=1}^8$ is a given sequence of covariates. Now for any $\gamma > 0$, RSVB joint-optimization problem is defined as $\min_{a \in \mathcal{A}} \max_{Q \in \mathcal{Q}} \left\{\mathbb{E}_q[R(a, \theta)] - \frac{1}{\gamma}KL(q||\pi_8)\right\}$. We fix the variational family $\mathcal{Q}$ to be the mean-field variational family, that is $q_\Lambda(\mu, \tau, \theta_{1\,i=1}^8) = \mathcal{N}_\mu(\cdot|\lambda_9)\mathcal{N}_\tau(\cdot|\lambda_{10})\prod_{i=1}^8 \mathcal{N}_{\theta_i}(\cdot|\lambda_i)$, where $\Lambda = \{\lambda_1, \lambda_2, \ldots, \lambda_{10}\}$, $\lambda_i$ is the mean and variance parameter for each Gaussian distribution. Following [15], we measure the performance of the RSVB method using the metric called *empirical risk reduction* (ERR) , $\mathcal{I} = \mathcal{ER}_{VB} - \mathcal{ER}_{RSVB(\gamma)}$, where , $\mathcal{ER}_{alg} = \frac{1}{N_Y}\sum_{j=1}^{N_Y} l(y_j^{test}, a_{alg})$, where $\mathcal{ER}_{alg}$ denote the empirical risk evaluated at the decision rule $a^{alg}$ obtained using method alg $\in$ {VB,RSVB$(\gamma)$}

for various values of $\gamma$, and $N_Y$ is the size of the test data $\{y_j^{\text{test}}\}$. Note that ERR is the empirical approximation of the difference of optimality gap between the two-step näive VB approach and RSVB($\gamma$) approach. Recall, in the näive VB method, we first compute the KL minimizer of the true posterior and then compute the optimal decision using this approximate posterior. Following [15], due to small size of the dataset in this example, the ERR is evaluated on the training data itself.

We modified the experiments in [15] by introducing $\gamma = \{5, 2.5, 1, 0.5\} = \{0.2^{-1}, 0.4^{-1}, 1^{-1}, 2^{-1}\}$ and obtain the results as summarized in Figure 2(a) and (b). In Figure 2(a), we observe that as $\gamma$ increases, the ERR also increases, which implies that the decisions are more optimistic as empirical loss of the RSVB decision rule decreases as $\gamma$ increases. Also, observe from Figure 2(b) that, as $\gamma$ increases, the RSVB posterior (joint marginal posterior distritbution for $(\theta_3, \tau)$) approaches the true posterior, and the variance of the approximate posterior also reduces.

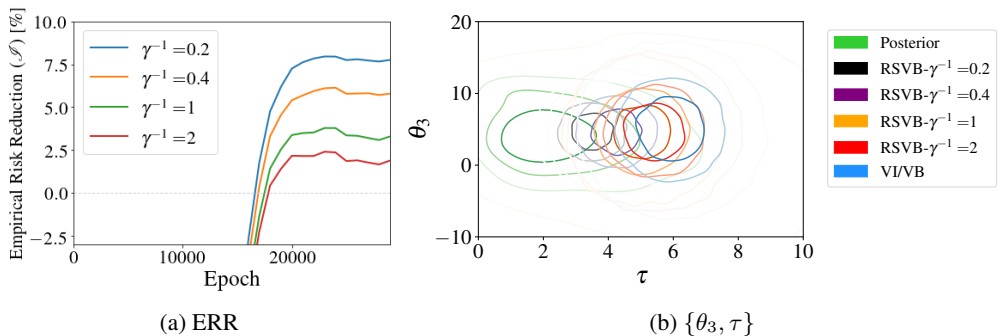

(a) ERR  (b) $\{\theta_3, \tau\}$

Figure 2: a) Empirical risk reduction plot for different level of risk sensitivity , b) Joint RSVB posterior distribution of $\{\theta_3, \tau\}$, plotted for different level of risk sensitivity.

Notice that it is not obvious that the variance of the RSVB posterior will reduce as $\gamma$ increases. We believe that it depends on the landscape of the expected risk function and the choice of the variational family. Intuitively, a possible explanation of this phenomenon can be provided using the equation (RSVB). Consider the RSVB formulation and note that KL $> 0$, therefore as $\gamma$ increases, there is more incentive to deviate from the true posterior and choose $Q \in \mathcal{Q}$ that maximizes expected risk for a given $a \in \mathcal{A}$. Note that a $Q$ that places more mass near the $\theta$ that maximizes the risk will be preferred over the one with more spread.

## 5 Discussion and Future Work

The RSVB formulation as stated in this paper requires us to solve a stochastic minimax optimization problem to compute a decision rule. This is often difficult to solve, particularly in the nonconvex-nonconcave setting of RSVB, and indeed, we are not aware of any computationally efficient methods for solving such problems in all generality [8]. To circumvent this, the authors in the LCVB literature maximize utility instead of minimizing risk, and thus convert the whole problem to a much simpler max-max optimization problem. If we replace risk with utility in the RSVB derivation, we will also get a max-max problem. In this formulation, as you increase $\gamma$, the incentive to deviate from the posterior and maximize the maximum utility increases. Therefore, increasing $\gamma$ corresponds to being more optimistic in making decisions than being more risk averse in the risk setting, as observed in the empirical results for the *eight schools model* in Section 4.3.

We note that we assumed the risk-function is lower-bounded, this is a natural assumption to place on risk functions. Converting our problem to a max-max problem would further require the risk to be upper bounded, which is also assumed for the methodologies presented (without theory) in [16, 15]. We emphasize that our theoretical results will continue to hold in this setting, but since our emphasis was on the statistical aspects of this problem, we chose to present our results with minimum assumptions. Developing an algorithm for solving general minimax optimization in RSVB without transforming it into a max-max problem is open and a part of our future work. Also an interesting topic for future work is identifying minimum additional assumptions to theoretically characterize computational aspects of this problem.

## 6 Acknowledgement

We thank the anonymous reviewers for their helpful comments and discussions. This work is supported by the National Science Foundation under grant DMS-1812197.

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

# A  Additional definitions

We provide the definitions of important terms used throughout the paper. First, recall the definition of covering numbers:

**Definition A.1** (Covering numbers). *Let $\mathcal{P} := \{P_\theta, \theta \in \Theta\}$ be a parametric family of distributions and $d : \mathcal{P} \times \mathcal{P} \mapsto [0, \infty)$ be a metric. An $\epsilon-$cover of a subset $\mathcal{P}_K := \{P_\theta : \theta \in K \subset \Theta\}$ of the parametric family of distributions is a set $K' \subset K$ such that, for each $\theta \in K$ there exists a $\theta' \in K'$ that satisfies $d(P_\theta, P_{\theta'}) \leq \epsilon$. The $\epsilon-$covering number of $\mathcal{P}_K$ is $N(\epsilon, \mathcal{P}_K, d) = \min\{card(K') : K' \text{ is an } \epsilon-cover \text{ of } K\}$, where $card(\cdot)$ represents the cardinality of the set.*

Next, recall the definition of a test function [26]:

**Definition A.2** (Test function). *Let $\tilde{X}_n$ be a sequence of random variables on measurable space $(\bigotimes_n \mathcal{X}, \mathcal{S}^n)$. Then any $\mathcal{S}^n$-measurable sequence of functions $\{\phi_n\}$, $\phi_n : \tilde{X}_n \mapsto [0, 1]\ \forall n \in \mathbb{N}$, is a test of a hypothesis that a probability measure on $\mathcal{S}^n$ belongs to a given set against the hypothesis that it belongs to an alternative set. The test $\phi_n$ is consistent for hypothesis $P_0^n$ against the alternative $P^n \in \{P_\theta^n : \theta \in \Theta \backslash \{\theta_0\}\}$ if $\mathbb{E}_{P^n}[\phi_n] \to \mathbb{1}_{\{\theta \in \Theta \backslash \{\theta_0\}\}}(\theta), \forall \theta \in \Theta \text{ as } n \to \infty$, where $\mathbb{1}_{\{\cdot\}}$ is an indicator function.*

A classic example of a test function is $\phi_n^{\text{KS}} = \mathbb{1}_{\{\text{KS}_n > K_\nu\}}(\theta)$ that is constructed using the Kolmogorov-Smirnov statistic $\text{KS}_n := \sup_t |\mathbb{F}_n(t) - \mathbb{F}_\theta(t)|$, where $\mathbb{F}_n(t)$ and $\mathbb{F}_\theta(t)$ are the empirical and true distribution respectively, and $K_\nu$ is the confidence level. If the null hypothesis is true, the Glivenko-Cantelli theorem [29, Theorem 19.1] shows that the KS statistic converges to zero as the number of samples increases to infinity.

Furthermore, we define the Hellinger distance $h(\theta_1, \theta_2)$ between the two probability distributions $P_{\theta_1}$ and $P_{\theta_2}$ is defined as $d_H(\theta_1, \theta_2) = \left( \int \left( \sqrt{dP_{\theta_1}} - \sqrt{dP_{\theta_2}} \right)^2 \right)^{1/2}$. We define the one-sided Hausdorff distance $H(A\|B)$ between sets $A$ and $B$ in a metric space $D$ with distance function $d$ is defined as:

$$H(A\|B) = \sup_{x \in A} d_h(x, B), \text{ where } d_h(x, B) = \inf_{y \in B} d(x, y).$$

Next, we define an arbitrary loss function $L_n : \Theta \times \Theta \mapsto \mathbb{R}$ that measures the distance between models $(P_{\theta_1}^n, P_{\theta_2}^n) \forall \{\theta_1, \theta_2\} \in \Theta$. At the outset, we assume that $L_n(\theta_1, \theta_2)$ is always positive. We define $\{\epsilon_n\}$ as a sequence such that $\epsilon_n \to 0$ as $n \to \infty$ and $n\epsilon_n^2 \geq 1$.

We also define

**Definition A.3** ($\Gamma-$convergence). *A sequence of functions $F_n : \mathcal{U} \mapsto \mathbb{R}$, for each $n \in \mathbb{N}$, $\Gamma-$converges to $F : \mathcal{U} \mapsto \mathbb{R}$, if*

- *for every $u \in \mathcal{U}$ and every $\{u_n, n \in \mathbb{N}\}$ such that $u_n \to u$, $F(x) \leq \liminf_{n \to \infty} F_n(u_n)$;*

- *for every $u \in \mathcal{U}$, there exists some $\{u_n, n \in \mathbb{N}\}$ such that $u_n \to u$, $F(x) \geq \limsup_{n \to \infty} F_n(u_n)$.*

In addition, we define

**Definition A.4** (Primal feasibility). *For any two functions $f : \mathcal{U} \mapsto \mathbb{R}$ and $b : \mathcal{U} \mapsto \mathbb{R}$, a point $u^* \in \mathcal{U}$ is primal feasible to the following constraint optimization problem*

$$\inf_{u \in \mathcal{U}} f(u) \text{ subject to } b(u) \leq c,$$

*if $b(u^*) \leq c$, for a given $c \in \mathbb{R}$.*

# B  Applications

## B.1  Single product newsvendor problem (cont.)

First, we fix the sieve set $\Theta_n(\epsilon) = \Theta$, which clearly implies that the restricted inverse-gamma prior $\Pi(\theta)$, places no mass on the complement of this set and therefore satisfies Assumption 2.2.

Second, under the condition that the true demand distribution is exponential with parameter $\theta_0$ (and $P_0 \equiv P_{\theta_0}$), we demonstrate the existence of test functions satisfying Assumption 2.1.

**Lemma B.1.** *Fix $n \geq 5$. Then, for any $\epsilon > \epsilon_n := \frac{1}{\sqrt{n}}$ with $\epsilon_n \to 0$, and $n\epsilon_n^2 \geq 1$, there exists a test function $\phi_n$ (depending on $\epsilon$) such that $L_n^{NV}(\theta, \theta_0) = n\left(\sup_{a \in \mathcal{A}} |R(a,\theta) - R(a,\theta_0)|\right)^2$ satisfies Assumption 2.1 with $C_0 = 20$ and $C = \frac{C_1}{2}(K_1^{NV})^{-2}$ for a constant $C_1 > 0$ and $K_1^{NV} = d_H(T,\theta_0)^{-1}\left[\left(\frac{h}{\theta_0} - \frac{h}{T}\right)^2 + (b+h)^2\left(\frac{e^{-aT}}{T} - \frac{e^{-a\theta_0}}{\theta_0}\right)^2\right]^{1/2}$.*

The proof of the above result follows by showing that $d_L^{NV} = n^{-1/2}\sqrt{L_n^{NV}(\theta, \theta_0)}$ can be bounded above by the Hellinger distance between two exponential distributions on $\Theta$ (under which a test function exists) in Lemma C.10 in the appendix.

Third, we show that there exist appropriate constants such that the inverse-gamma prior satisfies Assumption 2.3 when the demand distribution is exponential.

**Lemma B.2.** *Fix $n_2 \geq 2$ and any $\lambda > 1$. Let $A_n := \{\theta \in \Theta : D_{1+\lambda}(P_0^n \| P_\theta^n) \leq C_3 n\epsilon_n^2\}$, where $D_{1+\lambda}(P_0^n \| P_\theta^n)$ is the Rényi divergence between $P_0^n$ and $P_\theta^n$. Then for $\epsilon_n^2 = \frac{\log n}{n}$ and any $C_3 > 0$ such $C_2 = \alpha C_3 \geq 2$, the truncated inverse-gamma prior $Inv - \Gamma_\Theta(A; \alpha, \beta)$ satisfies $\Pi(A_n) \geq \exp(-nC_2\epsilon_n^2), \forall n \geq n_2$.*

Fourth, it is straightforward to see that the newsvendor model risk $R(a,\theta)$ is bounded below for a given $a \in \mathcal{A}$.

**Lemma B.3.** *For any $a \in \mathcal{A}$ and positive constants $h$ and $b$, the newsvendor model risk $R(a,\theta) = \left(ha - \frac{h}{\theta} + (b+h)\frac{e^{-a\theta}}{\theta}\right) \geq \left(\frac{h\underline{a}^2\theta^*}{(1+a\theta^*)}\right)$, where $\underline{a} := \min\{a \in \mathcal{A}\}$ and $\theta^*$ satisfies $h - (b+h)e^{-a\theta^*}(1+a\theta^*) = 0$.*

This implies that $R(a,\theta)$ satisfies Assumption 2.5. Finally, we also show that the newsvendor model risk satisfies Assumption 2.4.

**Lemma B.4.** *Fix $n \geq 1$ and $\gamma > 0$. For any $\epsilon > \epsilon_n$ and any $a \in \mathcal{A}$, $R(a,\theta)$ satisfies $\mathbb{E}_\Pi[\mathbb{1}_{\{R(a,\theta)\gamma > C_4(\gamma)n\epsilon^2\}}e^{\gamma R(a,\theta)}] \leq \exp(-C_5(\gamma)n\epsilon^2)$, for any $C_4(\gamma) > 2\gamma\left(h\bar{a} + \frac{b}{T}\right)$ and $C_5(\gamma) = C_4(\gamma) - 2\gamma\left(h\bar{a} + \frac{b}{T}\right)$, where $\bar{a} := \max\{a \in \mathcal{A}\}$.*

Note that Lemma B.1 implies that $C = \frac{C_1}{2(K_1^{NV})^2}$ for any constant $C_1 > 0$. Fixing $\alpha = 1$ and using Lemma B.2 we can choose $C_2 = C_3 = 2$. Now, $C_1$ can be chosen large enough such that $C > C_4(\gamma) + C_5(\gamma)$ for a given risk sensitivity $\gamma > 0$. Therefore, the condition on constants in Theorem 3.1 reduces to $C_5(\gamma) > 2 + C_2 + C_3 = 5$, and it can be satisfied easily by fixing $C_5(\gamma) = 5.1$(say).

These lemmas show that when the demand distribution is exponential and with a non-conjugate truncated inverse-gamma prior, our result in Theorem 3.2 can be used for RSVB method to bound the optimality gap in decisions and values for various values of the risk-sensitivity parameter $\gamma$. Recall that the bound obtained in Theorem 3.2 depends on $\epsilon_n^2$ and $\eta_n^R(\gamma)$.

Lemma B.2 implies that $\epsilon_n^2 = \frac{\log n}{n}$, but in order to get the complete bound we further need to characterize $\eta_n^R(\gamma)$. Recall that, as a consequence of Assumption 3.1 in Proposition 3.1, for a given $C_8 = -\inf_{Q \in \mathcal{Q}} \inf_{a \in \mathcal{A}} \mathbb{E}_Q[R(a,\theta)]$ that $C_9 > 0$ and $\eta_n^R(\gamma) \leq \gamma n^{-1}C_8 + C_9\epsilon_n'^2$.

Therefore, in our next result, we show that in the newsvendor setting, we can construct a sequence $\{Q_n(\theta)\} \subset \mathcal{Q}$ that satisfies Assumption 3.1, and thus identify $\epsilon_n'$ and the constant $C_9$. We fix $\mathcal{Q}$ to be the family of shifted gamma distributions with support $[T, \infty)$.

**Lemma B.5.** *Let $\{Q_n(\theta)\}$ be a sequence of shifted gamma distributions with shape parameter $a = n$ and rate parameter $b = \frac{n}{\theta_0}$, then for truncated inverse gamma prior and exponentially distributed likelihood model*

$$\frac{1}{n}\left[\mathrm{KL}\left(Q_n(\theta)\|\Pi(\theta)\right) + \mathbb{E}_{Q_n(\theta)}\left[\mathrm{KL}\left(dP_0^n(\tilde{X}_n))\|dP_\theta^n(\tilde{X}_n)\right)\right]\right] \leq C_9\epsilon_n'^2,$$

*where $\epsilon_n'^2 = \frac{\log n}{n}$ and $C_9 = \frac{1}{2} + \max\left(0, 2 + \frac{2\beta}{\theta_0} - \log\sqrt{2\pi} - \log\left(\frac{\beta^\alpha}{\Gamma(\alpha)}\right) + \alpha\log\theta_0\right)$ and prior parameters are chosen such that $C_9 > 0$.*

## B.2 Multi-product newsvendor problem

Analogous to the one-dimensional newsvendor loss function, the loss function in its multi-product version is defined as

$$\ell(a, \xi) := h^T (a - \xi)^+ + b^T (\xi - a)^+$$

where $h$ and $b$ are given vectors of underage and overage costs respectively for each product and mapping $(\cdot)^+$ is defined component-wise. We assume that there are $d$ items or products and $\xi \in \mathbb{R}^d$ denotes the random vector of demands. Let $a \in \mathcal{A} \subset \mathbb{R}^d_+$ be the inventory or decision variable, typically assumed to take values in a compact decision space $\mathcal{A}$ with $\underline{a} := \{\{\min\{a_i : a_i \in \mathcal{A}_i\}\}_{i=1}^d$ and $\bar{a} := \{\{\max\{a_i : a_i \in \mathcal{A}_i\}\}_{i=1}^d$, and $\underline{a} > 0$, where $\mathcal{A}_i$ is the marginal set of $i^{th}$ component of $\mathcal{A}$. The random demand is assumed to be multivariate Gaussian, with unknown mean parameter $\theta \in \mathbb{R}^d$ but with known covariance matrix $\Sigma$. We also assume that $\Sigma$ is a symmetric positive definite matrix and can be decomposed as $Q^T \Lambda Q$, where $Q$ is an orthogonal matrix and $\Lambda$ is a diagonal matrix consisting of respective eigenvalues of $\Sigma$. We also define $\overline{\Lambda} = \max_{i \in \{1,2,...d\}} \Lambda_{ii}$ and $\underline{\Lambda} = \min_{i \in \{1,2,...d\}} \Lambda_{ii}$. The model risk

$$R(a, \theta) = \mathbb{E}_{P_\theta}[\ell(a, \xi)] = \sum_{i=1}^{d} \mathbb{E}_{P_{\theta_i}} [h_i(a_i - \xi_i)^+ + b_i(\xi_i - a_i)^+]$$

$$= \sum_{i=1}^{d} \left[ (h_i + b_i) a_i \Phi \left( \frac{(a_i - \theta_i)}{\sigma_{ii}} \right) - b_i a_i + \theta_i (b_i - h_i) \right.$$

$$\left. + \sigma_{ii} \left[ h \frac{\phi \left( \frac{(a_i - \theta_i)}{\sigma_{ii}} \right)}{\Phi \left( \frac{(a_i - \theta_i)}{\sigma_{ii}} \right)} + b \frac{\phi \left( \frac{(a_i - \theta_i)}{\sigma_{ii}} \right)}{1 - \Phi \left( \frac{(a_i - \theta_i)}{\sigma_{ii}} \right)} \right] \right], \tag{11}$$

which is convex in $a$. Here $P_{\theta_i}$ is the marginal distribution of $\xi$ for $i^{th}$ product, $\phi(\cdot)$ and $\Phi(\cdot)$ are probability and cumulative distribution function of the standard Normal distribution. We also assume that the true mean parameter $\theta_0$ lies in a compact subspace $\Theta \subset \mathbb{R}^d$. We fix the prior to be uniformly distributed on $\Theta$ with no correlation across its components, that is $\pi(A) = \frac{m(A \bigcap \Theta)}{m(\Theta)} = \prod_{i=1}^d \frac{m(A_i \bigcap \Theta_i)}{m(\Theta_i)}$, where $m(B)$ is the Lebesgue measure (or volume) of $B \subset \mathbb{R}^d$ As in the previous example, we fix the sieve set $\Theta_n(\epsilon) = \Theta$, which clearly implies that $\Pi(\theta)$ places no mass on the complement of this set and therefore satisfies Assumption 2.2.

Then under the condition that the true demand distribution has a multivariate Gaussian distribution (with known $\Sigma$) and mean $\theta_0$ ($P_0 \equiv P_{\theta_0}$), we demonstrate the existence of test functions satisfying Assumption 2.1 by constructing a test function unlike the single-product newsvendor problem with exponential demand.

**Lemma B.6.** *Fix $n \geq 1$. Then, for any $\epsilon > \epsilon_n := \frac{1}{\sqrt{n}}$ with $\epsilon_n \to 0$, and $n\epsilon_n^2 \geq 1$ and test function $\phi_{n,\epsilon} := \mathbb{1}_{\left\{ \tilde{X}_n : \|\hat{\theta}_n - \theta_0\| > \sqrt{\tilde{C}\epsilon^2} \right\}}$, $L_n^{MNV}(\theta, \theta_0) = n \left( \sup_{a \in \mathcal{A}} |R(a, \theta) - R(a, \theta_0)| \right)^2$ satisfies Assumption 2.1 with $C_0 = 1$, $C_1 = 4K^2 C$ and $C = 1/8 \left( \frac{\tilde{C}}{d\overline{\Lambda}} - 1 \right)$ for sufficiently large $\tilde{C}$ such that $C > 1$ and $\overline{\Lambda} = \max_{i \in \{1,2,...d\}} \Lambda_{ii}$, where $K = \sup_{\mathcal{A},\Theta} \|\partial_\theta R(a, \theta)\|$.*

In the following result, we show that there exist appropriate constants such that prior distribution satisfies Assumption 2.3 when the demand distribution is a multivariate Gaussian with unknown mean.

**Lemma B.7.** *Fix $n_2 \geq 2$ and any $\lambda > 1$. Let $A_n := \left\{ \theta \in \Theta : D_{1+\lambda} \left( P_0^n \| P_\theta^n \right) \leq C_3 n\epsilon_n^2 \right\}$, where $D_{1+\lambda} \left( P_0^n \| P_\theta^n \right)$ is the Rényi Divergence between $P_0^n$ and $P_\theta^n$. Then for $\epsilon_n^2 = \frac{\log n}{n}$ and any $C_3 > 0$ such that $C_2 = \frac{4d}{\overline{\Lambda}(\lambda+1)\left( \prod_{i=1}^d m(\Theta_i) \right)^{2/d}} C_3 \geq 2$ and for large enough $n$, the uncorrelated uniform prior restricted to $\Theta$ satisfies $\Pi(A_n) \geq \exp(-nC_2\epsilon_n^2)$.*

Next, it is straightforward to see that the multi-product newsvendor model risk $R(a, \theta)$ is bounded below for a given $a \in \mathcal{A}$ on a compact set $\Theta$ and thus it satisfies Assumption 2.5. Finally, we also show that the newsvendor model risk satisfies Assumption 2.4.

**Lemma B.8.** *Fix $n \geq 1$ and $\gamma > 0$. For any $\epsilon > \epsilon_n$ and $a \in \mathcal{A}$, $R(a, \theta)$ satisfies*
$\mathbb{E}_{\Pi}[\mathbb{1}_{\{G(a,\theta)\gamma > C_4(\gamma)n\epsilon^2\}}e^{\gamma G(a,\theta)}] \leq \exp(-C_5(\gamma)n\epsilon_n^2)$, *for any* $C_4(\gamma) > 2\gamma \sup_{\{a,\theta\}\in\mathcal{A}\otimes\Theta} G(a,\theta)$
*and* $C_5(\gamma) = C_4(\gamma) - 2\gamma \sup_{\{a,\theta\}\in\mathcal{A}\otimes\Theta} G(a,\theta)$.

Similar to single product example, in our next result, we show that in the multi-product newsvendor setting, we can construct a sequence $\{Q_n(\theta)\} \in \mathcal{Q}$ that satisfies Assumption 3.1, and thus identify $\epsilon'_n$ and constant $C_9$. We fix $\mathcal{Q}$ to be the family of uncorrelated Gaussian distributions restricted to $\Theta$.

**Lemma B.9.** *Let $\{Q_n(\theta)\}$ be a sequence of product of $d$ univariate Gaussian distribution defined as*
$q_n^i(\theta) \propto \frac{1}{\sqrt{2\pi\sigma_{i,n}^2}}e^{-\frac{1}{2\sigma_{i,n}^2}(\theta-\mu_{i,n})^2}\mathbb{1}_{\Theta_i} = \frac{\mathcal{N}(\theta_i|\mu_{i,n},\sigma_{i,n})\mathbb{1}_{\Theta_i}}{\mathcal{N}(\Theta_i|\mu_{i,n},\sigma_{i,n})}$ *and fix* $\sigma_{i,n} = 1/\sqrt{n}$ *and* $\theta_i = \theta_0^i$ *for all*
$i \in \{1, 2, \ldots, d\}$. *Then for uncorrelated uniform distribution restricted to $\Theta$ and multivariate normal likelihood model* $\frac{1}{n}\left[\text{KL}\left(Q_n(\theta)\|\Pi(\theta)\right) + \mathbb{E}_{Q_n(\theta)}\left[\text{KL}\left(dP_0^n(\tilde{X}_n))\|dP_\theta^n(\tilde{X}_n)\right)\right]\right] \leq C_9\epsilon_n'^2$, *where*
$\epsilon_n'^2 = \frac{\log n}{n}$ *and* $C_9 := \frac{d}{2} + \max\left(0, -\sum_{i=1}^d[\log(\sqrt{2\pi e}) - \log(m(\Theta_i))] + \frac{d}{2}\underline{\Lambda}^{-1}\right)$.

Now, using the result established in lemmas above, we bound the optimality gap in values for the multi-product newsvendor model risk.

**Theorem B.1.** *Fix $\gamma > 0$. Suppose that the set $\mathcal{A}$ is compact. Then, for the multi-product newsvendor model with multivariate Gaussian distributed demand with known covariance matrix $\Sigma$ and unknown mean vector $\theta$ lying in a compact subset $\Theta \subset \mathbb{R}^d$, prior $\Pi(\cdot) = \prod_{i=1}^d \frac{m(\{\cdot\}\cap\Theta_i)}{m(\Theta_i)}$, and the variational family fixed to uncorrelated Gaussian distribution restricted to $\Theta$, and for any $\tau > 0$, the $P_0^n-$ probability of the following event $\left\{\tilde{X}_n : R(\mathtt{a}_{\mathtt{RS}}^*, \theta_0) - \inf_{z\in\mathcal{A}} R(z, \theta_0) \leq 2\tau M'(\gamma)\left(\frac{\log n}{n}\right)^{1/2}\right\}$ is at least $1 - \tau^{-1}$ for sufficiently large $n$ and for some mapping $M' : \mathbb{R}^+ \to \mathbb{R}^+$, where $R(\cdot, \theta)$ is the multi-product newsvendor model risk.*

*Proof.* The proof is a direct consequence of Theorem 3.2, Lemmas B.6, B.7, B.8, B.9, and Proposition 3.2. $\qquad\square$

## B.3 Gaussian process classification (cont.)

We define the distance function as $L_n^{GP}(\theta, \theta_0) = n\left(\sup_{a\in\mathcal{A}} |R(a, \theta) - R(a, \theta_0)|\right)^2$. In anticipation of demonstrating that the binary classification model with GP prior and distance function $L_n^{GP}$ satisfy the desired set of assumptions, we recall the following result, from [30], which will be central in establishing Assumptions 2.1, 2.2, and 2.3.

**Lemma B.10.** *[Theorem 2.1 [30]] Let $\theta(\cdot)$ be a Borel measurable, zero-mean Gaussian random element in a separable Banach space $(\Theta, \|\cdot\|)$ with reproducing kernel Hilbert space (RKHS) $(\mathbb{H}, \|\cdot\|_{\mathbb{H}})$ and let $\theta_0$ be contained in the closure of $\mathbb{H}$ in $\Theta$. For any $\epsilon > \epsilon_n$ satisfying $\varphi_{\theta_0}(\epsilon) \leq n\epsilon^2$, where*

$$\varphi_{\theta_0}(\epsilon) = \inf_{h\in\mathbb{H}:\|h-\theta_0\|<\epsilon} \|h\|_{\mathbb{H}}^2 - \log \Pi(\|\theta\| < \epsilon) \tag{12}$$

*and any $C_{10} > 1$ with $e^{-C_{10}n\epsilon_n^2} < 1/2$, there exists a measurable set $\Theta_n(\epsilon) \subset \Theta$ such that*

$$\log N(3\epsilon, \Theta_n(\epsilon), \|\cdot\|) \leq 6C_{10}n\epsilon^2, \tag{13}$$

$$\Pi(\theta \notin \Theta_n(\epsilon)) \leq e^{-C_{10}n\epsilon^2}, \tag{14}$$

$$\Pi(\|\theta - \theta_0\| < 4\epsilon_n) \geq e^{-n\epsilon_n^2}. \tag{15}$$

The proof of the lemma above can be easily adapted from the proof of [30, Theorem 2.1], which is specifically for $\epsilon = \epsilon_n$. Notice that the result above is true for any norm $\|\cdot\|$ on the Banach space if that satisfies $\varphi_{\theta_0}(\epsilon) \leq n\epsilon^2$. Moreover, if $\varphi_{\theta_0}(\epsilon_n) \leq n\epsilon_n^2$ is true, then it also holds for any $\epsilon > \epsilon_n$, since by definition $\varphi_{\theta_0}(\epsilon)$ is a decreasing function of $\epsilon$.

All the results in the previous lemma depend on $\varphi_{\theta_0}(\epsilon)$ being less than $n\epsilon^2$. In particular, observe that the second term in the definition of $\varphi_{\theta_0}(\epsilon)$ depends on the prior distribution on $\Theta$. Therefore,

[30, Theorem 4.5] show that $\varphi_{\theta_0}(\epsilon_n) \leq n\epsilon_n^2$ ( with $\|\cdot\|$ as supremum norm and for $\epsilon_n$ as defined later in (9) ) is satisfied by the Gaussian prior of type

$$W(\cdot) = \sum_{j=1}^{\bar{J}_\alpha} \sum_{k=1}^{2^{jd}} \mu_j Z_{j,k} \vartheta_{j,k}(\cdot), \tag{16}$$

where $\{\mu_j\}$ is a sequence that decreases with $j$, $\{Z_{i,j}\}$ are i.i.d. standard Gaussian random variables and $\{\vartheta_{j,k}\}$ form a double-indexed orthonormal basis (with respect to measure $\nu$), that is $\mathbb{E}_\nu[\vartheta_{j,k}\vartheta_{l,m}] = \mathbb{1}_{\{j=l,k=m\}}$). $\bar{J}_\alpha$ is the smallest integer satisfying $2^{\bar{J}_\alpha d} = n^{d/(2\alpha+d)}$ for a given $\alpha > 0$. In particular, the GP above is constructed using the function class that is supported on $[0,1]^d$ and has a wavelet expansion, $w(\cdot) = \sum_{j=1}^\infty \sum_{k=1}^{2^{jd}} w_{j,k}\vartheta_{j,k}(\cdot)$. The wavelet function space is equipped with the $L_2-$norm: $\|w\|_2 = \sum_{j=1}^\infty \left( \sum_{k=1}^{2^{jd}} |w_{j,k}|^2 \right)^{1/2}$; the supremum norm: $\|w\|_\infty = \sum_{j=1}^\infty 2^{jd} \max_{1\leq k\leq 2^{jd}} |w_{j,k}|$; and the Besov $(\beta,\infty,\infty)-$norm: $\|w\|_{\beta;\infty,\infty} = \sup_{1\leq j<\infty} 2^{j\beta} 2^{jd} \max_{1\leq k\leq 2^{jd}} |w_{j,k}|$. Note that $W$ induces a measure over the RKHS $\mathbb{H}$, defined as a collection of truncated wavelet functions $w(\cdot) = \sum_{j=1}^{\bar{J}_\alpha} \sum_{k=1}^{2^{jd}} w_{j,k}\vartheta_{j,k}(\cdot)$, with norm induced by the inner-product on $\mathbb{H}$ as $\|w\|_{\mathbb{H}}^2 = \sum_{j=1}^{\bar{J}_\alpha} \sum_{k=1}^{2^{jd}} \frac{w_{j,k}^2}{\mu_j^2}$. The RKHS kernel $K : [0,1]^d \times [0,1]^d \mapsto \mathbb{R}$ can be easily derived as

$$K(x,y) = \mathbb{E}[W(x)W(y)] = \mathbb{E}\left[ \left( \sum_{j=1}^{\bar{J}_\alpha} \sum_{k=1}^{2^{jd}} \mu_j Z_{j,k} \vartheta_{j,k}(y) \right) \left( \sum_{j=1}^{\bar{J}_\alpha} \sum_{k=1}^{2^{jd}} \mu_j Z_{j,k} \vartheta_{j,k}(x) \right) \right]$$

$$= \sum_{j=1}^{\bar{J}_\alpha} \sum_{k=1}^{2^{jd}} \mu_j^2 \vartheta_{j,k}(y)\vartheta_{j,k}(x).$$

Indeed, by the definition of this kernel and inner product, observe that $\langle K(x,\cdot), w(\cdot) \rangle = \sum_{j=1}^{\bar{J}_\alpha} \sum_{k=1}^{2^{jd}} w_{j,k} \mu_j^2 \vartheta_{j,k}(x) \frac{1}{\mu_j^2} = w(x)$. Moreover, $\langle K(x,\cdot), K(y,\cdot) \rangle = \sum_{j=1}^{\bar{J}_\alpha} \sum_{k=1}^{2^{jd}} \mu_j^2 \vartheta_{j,k}(x) \mu_j^2 \vartheta_{j,k}(y) \frac{1}{\mu_j^2} = K(x,y)$. It is clear from its definition that $W$ is a centered Gaussian random field on the RKHS.

Next, using the definition of the kernel, we derive the covariance operator of the Gaussian random field $W$. Recall that $Y \sim \nu$, which enables us to define the covariance operator $\mathcal{C}$, following [27, (6.19)] as $(\mathcal{C}h_\nu)(x) = \int_{[0,1]^d} K(x,y)h_\nu(y)d\nu(y)$. Also, observe that $\{\mu_j^2, \varphi_{j,k}\}$ is the eigenvalue and eigen function pair of the covariance operator $\mathcal{C}$. Consequently, using Karhunen Loéve expansion [27, Theorem 6.19] the prior induced by $W$ on $\mathbb{H}$ is a Gaussian distribution denoted as $\mathcal{N}(0, \mathcal{C})$. We also recall the Cameron-Martin space denoted as $\text{Im}(\mathcal{C}^{1/2})$ associated with a Gaussian measure $\mathcal{N}(0, \mathcal{C})$ on $\mathbb{H}$ to be the intersection of all linear spaces of full measure under $\mathcal{N}(0, \mathcal{C})$ [27, (page 530)]. In particular, $\text{Im}(\mathcal{C}^{1/2})$ is the Hilbert space with inner product $\langle \cdot, \cdot \rangle_\mathcal{C} = \langle \mathcal{C}^{-1/2}\cdot, \mathcal{C}^{-1/2}\cdot \rangle$.

Next, we show the existence of test functions in the following result.

**Lemma B.11.** *For any $\epsilon > \epsilon_n$ with $\epsilon_n \to 0$, $n\epsilon_n^2 \geq 2\log 2$, and $\varphi_{\theta_0}(\epsilon) \leq n\epsilon^2$, there exists a test function $\phi_n$ (depending on $\epsilon$) such that $L_n^{GP}(\theta, \theta_0) = n \left( \sup_{a\in\mathcal{A}} |R(a,\theta) - R(a,\theta_0)| \right)^2$ satisfies Assumption 2.1 with $C = 1/6$, $C_0 = 2$ and $C_1 = (\max(c_+, c_-))^2$.*

Assumption 2.2 is a direct consequence of (14) in Lemma B.10. Next, we prove that prior distribution and the likelihood model satisfy Assumption 2.3 using (15) of Lemma B.10.

**Lemma B.12.** *For any $\lambda > 1$, let $A_n := \{\theta \in \Theta : D_{1+\lambda}(P_0^n \| P_\theta^n) \leq C_3 n\epsilon_n^2\}$, where $D_{1+\lambda}(P_0^n \| P_\theta^n)$ is the Rényi Divergence between $P_0^n$ and $P_\theta^n$. Then for any $\epsilon > \epsilon_n$ satisfying $\varphi_{\theta_0}(\epsilon) \leq n\epsilon^2$ and $C_3 = 16(\lambda + 1)$ and $C_2 = 1$, the GP prior satisfies $\Pi(A_n) \geq \exp(-nC_2\epsilon_n^2)$.*

Assumption 2.4 and 2.5 are straightforward to satisfy since the model risk function $R(a, \theta)$ is bounded from above and below.

Now, suppose the variational family $\mathcal{Q}_{GP}$ is a class of Gaussian distributions on $\Theta$, defined as $\mathcal{N}(m_q, \mathcal{C}_q)$, $m_q$ belongs to $\Theta$ and $\mathcal{C}_q$ is the covariance operator defined as $\mathcal{C}_q = \mathcal{C}^{1/2}(I - S)\mathcal{C}^{1/2}$,

for any $S$ which is a symmetric and Hilbert-Schmidt (HS) operator on $\Theta$ (eigenvalues of HS operator are square summable). Note that $S$ and $m_q$ span the distributions in $\mathcal{Q}_{GP}$.

The following lemma verifies Assumption 3.1, for a specific sequence of distributions in $\mathcal{Q}$.

**Lemma B.13.** *For a given $J \in \mathbb{N}$, let $\{Q_n\}$ be a sequence variational distribution such that $Q_n$ is the measure induced by a GP, $W_Q(\cdot) = \theta_0^J(y) + \sum_{j=1}^J \sum_{k=1}^{2^{jd}} \zeta_j^2 Z_{j,k} \vartheta_{j,k}(\cdot)$, where $\theta_0^J(\cdot) = \sum_{j=1}^J \sum_{k=1}^{2^{jd}} \theta_{0;j,k} \vartheta_{j,k}(\cdot)$ and $\zeta_j^2 = \frac{\mu_j^2}{1+n\epsilon_n^2 \tau_j^2}$. Then for GP prior induced by $W = \sum_{j=1}^J \sum_{k=1}^{2^{jd}} \mu_j Z_{j,k} \vartheta_{j,k}$ and $\mu_j = 2^{-jd/2-ja}$ for some $a > 0$, $\|\theta_0\|_{\beta;\infty,\infty} < \infty$, and $\theta_0^J(y)$ lie in the Cameron-Martin space $Im(\mathcal{C}^{1/2})$, we have $\frac{1}{n}\mathrm{KL}(\mathcal{N}(\bar{\theta}_0^J, \mathcal{C}_q)\|\mathcal{N}(0, \mathcal{C})) + \frac{1}{n}\mathbb{E}_{Q_n}\mathrm{KL}(P_0^n\|P_\theta^n) \leq C_9\epsilon_n^2$, where $\epsilon_n$ is defined in 9 and $C_9 := \max\left(\|\theta_0\|_{\beta,\infty,\infty}^2, \frac{2^{-2a}-2^{-2Ja-2a}}{1-2^{-2a}}, 2^d/(2^d-1), C'\right)$, where $C'$ is a positive constant satisfying $\|\theta_0(y) - \theta_0^J(y))\|_\infty^2 \leq C' 2^{-2J\beta}$.*

Using the result above together with Proposition 3.2 implies that the RSVB posterior converges at the same rate as the true posterior, where the convergence rate of the true posterior is derived in [30, Theorem 4.5] for the binary GP classification problem with truncated wavelet GP prior. Finally, we use the results above to obtain bound on the optimality gap in values of the binary GP classification problem.

# C Proofs

## C.1 Alternative derivation of LCVB

We present the alternative derivation of LCVB. Consider the logarithm of the Bayes posterior risk,

$$
\begin{aligned}
\log \mathbb{E}_{\Pi(\theta|\tilde{X}_n)}[\exp(R(a,\theta))] &= \log \int_\Theta \exp(R(a,\theta)) d\Pi(\theta|\tilde{X}_n) \\
&= \log \int_\Theta \frac{dQ(\theta)}{dQ(\theta)} \exp(R(a,\theta)) d\Pi(\theta|\tilde{X}_n) \\
&\geq -\int_\Theta dQ(\theta) \log \frac{dQ(\theta)}{\exp(R(a,\theta)) d\Pi(\theta|\tilde{X}_n)} =: \mathcal{F}(a; Q(\cdot), \tilde{X}_n) \quad (17)
\end{aligned}
$$

where the inequality follows from an application of Jensen's inequality (since, without loss of generality, $\exp(R(a,\theta)) > 0$ for all $a \in \mathcal{A}$ and $\theta \in \Theta$), and $Q \in \mathcal{Q}$. Then, it follows that

$$
\begin{aligned}
\min_{a\in\mathcal{A}} \log \mathbb{E}_{\Pi(\theta|\tilde{X}_n)}[\exp(R(a,\theta))] &\geq \min_{a\in\mathcal{A}} \max_{q\in\mathcal{Q}} \mathcal{F}(a; Q(\theta), \tilde{X}_n) \\
&= \min_{a\in\mathcal{A}} \max_{q\in\mathcal{Q}} - \mathrm{KL}\left(Q(\theta)\|\Pi(\theta|\tilde{X}_n)\right) + \int_\Theta R(a,\theta) dQ(\theta). \quad (18)
\end{aligned}
$$

## C.2 Proof of Theorem 3.1

We prove our main result after a series of important lemmas. For brevity we denote $\mathcal{LR}_n(\theta,\theta_0) = \frac{p(\tilde{X}_n|\theta)}{p(\tilde{X}_n|\theta_0)}$.

**Lemma C.1.** *For any $a' \in \mathcal{A}$, $\gamma > 0$, and $\zeta > 0$,*

$$
\begin{aligned}
&\mathbb{E}_{P_0^n}\left[\zeta \int_\Theta L_n(\theta,\theta_0) \, dQ_{a',\gamma}^*(\theta|\tilde{X}_n)\right] \\
&\leq \log \mathbb{E}_{P_0^n}\left[\int_\Theta e^{\zeta L_n(\theta,\theta_0)} \frac{e^{\gamma R(a',\theta)} \mathcal{LR}_n(\theta,\theta_0) d\Pi(\theta)}{\int_\Theta e^{\gamma R(a',\theta)} \mathcal{LR}_n(\theta,\theta_0) d\Pi(\theta)}\right] + \inf_{Q\in\mathcal{Q}} \mathbb{E}_{P_0^n}\left[\mathrm{KL}(Q(\theta)\|\Pi(\theta|\tilde{X}_n))\right] \\
&\quad - \gamma \inf_{a\in\mathcal{A}} \mathbb{E}_Q[R(a,\theta)]\right] + \log \mathbb{E}_{P_0^n}\left[\int_\Theta e^{\gamma R(a',\theta)} \frac{\mathcal{LR}_n(\theta,\theta_0) d\Pi(\theta)}{\int_\Theta \mathcal{LR}_n(\theta,\theta_0) d\Pi(\theta)}\right]. \quad (19)
\end{aligned}
$$

*Proof.* For any fixed $a' \in \mathcal{A}, \gamma > 0$, and $\zeta > 0$, and using the fact that KL is non-negative, observe that the integral in the LHS of equation (19) satisfies,

$$
\zeta \mathbb{E}_{Q^*_{a',\gamma}(\theta|\tilde{X}_n)}\left[L_n(\theta,\theta_0)\right] \leq \mathbb{E}_{Q^*_{a',\gamma}(\theta|\tilde{X}_n)}\left[\log e^{\zeta L_n(\theta,\theta_0)}\right]
$$
$$
+ \text{KL}\left(dQ^*_{a',\gamma}(\theta|\tilde{X}_n)\left\|\frac{e^{\zeta L_n(\theta,\theta_0)}e^{\gamma R(a',\theta)}\,d\Pi(\theta|\tilde{X}_n)}{\int_\Theta e^{\zeta L_n(\theta,\theta_0)}e^{\gamma R(a',\theta)}\,d\Pi(\theta|\tilde{X}_n)}\right.\right)
$$
$$
= \mathbb{E}_{Q^*_{a',\gamma}(\theta|\tilde{X}_n)}\left[\log e^{\zeta L_n(\theta,\theta_0)}\right] + \log \mathbb{E}_{\Pi_n}\left[e^{\zeta L_n(\theta,\theta_0)}e^{\gamma R(a',\theta)}\right]
$$
$$
+ \mathbb{E}_{Q^*_{a',\gamma}(\theta|\tilde{X}_n)}\left[\log \frac{dQ^*_{a',\gamma}(\theta|\tilde{X}_n)}{e^{\zeta L_n(\theta,\theta_0)}e^{\gamma R(a',\theta)}\,d\Pi(\theta|\tilde{X}_n)}\right]
$$
$$
= \log \mathbb{E}_{\Pi_n}\left[e^{\zeta L_n(\theta,\theta_0)}e^{\gamma R(a',\theta)}\right] + \mathbb{E}_{Q^*_{a',\gamma}(\theta|\tilde{X}_n)}\left[\log \frac{dQ^*_{a',\gamma}(\theta|\tilde{X}_n)}{e^{\gamma R(a',\theta)}\,d\Pi(\theta|\tilde{X}_n)}\right].
$$

Next, using the definition of $Q^*_{a',\gamma}(\theta|\tilde{X}_n)$ in the second term of last equality, for any other $Q(\cdot) \in \mathcal{Q}$

$$
\zeta \mathbb{E}_{Q^*_{a',\gamma}(\theta|\tilde{X}_n)}\left[L_n(\theta,\theta_0)\right] \leq \log \mathbb{E}_{\Pi_n}\left[e^{\zeta L_n(\theta,\theta_0)}e^{\gamma R(a',\theta)}\right] + \mathbb{E}_Q\left[\log \frac{dQ(\theta)}{e^{\gamma R(a',\theta)}\,d\Pi(\theta|\tilde{X}_n)}\right].
$$

Finally, it follows from the definition of the posterior distribution that

$$
\zeta \mathbb{E}_{Q^*_{a',\gamma}(\theta|\tilde{X}_n)}\left[L_n(\theta,\theta_0)\right]
$$
$$
\leq \log \int_\Theta e^{\zeta L_n(\theta,\theta_0)}e^{\gamma R(a',\theta)}\frac{\mathcal{L}\mathcal{R}_n(\theta,\theta_0)d\Pi(\theta)}{\int_\Theta \mathcal{L}\mathcal{R}_n(\theta,\theta_0)d\Pi(\theta)} + \mathbb{E}_Q\left[\log \frac{dQ(\theta)}{e^{\gamma R(a',\theta)}\,d\Pi(\theta|\tilde{X}_n)}\right],
$$
$$
= \log \int_\Theta e^{\zeta L_n(\theta,\theta_0)}\frac{e^{\gamma R(a',\theta)}\mathcal{L}\mathcal{R}_n(\theta,\theta_0)d\Pi(\theta)}{\int_\Theta e^{\gamma R(a',\theta)}\mathcal{L}\mathcal{R}_n(\theta,\theta_0)d\Pi(\theta)} + \mathbb{E}_Q\left[\log \frac{dQ(\theta)}{e^{\gamma R(a',\theta)}\,d\Pi(\theta|\tilde{X}_n)}\right]
$$
$$
+ \log \int_\Theta e^{\gamma R(a',\theta)}\frac{\mathcal{L}\mathcal{R}_n(\theta,\theta_0)d\Pi(\theta)}{\int_\Theta \mathcal{L}\mathcal{R}_n(\theta,\theta_0)d\Pi(\theta)}, \tag{20}
$$

where the last equality follows from adding and subtracting $\log \mathbb{E}_\Pi\left[e^{\gamma R(a',\theta)}\mathcal{L}\mathcal{R}_n(\theta,\theta_0)\right]$. Now taking expectation on either side of equation (20) and using Jensen's inequality on the first and the last term in the RHS yields

$$
\mathbb{E}_{P_0^n}\left[\zeta \mathbb{E}_{Q^*_{a',\gamma}(\theta|\tilde{X}_n)}\left[L_n(\theta,\theta_0)\right]\right]
$$
$$
\leq \log \mathbb{E}_{P_0^n}\left[\int_\Theta e^{\zeta L_n(\theta,\theta_0)}\frac{e^{\gamma R(a',\theta)}\,\mathcal{L}\mathcal{R}_n(\theta,\theta_0)d\Pi(\theta)}{\int_\Theta e^{\gamma R(a',\theta)}\,\mathcal{L}\mathcal{R}_n(\theta,\theta_0)d\Pi(\theta)}\right] + \inf_{Q \in \mathcal{Q}}\mathbb{E}_{P_0^n}\left[\text{KL}(Q\|\Pi_n)\right]
$$
$$
- \gamma \inf_{a \in \mathcal{A}}\mathbb{E}_Q\left[R(a,\theta)\right] + \log \mathbb{E}_{P_0^n}\left[\int_\Theta e^{\gamma R(a',\theta)}\frac{\mathcal{L}\mathcal{R}_n(\theta,\theta_0)d\Pi(\theta)}{\int_\Theta \mathcal{L}\mathcal{R}_n(\theta,\theta_0)d\Pi(\theta)}\right], \tag{21}
$$

where in the second term in RHS of (20), we first take infimum over all $a \in \mathcal{A}$ which upper bounds the second term in (20) and then take infimum over all $Q \in \mathcal{Q}$, since the LHS does not depend on $Q$. $\qquad\square$

Next, we state a technical result that is important in proving our next lemma.

**Lemma C.2** (Lemma 6.4 of [34]). *Suppose random variable X satisfies*

$$
\mathbb{P}(X \geq t) \leq c_1 \exp(-c_2 t),
$$

*for all $t \geq t_0 > 0$. Then for any $0 < \beta \leq c_2/2$,*

$$
\mathbb{E}[\exp(\beta X)] \leq \exp(\beta t_0) + c_1.
$$

*Proof.* Refer Lemma 6.4 of [34].

$\qquad\square$

In the following result, we bound the first term on the RHS of equation (19). The arguments in the proof are essentially similar to Lemma 6.3 in [34]

**Lemma C.3.** *Under Assumptions 2.1, 2.2, 2.3, 2.4, and 2.5 and for* $\min(C, C_4(\gamma) + C_5(\gamma)) > C_2 + C_3 + C_4(\gamma) + 2$ *and any* $\epsilon \geq \epsilon_n$,

$$\mathbb{E}_{P_0^n} \left[ \int_\Theta e^{\zeta L_n(\theta, \theta_0)} \frac{e^{\gamma R(a', \theta)} \mathcal{L} \mathcal{R}_n(\theta, \theta_0) d\Pi(\theta)}{\int_\Theta e^{\gamma R(a', \theta)} \mathcal{L} \mathcal{R}_n(\theta, \theta_0) d\Pi(\theta)} \right] \leq e^{\zeta C_1 n \epsilon^2} + (1 + C_0 + 3W^{-\gamma}), \quad (22)$$

*for* $0 < \zeta \leq C_{10}/2$, *where* $C_{10} = \min\{\lambda, C, 1\}/C_1$ *for any* $\lambda > 0$.

*Proof.* First define the set

$$B_n := \left\{ \tilde{X}_n : \int_\Theta \mathcal{L} \mathcal{R}_n(\theta, \theta_0) d\Pi(\theta) \geq e^{-(1+C_3)n\epsilon^2} \Pi(A_n) \right\}, \quad (23)$$

where set $A_n$ is defined in Assumption 2.3. We demonstrate that, under Assumption 2.3, $P_0^n(B_n^c)$ is bounded above by an exponentially decreasing(in $n$) term. Note that for $A_n$ as defined in Assumption 2.3:

$$\mathbb{P}_0^n \left( \frac{1}{\Pi(A_n)} \int_\Theta \mathcal{L} \mathcal{R}_n(\theta, \theta_0) d\Pi(\theta) \leq e^{-(1+C_3)n\epsilon^2} \right)$$

$$\leq \mathbb{P}_0^n \left( \frac{1}{\Pi(A_n)} \int_{\Theta \cap A_n} \mathcal{L} \mathcal{R}_n(\theta, \theta_0) d\Pi(\theta) \leq e^{-(1+C_3)n\epsilon^2} \right). \quad (24)$$

Let $d\tilde{\Pi}(\theta) := \frac{\mathbb{1}_{\{\Theta \cap A_n\}}(\theta)}{\Pi(A_n)} d\Pi(\theta)$, and use this in (24) for any $\lambda > 0$ to obtain,

$$\mathbb{P}_0^n \left( \int_\Theta \mathcal{L} \mathcal{R}_n(\theta, \theta_0) d\tilde{\Pi}(\theta) \leq e^{-(1+C_3)n\epsilon^2} \right) = \mathbb{P}_0^n \left( \left[ \int_\Theta \mathcal{L} \mathcal{R}_n(\theta, \theta_0) d\tilde{\Pi}(\theta) \right]^{-\lambda} \geq e^{(1+C_3)\lambda n\epsilon^2} \right).$$

Then, using the Markov's inequality in the last equality above, we have

$$\mathbb{P}_0^n \left( \int_\Theta \mathcal{L} \mathcal{R}_n(\theta, \theta_0) d\tilde{\Pi}(\theta) \leq e^{-(1+C_3)n\epsilon^2} \right) \leq e^{-(1+C_3)\lambda n\epsilon^2} \mathbb{E}_{P_0^n} \left( \left[ \int_\Theta \mathcal{L} \mathcal{R}_n(\theta, \theta_0) d\tilde{\Pi}(\theta) \right]^{-\lambda} \right)$$

$$\leq e^{-(1+C_3)\lambda n\epsilon^2} \left[ \int_\Theta \mathbb{E}_{P_0^n} \left( [\mathcal{L} \mathcal{R}_n(\theta, \theta_0)]^{-\lambda} \right) d\tilde{\Pi}(\theta) \right]$$

$$= e^{-(1+C_3)\lambda n\epsilon^2} \left[ \int_\Theta \exp(\lambda D_{\lambda+1} (P_0^n \| P_\theta^n)) d\tilde{\Pi}(\theta) \right]$$

$$\leq e^{-(1+C_3)\lambda n\epsilon^2} e^{\lambda C_3 n \epsilon_n^2} \leq \epsilon^{-\lambda n\epsilon^2}, \quad (25)$$

where the second inequality follows from first applying Jensen's inequality (on the term inside $[\cdot]$) and then using Fubini's theorem, and the penultimate inequality follows from Assumption 2.3 and the definition of $\tilde{\Pi}(\theta)$.

Next, define the set $K_n := \{\theta \in \Theta : L_n(\theta, \theta_0) > C_1 n \epsilon^2\}$. Notice that set $K_n$ is the set of alternate hypothesis as defined in Assumption 2.1. We bound the calibrated posterior probability of this set $K_n$ to get a bound on the first term in the RHS of equation (19). Recall the sequence of test function $\{\phi_{n,\epsilon}\}$ from Assumption 2.1. Observe that

$$\mathbb{E}_{P_0^n} \left[ \frac{\int_{K_n} e^{\gamma R(a', \theta)} \mathcal{L} \mathcal{R}_n(\theta, \theta_0) d\Pi(\theta)}{\int_\Theta e^{\gamma R(a', \theta)} \mathcal{L} \mathcal{R}_n(\theta, \theta_0) d\Pi(\theta)} \right]$$

$$= \mathbb{E}_{P_0^n} \left[ (\phi_{n,\epsilon} + 1 - \phi_{n,\epsilon}) \frac{\int_{K_n} e^{\gamma R(a', \theta)} \mathcal{L} \mathcal{R}_n(\theta, \theta_0) d\Pi(\theta)}{\int_\Theta e^{\gamma R(a', \theta)} \mathcal{L} \mathcal{R}_n(\theta, \theta_0) d\Pi(\theta)} \right]$$

$$\leq \mathbb{E}_{P_0^n}[\phi_{n,\epsilon}] + \mathbb{E}_{P_0^n} \left[ (1 - \phi_{n,\epsilon}) \mathbb{1}_{B_n^C} \right]$$

$$+ \mathbb{E}_{P_0^n} \left[ (1 - \phi_{n,\epsilon}) \mathbb{1}_{B_n} \frac{\int_{K_n} e^{\gamma R(a', \theta)} \mathcal{L} \mathcal{R}_n(\theta, \theta_0) d\Pi(\theta)}{\int_\Theta e^{\gamma R(a', \theta)} \mathcal{L} \mathcal{R}_n(\theta, \theta_0) d\Pi(\theta)} \right]$$

$$\leq \mathbb{E}_{P_0^n} \phi_{n,\epsilon} + \mathbb{E}_{P_0^n} \left[ \mathbb{1}_{B_n^C} \right] + \mathbb{E}_{P_0^n} \left[ (1 - \phi_{n,\epsilon}) \mathbb{1}_{B_n} \frac{\int_{K_n} e^{\gamma R(a', \theta)} \mathcal{L} \mathcal{R}_n(\theta, \theta_0) d\Pi(\theta)}{\int_\Theta e^{\gamma R(a', \theta)} \mathcal{L} \mathcal{R}_n(\theta, \theta_0) d\Pi(\theta)} \right], \quad (26)$$

where in the second inequality, we first divide the second term over set $B_n$ and its complement and then use the fact that $\frac{\int_{K_n} e^{\gamma R(a',\theta)} \mathcal{LR}_n(\theta,\theta_0)d\Pi(\theta)}{\int_{\Theta} e^{\gamma R(a',\theta)} \mathcal{LR}_n(\theta,\theta_0)d\Pi(\theta)} \leq 1$. The third inequality is due the fact that $\phi_{n,\epsilon} \in [0,1]$. Next, using Assumption 2.3 and 2.5 observe that on set $B_n$

$$\int_{\Theta} e^{\gamma R(a',\theta)} \mathcal{LR}_n(\theta,\theta_0)d\Pi(\theta) \geq W^{\gamma} \int_{\Theta} \mathcal{LR}_n(\theta,\theta_0)d\Pi(\theta)$$
$$\geq W^{\gamma} e^{-(1+C_2+C_3)n\epsilon_n^2} \geq W^{\gamma} e^{-(1+C_2+C_3)n\epsilon^2}.$$

Substituting the equation above in the third term of equation (26), we obtain

$$\mathbb{E}_{P_0^n}\left[(1-\phi_{n,\epsilon})\mathbb{1}_{B_n} \frac{\int_{K_n} e^{\gamma R(a',\theta)} \mathcal{LR}_n(\theta,\theta_0)d\Pi(\theta)}{\int_{\Theta} e^{\gamma R(a',\theta)} \mathcal{LR}_n(\theta,\theta_0)d\Pi(\theta)}\right]$$
$$\leq W^{-\gamma} e^{(1+C_2+C_3)n\epsilon^2} \mathbb{E}_{P_0^n}\left[(1-\phi_{n,\epsilon})\mathbb{1}_{B_n} \int_{K_n} e^{\gamma R(a',\theta)} \mathcal{LR}_n(\theta,\theta_0)d\Pi(\theta)\right]$$
$$\leq W^{-\gamma} e^{(1+C_2+C_3)n\epsilon^2} \mathbb{E}_{P_0^n}\left[(1-\phi_{n,\epsilon}) \int_{K_n} e^{\gamma R(a',\theta)} \mathcal{LR}_n(\theta,\theta_0)d\Pi(\theta)\right]. \qquad (\star)$$

Now using Fubini's theorem observe that,

$$(\star) = W^{-\gamma} e^{(1+C_2+C_3)n\epsilon^2} \int_{K_n} e^{\gamma R(a',\theta)} \mathbb{E}_{P_{\theta}^n}\left[(1-\phi_{n,\epsilon})\right] d\Pi(\theta)$$
$$\leq W^{-\gamma} e^{(1+C_2+C_3+C_4(\gamma))n\epsilon^2}\left[\int_{K_n \cap \{e^{\gamma R(a',\theta)} \leq e^{C_4(\gamma)n\epsilon^2}\}} \mathbb{E}_{P_{\theta}^n}\left[(1-\phi_{n,\epsilon})\right] d\Pi(\theta)\right.$$
$$\left. + e^{-C_4(\gamma)n\epsilon^2} \int_{K_n \cap \{e^{\gamma R(a',\theta)} > e^{C_4(\gamma)n\epsilon^2}\}} e^{\gamma R(a',\theta)} d\Pi(\theta)\right],$$

where in the last inequality, we first divide the integral over set $\{\theta \in \Theta : e^{\gamma R(a',\theta)} \leq e^{C_4(\gamma)n\epsilon^2}\}$ and its complement and then use the upper bound on $e^{\gamma R(a',\theta)}$ in the first integral. Now, it follows that

$$(\star) \leq W^{-\gamma} e^{(1+C_2+C_3+C_4(\gamma))n\epsilon^2}\left[\int_{K_n} \mathbb{E}_{P_{\theta}^n}\left[(1-\phi_{n,\epsilon})\right] d\Pi(\theta)\right.$$
$$\left. + e^{-C_4(\gamma)n\epsilon^2} \int_{\{e^{\gamma R(a',\theta)} > e^{C_4(\gamma)n\epsilon^2}\}} e^{\gamma R(a',\theta)} d\Pi(\theta)\right]$$
$$= W^{-\gamma} e^{(1+C_2+C_3+C_4(\gamma))n\epsilon^2}\left[\int_{K_n \cap \Theta_n(\epsilon)} \mathbb{E}_{P_{\theta}^n}\left[(1-\phi_{n,\epsilon})\right] d\Pi(\theta)\right.$$
$$\left. + \int_{K_n \cap \Theta_n(\epsilon)^c} \mathbb{E}_{P_{\theta}^n}\left[(1-\phi_{n,\epsilon})\right] d\Pi(\theta) + e^{-C_4(\gamma)n\epsilon^2} \int_{\{e^{\gamma R(a',\theta)} > e^{C_4(\gamma)n\epsilon^2}\}} e^{\gamma R(a',\theta)} d\Pi(\theta)\right]$$
$$\leq W^{-\gamma} e^{(1+C_2+C_3+C_4(\gamma))n\epsilon^2}\left[\int_{K_n \cap \Theta_n(\epsilon)} \mathbb{E}_{P_{\theta}^n}\left[(1-\phi_{n,\epsilon})\right] d\Pi(\theta) + \Pi(\Theta_n(\epsilon)^c)\right.$$
$$\left. + e^{-C_4(\gamma)n\epsilon^2} \int_{\{e^{\gamma R(a',\theta)} > e^{C_4(\gamma)n\epsilon^2}\}} e^{\gamma R(a',\theta)} d\Pi(\theta)\right],$$

where the second equality is obtained by dividing the first integral on set $\Theta_n(\epsilon)$ and its complement, and the second inequality is due the fact that $\phi_{n,\epsilon} \in [0,1]$. Now, using the equation above and Assumption 2.1, 2.2, and 2.4 observe that

$$\mathbb{E}_{P_0^n}\left[(1-\phi_{n,\epsilon})\mathbb{1}_{B_n} \frac{\int_{K_n} e^{\gamma R(a',\theta)} \mathcal{LR}_n(\theta,\theta_0)d\Pi(\theta)}{\int_{\Theta} e^{\gamma R(a',\theta)} \mathcal{LR}_n(\theta,\theta_0)d\Pi(\theta)}\right]$$
$$\leq W^{-\gamma} e^{(1+C_2+C_3+C_4(\gamma))n\epsilon^2}\left[2e^{-Cn\epsilon^2} + e^{-(C_5(\gamma)+C_4(\gamma))n\epsilon^2}\right].$$

Hence, choosing $C, C_2, C_3, C_4(\gamma)$ and $C_5(\gamma)$ such that $-1 > 1+C_2+C_3+C_4(\gamma)-\min(C,(C_4(\gamma)+C_5(\gamma)))$ implies

$$\mathbb{E}_{P_0^n}\left[(1-\phi_{n,\epsilon})\mathbb{I}_{B_n}\frac{\int_{K_n}e^{\gamma R(a',\theta)}\,\mathcal{L}\mathcal{R}_n(\theta,\theta_0)d\Pi(\theta)}{\int_\Theta e^{\gamma R(a',\theta)}\,\mathcal{L}\mathcal{R}_n(\theta,\theta_0)d\Pi(\theta)}\right]\le 3W^{-\gamma}e^{-n\epsilon^2}. \tag{27}$$

By Assumption 2.1, we have

$$\mathbb{E}_{P_0^n}\phi_{n,\epsilon}\le C_0e^{-Cn\epsilon^2}. \tag{28}$$

Therefore, substituting equation (25), equation (27), and (28) into (26), we obtain

$$\mathbb{E}_{P_0^n}\left[\frac{\int_{K_n}e^{\gamma R(a',\theta)}\,\mathcal{L}\mathcal{R}_n(\theta,\theta_0)d\Pi(\theta)}{\int_\Theta e^{\gamma R(a',\theta)}\,\mathcal{L}\mathcal{R}_n(\theta,\theta_0)d\Pi(\theta)}\right]\le(1+C_0+3W^{-\gamma})e^{-C_{10}C_1n\epsilon^2}, \tag{29}$$

where $C_{10}=\min\{\lambda,C,1\}/C_1$. Using Fubini's theorem, observe that the LHS in the equation (29) can be expressed as $\mu(K_n)$, where

$$d\mu(\theta)=\mathbb{E}_{P_0^n}\left[\frac{\mathcal{L}\mathcal{R}_n(\theta,\theta_0)}{\int_\Theta e^{\gamma R(a',\theta)}\,\mathcal{L}\mathcal{R}_n(\theta,\theta_0)d\Pi(\theta)}\right]\Pi(\theta)e^{\gamma R(a',\theta)}d\theta.$$

Next, recall that the set $K_n=\{\theta\in\Theta:L_n(\theta,\theta_0)>C_1n\epsilon^2\}$. Applying Lemma C.2 above with $X=L_n(\theta,\theta_0)$, $c_1=(1+C_0+3W^{-\gamma})$, $c_2=C_{10}$, $t_0=C_1n\epsilon_n^2$, and for $0<\zeta\le C_{10}/2$, we obtain

$$\mathbb{E}_{P_0^n}\left[\int_\Theta e^{\zeta L_n(\theta,\theta_0)}\frac{e^{\gamma R(a',\theta)}\,\mathcal{L}\mathcal{R}_n(\theta,\theta_0)\Pi(\theta)}{\int_\Theta e^{\gamma R(a',\theta)}\,\mathcal{L}\mathcal{R}_n(\theta,\theta_0)d\Pi(\theta)}d\theta\right]\le e^{\zeta C_1n\epsilon_n^2}+(1+C_0+3W^{-\gamma}). \tag{30}$$

$\square$

Further, we have another technical lemma, that will be crucial in proving the subsequent lemma that upper bounds the last term in the equation (19).

**Lemma C.4.** *Suppose a positive random variable X satisfies*

$$\mathbb{P}(X\ge e^t)\le c_1\exp(-(c_2+1)t),$$

*for all $t\ge t_0>0$, $c_1>0$, and $c_2>0$. Then,*

$$\mathbb{E}[X]\le\exp(t_0)+\frac{c_1}{c_2}.$$

*Proof.* For any $Z_0>1$,

$$\mathbb{E}[X]\le Z_0+\int_{Z_0}^\infty\mathbb{P}(X\ge x)dx$$

$$=Z_0+\int_{\ln Z_0}^\infty\mathbb{P}(X\ge e^y)e^ydy\le Z_0+c_1\int_{\ln Z_0}^\infty\exp(-c_2y)dy.$$

Therefore, choosing $Z_0=\exp(t_0)$,

$$\mathbb{E}[X]\le\exp(t_0)+\frac{c_1}{c_2}\exp(-c_2t_0)\le\exp(t_0)+\frac{c_1}{c_2}.$$

$\square$

Next, we establish the following bound on the last term in equation (19).

**Lemma C.5.** *Under Assumptions 2.1, 2.2, 2.3, 2.4, 2.5, and for $C_5(\gamma)>C_2+C_3+2$,*

$$\mathbb{E}_{P_0^n}\left[\int_\Theta\frac{e^{\gamma R(a',\theta)}\,\mathcal{L}\mathcal{R}_n(\theta,\theta_0)d\Pi(\theta)}{\int_\Theta\mathcal{L}\mathcal{R}_n(\theta,\theta_0)d\Pi(\theta)}\right]\le e^{C_4(\gamma)n\epsilon_n^2}+2C_4(\gamma). \tag{31}$$

*for any $\lambda\ge1+C_4(\gamma)$.*

*Proof.* Define the set

$$M_n := \{\theta \in \Theta : e^{\gamma R(a',\theta)} > e^{C_4(\gamma)n\epsilon^2}\}. \tag{32}$$

Using the set $B_n$ in equation (23), observe that the measure of the set $M_n$, under the posterior distribution satisfies,

$$\mathbb{E}_{P_0^n}\left[\frac{\int_{M_n} \mathcal{LR}_n(\theta,\theta_0)d\Pi(\theta)}{\int_\Theta \mathcal{LR}_n(\theta,\theta_0)d\Pi(\theta)}\right] \leq \mathbb{E}_{P_0^n}\left[\mathbb{1}_{B_n^c}\right] + \mathbb{E}_{P_0^n}\left[\mathbb{1}_{B_n}\frac{\int_{M_n} \mathcal{LR}_n(\theta,\theta_0)d\Pi(\theta)}{\int_\Theta \mathcal{LR}_n(\theta,\theta_0)d\Pi(\theta)}\right]. \tag{33}$$

Now, the second term of equation (33) can be bounded as follows: recall Assumption 2.3 and the definition of set $B_n$, both together imply that,

$$\mathbb{E}_{P_0^n}\left[\mathbb{1}_{B_n}\frac{\int_{M_n} \mathcal{LR}_n(\theta,\theta_0)d\Pi(\theta)}{\int_\Theta \mathcal{LR}_n(\theta,\theta_0)d\Pi(\theta)}\right] \leq e^{(1+C_2+C_3)n\epsilon^2}\mathbb{E}_{P_0^n}\left[\mathbb{1}_{B_n}\int_{M_n} \mathcal{LR}_n(\theta,\theta_0)d\Pi(\theta)\right]$$

$$\leq e^{(1+C_2+C_3)n\epsilon^2}\mathbb{E}_{P_0^n}\left[\int_{M_n} \mathcal{LR}_n(\theta,\theta_0)d\Pi(\theta)\right]. \quad (\star\star)$$

Then, using Fubini's Theorem $(\star\star) = e^{(1+C_2+C_3)n\epsilon^2}\Pi(M_n)$. Next, using the definition of set $M_n$ and then Assumption 2.4, we obtain

$$\mathbb{E}_{P_0^n}\left[\mathbb{1}_{B_n}\frac{\int_{M_n} \mathcal{LR}_n(\theta,\theta_0)d\Pi(\theta)}{\int_\Theta \mathcal{LR}_n(\theta,\theta_0)d\Pi(\theta)}\right] \leq e^{(1+C_2+C_3)n\epsilon^2}e^{-C_4(\gamma)n\epsilon^2}\int_{M_n} e^{\gamma R(a',\theta)}d\Pi(\theta)$$

$$\leq e^{(1+C_2+C_3)n\epsilon^2}e^{-C_4(\gamma)n\epsilon^2}e^{-C_5(\gamma)n\epsilon^2},$$

Hence, choosing the constants $C_2, C_3, C_4(\gamma)$ and $C_5(\gamma)$ such that $-1 > 1 + C_2 + C_3 - C_5(\gamma)$ implies

$$\mathbb{E}_{P_0^n}\left[\mathbb{1}_{B_n}\frac{\int_{M_n} \mathcal{LR}_n(\theta,\theta_0)d\Pi(\theta)}{\int_\Theta \mathcal{LR}_n(\theta,\theta_0)d\Pi(\theta)}\right] \leq e^{-(1+C_4(\gamma))n\epsilon^2} \tag{34}$$

Therefore, substituting (25) and (34) into (33)

$$\mathbb{E}_{P_0^n}\left[\frac{\int_{M_n} \mathcal{LR}_n(\theta,\theta_0)d\Pi(\theta)}{\int_\Theta \mathcal{LR}_n(\theta,\theta_0)d\Pi(\theta)}\right] \leq 2e^{-C_4(\gamma)(C_{11}(\gamma)+1)n\epsilon^2}, \tag{35}$$

where $C_{11}(\gamma) = \min\{\lambda, 1 + C_4(\gamma)\}/C_4(\gamma) - 1$. Using Fubini's theorem, observe that the RHS in (35) can be expressed as $\nu(M_n)$, where the measure

$$d\nu(\theta) = \mathbb{E}_{P_0^n}\left[\frac{\mathcal{LR}_n(\theta,\theta_0)}{\int_\Theta \mathcal{LR}_n(\theta,\theta_0)d\Pi(\theta)}\right]d\Pi(\theta).$$

Applying Lemma C.4 for $X = e^{\gamma R(a',\theta)}$, $c_1 = 2$, $c_2 = C_{11}(\gamma)$, $t_0 = C_4(\gamma)n\epsilon_n^2$ and $\lambda \geq 1 + C_4(\gamma)$, we obtain

$$\mathbb{E}_{P_0^n}\left[\int_\Theta \frac{e^{\gamma R(a',\theta)} \mathcal{LR}_n(\theta,\theta_0)d\Pi(\theta)}{\int_\Theta \mathcal{LR}_n(\theta,\theta_0)d\Pi(\theta)}\right] \leq e^{C_4(\gamma)n\epsilon_n^2} + \frac{2}{C_{11}(\gamma)} \leq e^{C_4(\gamma)n\epsilon_n^2} + 2C_4(\gamma). \tag{36}$$

$\square$

*Proof.* Proof of Theorem 3.1: Finally, recall (19),

$$\zeta\mathbb{E}_{P_0^n}\left[\int_\Theta L_n(\theta,\theta_0)\,dQ_{a',\gamma}^*(\theta|\tilde{X}_n)\right]$$

$$\leq \log\mathbb{E}_{P_0^n}\left[\int_\Theta e^{\zeta L_n(\theta,\theta_0)}\frac{e^{\gamma R(a',\theta)} \mathcal{LR}_n(\theta,\theta_0)d\Pi(\theta)}{\int_\Theta e^{\gamma R(a',\theta)} \mathcal{LR}_n(\theta,\theta_0)d\Pi(\theta)}\right] + \inf_{Q\in\mathcal{Q}}\mathbb{E}_{P_0^n}\left[\mathrm{KL}(Q\|\Pi_n)\right]$$

$$- \gamma\inf_{a\in\mathcal{A}}\mathbb{E}_Q[R(a,\theta)] + \log\mathbb{E}_{P_0^n}\left[\int_\Theta e^{\gamma R(a',\theta)}\frac{\mathcal{LR}_n(\theta,\theta_0)d\Pi(\theta)}{\int_\Theta \mathcal{LR}_n(\theta,\theta_0)d\Pi(\theta)}\right]. \tag{37}$$

Substituting (31) and (22) into the equation above and then using the definition of $\eta_n^R(\gamma)$, we get

$$
\mathbb{E}_{P_0^n}\left[\int_\Theta L_n(\theta, \theta_0)\, dQ_{a',\gamma}^*(\theta|\tilde{X}_n)\right]
$$

$$
\leq \frac{1}{\zeta}\left\{\log(e^{\zeta C_1 n\epsilon_n^2} + (1 + C_0 + 3W^{-\gamma})) + \log\left(e^{C_4(\gamma)n\epsilon_n^2} + 2C_4(\gamma)\right) + n\eta_n^R(\gamma)\right\}
$$

$$
\leq \left(C_1 + \frac{1}{\zeta}C_4(\gamma)\right)n\epsilon_n^2 + \frac{1}{\zeta}n\eta_n^R(\gamma) + \frac{(1 + C_0 + 3W^{-\gamma})e^{(-\zeta C_1 n\epsilon_n^2)} + 2C_4(\gamma)e^{-C_4(\gamma)n\epsilon_n^2}}{\zeta},
$$

where the last inequality uses the fact that $\log x \leq x - 1$. Choosing $\zeta = C_{10}/2 = \frac{\min(C, \lambda, 1)}{2C_1}$,

$$
\mathbb{E}_{P_0^n}\left[\int_\Theta L_n(\theta, \theta_0)\, dQ_{a',\gamma}^*(\theta|\tilde{X}_n)\right]
$$

$$
\leq M(\gamma)n(\epsilon_n^2) + M'n\eta_n^R(\gamma) + \frac{2(1 + C_0 + 3W^{-\gamma})e^{(-\frac{C_{10}}{2}n\epsilon_n^2)} + 4C_4(\gamma)e^{-C_4(\gamma)n\epsilon_n^2}}{C_{10}} \quad (38)
$$

where $M(\gamma) = C_1 + \frac{1}{\zeta}C_4(\gamma)$ and $M' = \frac{1}{\zeta}$ depend on $C, C_1, C_4(\gamma), W$ and $\lambda$. Since the last two terms in (38) decrease and the first term increases as $n$ increases, we can choose $M'$ large enough, such that for all $n \geq 1$

$$
M'n\eta_n^R(\gamma) > \frac{2(1 + C_0 + 3W^{-\gamma})}{C_{10}} + \frac{4C_4(\gamma)}{C_{10}},
$$

and therefore for $M = 2M'$,

$$
\mathbb{E}_{P_0^n}\left[\int_\Theta L_n(\theta, \theta_0)\, dQ_{a',\gamma}^*(\theta|\tilde{X}_n)\right] \leq M(\gamma)n(\epsilon_n^2) + Mn\eta_n^R(\gamma). \quad (39)
$$

Also, observe that the LHS in the above equation is always positive, therefore $M(\gamma)\epsilon_n^2 + M\eta_n^R(\gamma) \geq 0 \; \forall n \geq 1$ and $\gamma > 0$.

□

### C.3  Proof of Theorem 3.2

**Lemma C.6.** *Given $a' \in \mathcal{A}$ and for a constant M, as defined in Theorem 3.1*

$$
\mathbb{E}_{P_0^n}\left[\sup_{a\in\mathcal{A}}\left|\mathbb{E}_{Q_{a',\gamma}^*(\theta|\tilde{X}_n)}[R(a,\theta)] - R(a,\theta_0)\right|\right] \leq \left[M(\gamma)\epsilon_n^2 + M\eta_n^R(\gamma)\right]^{\frac{1}{2}}. \quad (40)
$$

*Proof.* First, observe that

$$
\left(\sup_{a\in\mathcal{A}}\left|\mathbb{E}_{Q_{a',\gamma}^*(\theta|\tilde{X}_n)}[R(a,\theta)] - R(a,\theta)\right|\right)^2 \leq \left(\mathbb{E}_{Q_{a',\gamma}^*(\theta|\tilde{X}_n)}\left[\sup_{a\in\mathcal{A}}|R(a,\theta) - R(a,\theta_0)|\right]\right)^2
$$

$$
\leq \mathbb{E}_{Q_{a',\gamma}^*(\theta|\tilde{X}_n)}\left[\left(\sup_{a\in\mathcal{A}}|R(a,\theta) - R(a,\theta_0)|\right)^2\right],
$$

where the last inequality follows from Jensen's inequality. Now, using the Jensen's inequality again

$$
\left(\mathbb{E}_{P_0^n}\left[\sup_{a\in\mathcal{A}}\left|\mathbb{E}_{Q_{a',\gamma}^*(\theta|\tilde{X}_n)}[R(a,\theta)] - R(a,\theta_0)\right|\right]\right)^2
$$

$$
\leq \mathbb{E}_{P_0^n}\left[\left(\sup_{a\in\mathcal{A}}\left|\mathbb{E}_{Q_{a',\gamma}^*(\theta|\tilde{X}_n)}[R(a,\theta)] - R(a,\theta_0)\right|\right)^2\right].
$$

Now, using Theorem 3.1 the result follows immediately.

□

*Proof of Theorem 3.2.* Observe that

$$
R(\mathsf{a}_{\mathrm{RS}}^*, \theta_0) - \inf_{z \in \mathcal{A}} R(z, \theta_0)
$$

$$
= |R(\mathsf{a}_{\mathrm{RS}}^*, \theta_0) - \inf_{z \in \mathcal{A}} R(z, \theta_0)|
$$

$$
= R(\mathsf{a}_{\mathrm{RS}}^*, \theta_0) - \mathbb{E}_{Q_{\mathsf{a}_{\mathrm{RS}}^*, \gamma}^*(\theta|\tilde{X}_n)}[R(\mathsf{a}_{\mathrm{RS}}^*, \theta)] + \mathbb{E}_{Q_{\mathsf{a}_{\mathrm{RS}}^*, \gamma}^*(\theta|\tilde{X}_n)}[R(\mathsf{a}_{\mathrm{RS}}^*, \theta)] - \inf_{z \in \mathcal{A}} R(z, \theta_0)
$$

$$
\leq \left| R(\mathsf{a}_{\mathrm{RS}}^*, \theta_0) - \mathbb{E}_{Q_{\mathsf{a}_{\mathrm{RS}}^*, \gamma}^*(\theta|\tilde{X}_n)}[R(\mathsf{a}_{\mathrm{RS}}^*, \theta)] \right| + \left| \mathbb{E}_{Q_{\mathsf{a}_{\mathrm{RS}}^*, \gamma}^*(\theta|\tilde{X}_n)}[R(\mathsf{a}_{\mathrm{RS}}^*, \theta)] - \inf_{a \in \mathcal{A}} R(a, \theta_0) \right|
$$

$$
\leq 2 \sup_{a \in \mathcal{A}} \left| \int R(a, \theta) dQ_{\mathsf{a}_{\mathrm{RS}}^*, \gamma}^*(\theta|\tilde{X}_n) - R(a, \theta_0) \right|. \tag{41}
$$

Given $\mathsf{a}_{\mathrm{RS}}^* \in \mathcal{A}$ and for a constant M (defined in Theorem 3.1), we have from Lemma C.6 for $a' = \mathsf{a}_{\mathrm{RS}}^*$

$$
\mathbb{E}_{P_0^n} \left[ \sup_{a \in \mathcal{A}} \left| \int R(a, \theta) dQ_{\mathsf{a}_{\mathrm{RS}}^*, \gamma}^*(\theta|\tilde{X}_n) - R(a, \theta_0) \right| \right] \leq \left[ M(\gamma)\epsilon_n^2 + M\eta_n^R(\gamma) \right]^{\frac{1}{2}}. \tag{42}
$$

It follows from above that the $P_0^n-$ probability of the following event is at least $1 - \tau^{-1}$:

$$
\left\{ \tilde{X}_n : R(\mathsf{a}_{\mathrm{RS}}^*, \theta_0) - \inf_{z \in \mathcal{A}} R(z, \theta_0) \leq 2\tau \left[ M(\gamma)\epsilon_n^2 + M\eta_n^R(\gamma) \right]^{\frac{1}{2}} \right\}. \tag{43}
$$

$\square$

## C.4 Proofs in Section 3.1

*Proof of Proposition 3.1.* Using the definition of $\eta_n^R(\gamma)$ and the posterior distribution $\Pi(\theta|\tilde{X}_n)$, observe that

$$
n\eta_n^R(\gamma) = \inf_{Q \in \mathcal{Q}} \mathbb{E}_{P_0^n} \left[ \mathrm{KL}(Q\|\Pi_n) - \gamma \inf_{a \in \mathcal{A}} \mathbb{E}_Q[R(a, \theta)] \right]
$$

$$
= \inf_{Q \in \mathcal{Q}} \mathbb{E}_{P_0^n} \left[ \mathrm{KL}(Q\|\Pi) + \int_\Theta dQ(\theta) \log \left( \frac{\int d\Pi(\theta)p(\tilde{X}_n|\theta)}{p(\tilde{X}_n|\theta)} \right) - \gamma \inf_{a \in \mathcal{A}} \mathbb{E}_Q[R(a, \theta)] \right]
$$

$$
= \inf_{Q \in \mathcal{Q}} \left[ \mathrm{KL}(Q\|\Pi) - \gamma \inf_{a \in \mathcal{A}} \mathbb{E}_Q[R(a, \theta)] + \mathbb{E}_{P_0^n} \left[ \mathbb{E}_Q \left[ \log \left( \frac{\int d\Pi(\theta)p(\tilde{X}_n|\theta)}{p(\tilde{X}_n|\theta)} \right) \right] \right] \right].
$$

Now, using Fubini's in the last term of the equation above, we obtain

$$
n\eta_n^R(\gamma) = \inf_{Q \in \mathcal{Q}} \left[ \mathrm{KL}(Q(\theta)\|\Pi(\theta)) - \gamma \inf_{a \in \mathcal{A}} \mathbb{E}_Q[R(a, \theta)] \right.
$$

$$
\left. + \mathbb{E}_Q \left[ \mathrm{KL}\left( dP_0^n \| p(\tilde{X}_n|\theta) \right) - \mathrm{KL}\left( dP_0^n \| \int d\Pi(\theta)p(\tilde{X}_n|\theta) \right) \right] \right]. \tag{44}
$$

Observe that, $\int_{\mathcal{X}^n} \int d\Pi(\theta)p(\tilde{X}_n|\theta)d\tilde{X}_n = 1$. Since, KL is always non-negative, it follows from the equation above that

$$
\eta_n^R(\gamma)
$$

$$
\leq \frac{1}{n} \inf_{Q \in \mathcal{Q}} \left[ \mathrm{KL}\left( Q(\theta)\|\Pi(\theta) \right) - \gamma \inf_{a \in \mathcal{A}} \mathbb{E}_Q[R(a, \theta)] + \mathbb{E}_Q \left[ \mathrm{KL}\left( dP_0^n \| p(\tilde{X}_n|\theta) \right) \right] \right]
$$

$$
\leq \frac{1}{n} \inf_{Q \in \mathcal{Q}} \left[ \mathrm{KL}\left( Q(\theta)\|\Pi(\theta) \right) + \mathbb{E}_Q \left[ \mathrm{KL}\left( dP_0^n \| p(\tilde{X}_n|\theta) \right) \right] \right] - \frac{\gamma}{n} \inf_{Q \in \mathcal{Q}} \inf_{a \in \mathcal{A}} \mathbb{E}_Q[R(a, \theta)], \tag{45}
$$

where the last inequality follows from the following fact, for any functions $f(\cdot)$ and $g(\cdot)$,

$$
\inf(f - g) \leq \inf f - \inf g.
$$

Recall $\epsilon'_n \geq \frac{1}{\sqrt{n}}$. Now, using Assumption 3.1, it is straightforward to observe that the first term in (45),

$$\frac{1}{n} \inf_{Q \in \mathcal{Q}} \left[ \mathrm{KL}\left(Q(\theta)\|\Pi(\theta)\right) + \mathbb{E}_Q \left[ \mathrm{KL}\left(dP_0^n \| p(\tilde{X}_n | \theta)\right) \right] \right] \leq C_9 \epsilon'^2_n. \tag{46}$$

Now consider the last term in (45). Notice that the coefficient of $\frac{1}{n}$ is independent of $n$ and is bounded from below. Therefore, there exist a constant $C_8 = -\inf_{Q \in \mathcal{Q}} \inf_{a \in \mathcal{A}} \mathbb{E}_Q[R(a, \theta)]$, such that with equation (46) it follows that $\eta_n^R(\gamma) \leq \gamma n^{-1} C_8 + C_9 \epsilon'^2_n$ and the result follows.

$\square$

*Proof of Proposition 3.2.* First recall that

$$n\eta_n^R(\gamma) = \inf_{Q \in \mathcal{Q}} \mathbb{E}_{P_0^n} \left[ \mathrm{KL}(Q(\theta)\|\Pi(\theta|\tilde{X}_n)) - \gamma \inf_{a \in \mathcal{A}} \mathbb{E}_Q[R(a, \theta)] \right]$$

$$= \inf_{Q \in \mathcal{Q}} \mathbb{E}_{P_0^n} \left[ \mathrm{KL}(Q(\theta)\|\Pi(\theta|\tilde{X}_n)) \right] - \gamma \inf_{a \in \mathcal{A}} \mathbb{E}_Q[R(a, \theta)]. \tag{47}$$

Observe that the optimization problem is equivalent to solving :

$$\min_{Q \in \mathcal{Q}} \mathbb{E}_{P_0^n} \left[ \mathrm{KL}(Q(\theta)\|\Pi(\theta|\tilde{X}_n)) \right] \text{ s.t. } - \inf_{a \in \mathcal{A}} \mathbb{E}_Q[R(a, \theta)] \leq 0. \tag{48}$$

Now for any $\gamma > 0$, $Q_\gamma^*(\theta) \in \mathcal{Q}$ that minimizes the objective in (47) is primal feasible if

$$- \inf_{a \in \mathcal{A}} \int_\Theta dQ_\gamma^*(\theta) R(a, \theta) \leq 0.$$

Therefore, it is straightforward to observe that as $\gamma$ increases $n\eta_n^R(\gamma)$ decreases that is

$$\mathbb{E}_{P_0^n} \left[ \int_\Theta dQ_\gamma^*(\theta) \log \frac{dQ_\gamma^*(\theta)}{d\Pi(\theta|\tilde{X}_n)} - \gamma \inf_{a \in \mathcal{A}} \int_\Theta dQ_\gamma^*(\theta) R(a, \theta) \right].$$

$\square$

## C.5  Sufficient conditions on $R(a, \theta)$ for existence of tests

To show the existence of test functions, as required in Assumption 2.1, we will use the following result from [11, Theorem 7.1], that is applicable only to distance measures that are bounded above by the Hellinger distance.

**Lemma C.7** (Theorem 7.1 of [11])**.** *Suppose that for some non-increasing function $D(\epsilon)$, some $\epsilon_n > 0$ and for every $\epsilon > \epsilon_n$,*

$$N\left(\frac{\epsilon}{2}, \{P_\theta : \epsilon \leq m(\theta, \theta_0) \leq 2\epsilon\}, m\right) \leq D(\epsilon),$$

*where $m(\cdot, \cdot)$ is any distance measure bounded above by Hellinger distance. Then for every $\epsilon > \epsilon_n$, there exists a test $\phi_n$ (depending on $\epsilon > 0$) such that, for every $j \geq 1$,*

$$\mathbb{E}_{P_0^n}[\phi_n] \leq D(\epsilon) \exp\left(-\frac{1}{2}n\epsilon^2\right) \frac{1}{1 - \exp\left(-\frac{1}{2}n\epsilon^2\right)}, \text{ and}$$

$$\sup_{\{\theta \in \Theta_n(\epsilon) : m(\theta, \theta_0) > j\epsilon\}} \mathbb{E}_{P_\theta^n}[1 - \phi_n] \leq \exp\left(-\frac{1}{2}n\epsilon^2 j\right).$$

*Proof of Lemma C.7:* Refer Theorem 7.1 of [11]. $\square$

For the remaining part of this subsection we assume that $\Theta \subseteq \mathbb{R}^d$. In the subsequent paragraph, we state further assumptions on the risk function to show $L_n(\cdot, \cdot)$ as defined in (6) satisfies Assumption 2.1. For brevity we denote $n^{-1/2}\sqrt{L_n(\theta, \theta_0)}$ by $d_L(\theta, \theta_0)$, that is

$$d_L(\theta_1, \theta_2) := \sup_{a \in \mathcal{A}} |R(a, \theta_1) - R(a, \theta_2)|, \ \forall \{\theta_1, \theta_2\} \in \Theta \tag{49}$$

and the covering number of the set $T(\epsilon) := \{P_\theta : d_L(\theta, \theta_0) < \epsilon\}$ as $N(\delta, T(\epsilon), d_L)$, where $\delta > 0$ is the radius of each ball in the cover. We assume that the risk function $R(a, \cdot)$ satisfies the following bound.

**Assumption C.1.** *The model risk satisfies*

$$d_L(\theta_1, \theta_2)| \leq K_1 d_H(\theta, \theta_0),$$

*where $d_H(\theta_1, \theta_2)$ is the Hellinger distance between two models $P_{\theta_1}$ and $P_{\theta_2}$.*

For instance, suppose the definition of model risk is $R(a, \theta) = \int_{\mathcal{X}} \ell(x, a) p(y|\theta) dx$, where $\ell(x, a)$ is an underlying loss function. Then, observe that Assumption C.1 is trivially satisfied if $\ell(x, a)$ is bounded in $x$ for a given $a \in \mathcal{A}$ and $\mathcal{A}$ is compact, since $d_L(\theta_1, \theta_2)$ can be bounded by the total variation distance $d_{TV}(\theta_1, \theta_2) = \frac{1}{2} \int |dP_{\theta_1}(x) - dP_{\theta_2}(x)|$ and total variation distance is bounded above by the Hellinger distance [12]. Under the assumption above it also follows that we can apply Lemma C.7 to the metric $d_L(\cdot, \cdot)$ defined in (49). Now, we will also assume an additional regularity condition on the risk function.

**Assumption C.2.** *For every $\{\theta_1, \theta_2\} \in \Theta$, there exists a constant $K_2 > 0$ such that*

$$d_L(\theta_1, \theta_2) \leq K_2 \|\theta_1 - \theta_2\|,$$

We can now show that the covering number of the set $T(\epsilon)$ satisfies

**Lemma C.8.** *Given $\epsilon > \delta > 0$, and under Assumption C.2,*

$$N(\delta, T(\epsilon), d_L) < \left( \frac{2\epsilon}{\delta} + 2 \right)^d. \tag{50}$$

*Proof of Lemma C.8:* For any positive $k$ and $\epsilon$, let $\theta \in [\theta_0 - k\epsilon, \theta_0 + k\epsilon]^d \subset \Theta \subset \mathbb{R}^d$. Now consider a set $H_i = \{\theta_i^0, \theta_i^1, \ldots \theta_i^J, \theta_i^{J+1}\}$ and $H = \bigotimes_d H_i$ with $J = \lfloor \frac{2k\epsilon}{\delta'} \rfloor$, where $\theta_i^j = \theta_0 - k\epsilon + i\delta'$ for $j = \{0, 1, \ldots, J\}$ and $\theta_i^{J+1} = \theta_0 + k\epsilon$. Observe that for any $\theta \in [\theta_0 - k\epsilon, \theta_0 + k\epsilon]^d$, there exists a $\theta^j \in H$ such that $\|\theta - \theta^j\| < \delta'$. Hence, union of the $\delta'-$balls for each element in set $H$ covers $[\theta_0 - k\epsilon, \theta_0 + k\epsilon]^d$, therefore $N(\delta', [\theta_0 - k\epsilon, \theta_0 + k\epsilon]^d, \|\cdot\|) = (J + 2)^d$.

Now, due to Assumption C.2, for any $\theta \in [\theta_0 - k\epsilon, \theta_0 + k\epsilon]^d$

$$d_L(\theta, \theta_0) \leq K_2 \|\theta - \theta^j\| \leq K_2 \delta',$$

For brevity, we denote $n^{-1} L_n(\theta, \theta_0)$ by $d_L(\theta, \theta_0)$, that is

$$d_L(\theta_1, \theta_2) := \sup_{a \in \mathcal{A}} |R(a, \theta_1) - R(a, \theta_2)|, \ \forall \{\theta_1, \theta_2\} \in \Theta, \tag{51}$$

and the covering number of the set $T(\epsilon) := \{P_\theta : d_L(\theta, \theta_0) < \epsilon\}$ as $N(\delta, T(\epsilon), d_L)$, where $\delta > 0$ is the radius of each ball in the cover.

Hence, $\delta'$-cover of set $[\theta_0 - k\epsilon, \theta_0 + k\epsilon]^d$ is $K_1 \delta'$ cover of set $T(\epsilon)$ with $k = 1/K_2$. Finally,

$$N(K_2 \delta', T(\epsilon), d_L) \leq (J + 2)^d \leq \left( \frac{2k\epsilon}{\delta'} + 2 \right)^d = \left( \frac{2\epsilon}{K_2 \delta'} + 2 \right)^d$$

which implies for $\delta = K_2 \delta'$,

$$N(\delta, T(\epsilon), d_L) \leq \left( \frac{2\epsilon}{\delta} + 2 \right)^s.$$

$\square$

Observe that the RHS in (50) is a decreasing function of $\delta$, infact for $\delta = \epsilon/2$, it is a constant in $\epsilon$. Therefore, using Lemmas C.7 and C.8, we show in the following result that $L_n(\theta, \theta_0)$ in (6) satisfies Assumption 2.1.

**Lemma C.9.** *Fix $n \geq 1$. For a given $\epsilon_n > 0$ and every $\epsilon > \epsilon_n$, such that $n\epsilon_n^2 \geq 1$. Under Assumption C.1 and C.2, $L_n(\theta, \theta_0) = n \left(\sup_{a \in \mathcal{A}} |R(a, \theta) - R(a, \theta_0)|\right)^2$ satisfies*

$$\mathbb{E}_{P_0^n}[\phi_n] \leq C_0 \exp(-Cn\epsilon^2), \tag{52}$$

$$\sup_{\{\theta \in \Theta : L_n(\theta, \theta_0) \geq C_1 n\epsilon^2\}} \mathbb{E}_{P_\theta^n}[1 - \phi_n] \leq \exp(-Cn\epsilon^2), \tag{53}$$

*where $C_0 = 2 * 10^s$ and $C = \frac{C_1}{2K_1^2}$ for a constant $C_1 > 0$.*

*Proof of Lemma C.9:* Recall $d_L(\theta, \theta_0) = (\sup_{a \in \mathcal{A}} |R(a, \theta) - R(a, \theta_0)|)$ and $T(\epsilon) = \{P_\theta : d_L(\theta, \theta_0) < \epsilon\}$. Using Lemma C.8, observe that for every $\epsilon > \epsilon_n > 0$,

$$N \left(\frac{\epsilon}{2}, \{\theta : \epsilon \leq d_L(\theta, \theta_0) \leq 2\epsilon\}, d_L\right) \leq N \left(\frac{\epsilon}{2}, \{\theta : d_L(\theta, \theta_0) \leq 2\epsilon\}, d_L\right) < 10^d.$$

Next, using Assumption C.1 we have

$$d_L(\theta, \theta_0) \leq K_1 d_H(\theta, \theta_0).$$

It follows from the above two observations and Lemma 2 that, for every $\epsilon > \epsilon_n > 0$, there exist tests $\{\phi_{n,\epsilon}\}$ such that

$$\mathbb{E}_{P_0^n}[\phi_{n,\epsilon}] \leq 10^d \frac{\exp(-C'n\epsilon^2)}{1 - \exp(-C'n\epsilon^2)}, \tag{54}$$

$$\sup_{\{\theta \in \Theta : d_L(\theta, \theta_0) \geq \epsilon\}} \mathbb{E}_{P_\theta^n}[1 - \phi_{n,\epsilon}] \leq \exp(-C'n\epsilon^2), \tag{55}$$

where $C' = \frac{1}{2K_1^2}$. Since the above two conditions hold for every $\epsilon > \epsilon_n$, we can choose a constant $K > 0$ such that for every $\epsilon > \epsilon_n$

$$\mathbb{E}_{P_0^n}[\phi_{n,\epsilon}] \leq 10^d \frac{\exp(-C'K^2n\epsilon^2)}{1 - \exp(-C'K^2n\epsilon^2)} \leq 2(10^d)e^{-C'K^2n\epsilon^2}, \tag{56}$$

$$\sup_{\{\theta \in \Theta : L_n(\theta, \theta_0) \geq K^2n\epsilon^2\}} \mathbb{E}_{P_\theta^n}[1 - \phi_{n,\epsilon}] = \sup_{\{\theta \in \Theta : d_L(\theta, \theta_0) \geq K\epsilon\}} \mathbb{E}_{P_\theta^n}[1 - \phi_{n,\epsilon}] \leq e^{-C'K^2n\epsilon^2}, \tag{57}$$

where the second inequality in (56) holds $\forall n \geq n_0$, where $n_0 := \min\{n \geq 1 : C'K^2n\epsilon^2 \geq \log(2)\}$. Hence, the result follows for $C_1 = K^2$ and $C = C'K^2$. $\square$

Since $L_n(\theta, \theta_0) = \frac{1}{n}d_L^2$ satisfies Assumption 2.1, Theorem 3.1 implies the following bound.

**Corollary C.1.** *Fix $a' \in \mathcal{A}$ and $\gamma > 0$. Let $\epsilon_n$ be a sequence such that $\epsilon_n \to 0$ as $n \to \infty$, $n\epsilon_n^2 \geq 1$ and*

$$L_n(\theta, \theta_0) = n \left(\sup_{a \in \mathcal{A}} |R(a, \theta) - R(a, \theta_0)|\right)^2.$$

*Then under the Assumptions of Theorem 3.1 and Lemma C.9 ; for $C = \frac{C_1}{2K_1^2}$, $C_0 = 2 * 10^s$, $C_1 > 0$ such that $\min(C, C_4(\gamma) + C_5(\gamma)) > C_2 + C_3 + C_4(\gamma) + 2$, and for $\eta_n^R(\gamma)$ as defined in Theorem 3.1, the RSVB approximator of the true posterior $Q_{a',\gamma}^*(\theta | \tilde{X}_n)$ satisfies,*

$$\mathbb{E}_{P_0^n} \left[\int_\Theta L_n(\theta, \theta_0) Q_{a',\gamma}^*(\theta | \tilde{X}_n) d\theta\right] \leq n(M(\gamma)\epsilon_n^2 + M\eta_n^R(\gamma)), \tag{58}$$

*for sufficiently large $n$ and for a function $M(\gamma) = 2 \left(C_1 + MC_4(\gamma)\right)$, where $M = \frac{2C_1}{\min(C, \lambda, 1)}$ .*

*Proof of Corollary C.1:* Using Lemma C.9 observe that for any $\Theta_n(\epsilon) \subseteq \Theta$, $L_n(\theta, \theta_0)$ satisfies Assumption 2.1 with $C_0 = 2 * 10^s$, $C = \frac{C_1}{2K_1^2}$ and for any $C_1 > 0$, since

$$\sup_{\{\theta \in \Theta_n(\epsilon) : L_n(\theta, \theta_0) \geq C_1 n\epsilon_n^2\}} \mathbb{E}_{P_\theta^n}[1 - \phi_{n,\epsilon}] \leq \sup_{\{\theta \in \Theta : L_n(\theta, \theta_0) \geq C_1 n\epsilon_n^2\}} \mathbb{E}_{P_\theta^n}[1 - \phi_{n,\epsilon}] \leq e^{-Cn\epsilon_n^2}.$$

Hence, applying Theorem 3.1 the proof follows. $\square$

## C.6 Newsvendor Problem

We fix $n^{-1/2}\sqrt{L_n^{NV}(\theta,\theta_0)} = (\sup_{a\in\mathcal{A}}|R(a,\theta)-R(a,\theta_0)|)$. Next, we aim to show that the exponentially distributed model $P_\theta$ satisfies Assumption 2.1, for distance function $L_n^{NV}(\theta,\theta_0)$. To show this, in the next result we first prove that $d_L^{NV}(\theta,\theta_0) = n^{-1/2}\sqrt{L_n^{NV}(\theta,\theta_0)}$ satisfy Assumption C.1. Also, recall that the square of Hellinger distance between two exponential distributions with rate parameter $\theta$ and $\theta_0$ is $d_H^2(\theta,\theta_0) = 1 - 2\frac{\sqrt{\theta\theta_0}}{\theta+\theta_0} = 1 - 2\frac{\sqrt{\theta_0/\theta}}{1+\theta_0/\theta}$.

**Lemma C.10.** *For any $\theta\in\Theta = [T,\infty)$, and $a\in\mathcal{A}$,*

$$d_L^{NV}(\theta,\theta_0) \leq \left[\frac{\left(\frac{h}{\theta_0}-\frac{h}{T}\right)^2 + (b+h)^2\left(\frac{e^{-\underline{a}T}}{T}-\frac{e^{-\underline{a}\theta_0}}{\theta_0}\right)^2}{d_H^2(T,\theta_0)}\right]^{1/2} d_H(\theta,\theta_0)$$

*where $\underline{a} := \min\{a\in\mathcal{A}\}$ and $\underline{a} > 0$ and $\theta_0$ lies in the interior of $\Theta$.*

*Proof.* Observe that for any $a\in\mathcal{A}$,

$$|R(a,\theta)-R(a,\theta_0)|^2$$
$$= \left|\frac{h}{\theta_0}-\frac{h}{\theta} + (b+h)\left(\frac{e^{-a\theta}}{\theta}-\frac{e^{-a\theta_0}}{\theta_0}\right)\right|^2$$
$$= \left(\frac{h}{\theta_0}-\frac{h}{\theta}\right)^2 + (b+h)^2\left(\frac{e^{-a\theta}}{\theta}-\frac{e^{-a\theta_0}}{\theta_0}\right)^2 + 2\left(\frac{h}{\theta_0}-\frac{h}{\theta}\right)(b+h)\left(\frac{e^{-a\theta}}{\theta}-\frac{e^{-a\theta_0}}{\theta_0}\right)$$
$$\leq \left(\frac{h}{\theta_0}-\frac{h}{\theta}\right)^2 + (b+h)^2\left(\frac{e^{-a\theta}}{\theta}-\frac{e^{-a\theta_0}}{\theta_0}\right)^2, \tag{59}$$

where the last inequality follows since for $\theta\geq\theta_0$, $\left(\frac{h}{\theta_0}-\frac{h}{\theta}\right)\geq 0$ and $\left(\frac{e^{-a\theta}}{\theta}-\frac{e^{-a\theta_0}}{\theta_0}\right) < 0$ and vice versa if $\theta < \theta_0$ that together makes the last term in the penultimate equality negative for all $\theta\in\Theta$. Moreover, the first derivative of the upperbound with respect to $\theta$ is

$$2\left(\frac{h}{\theta_0}-\frac{h}{\theta}\right)\frac{h}{\theta^2} - 2(b+h)^2\left(\frac{e^{-a\theta}}{\theta}-\frac{e^{-a\theta_0}}{\theta_0}\right)e^{-a\theta}\left[\frac{1}{\theta^2}+\frac{a}{\theta}\right],$$

and it is negative when $\theta\leq\theta_0$ and positive when $\theta > \theta_0$ for all $b > 0, h > 0$, and $a\in\mathcal{A}$. Therefore, the upperbound in (59) above is decreasing function of $\theta$ for all $\theta\leq\theta_0$ and increasing function of $\theta$ for all $\theta > \theta_0$. The upperbound is tight at $\theta = \theta_0$.

Now recall that the squared Hellinger distance between two exponential distributions with rate parameter $\theta$ and $\theta_0$ is

$$d_H^2(\theta,\theta_0) = 1 - 2\frac{\sqrt{\theta\theta_0}}{\theta+\theta_0} = 1 - 2\frac{\sqrt{\theta_0/\theta}}{1+\theta_0/\theta} = \frac{(1-\sqrt{\theta_0/\theta})^2}{1+(\sqrt{\theta_0/\theta})^2}.$$

Note that for $\theta\leq\theta_0$, $d_H^2(\theta,\theta_0)$ is a decreasing function of $\theta$ and for all $\theta > \theta_0$ it is an increasing function of $\theta$. Also, note that as $\theta\to\infty$, the squared Hellinger distance as well as the upperbound computed in (59) converges to a constant for a given $h, b, \theta_0$ and $a$. However, as $\theta\to 0$, the $d_H^2(\theta,\theta_0)\to 1$ but the upperbound computed in (59) diverges.

Since, $\Theta = [T,\infty)$ for some $T > 0$ and $T\leq\theta_0$, observe that if we scale $d_H^2(\theta,\theta_0)$ by factor by which the upperbound computed in (59) is greater than $d_H$ at $\theta = T$, then

$$\left(\frac{h}{\theta_0}-\frac{h}{\theta}\right)^2 + (b+h)^2\left(\frac{e^{-a\theta}}{\theta}-\frac{e^{-a\theta_0}}{\theta_0}\right)^2$$
$$\leq \frac{\left(\frac{h}{\theta_0}-\frac{h}{T}\right)^2 + (b+h)^2\left(\frac{e^{-aT}}{T}-\frac{e^{-a\theta_0}}{\theta_0}\right)^2}{d_H^2(T,\theta_0)}d_H^2(\theta,\theta_0)$$
$$\leq \frac{\left(\frac{h}{\theta_0}-\frac{h}{T}\right)^2 + (b+h)^2\left(\frac{e^{-\underline{a}T}}{T}-\frac{e^{-\underline{a}\theta_0}}{\theta_0}\right)^2}{d_H^2(T,\theta_0)}d_H^2(\theta,\theta_0),$$

where $\underline{a} = \inf\{a : a \in \mathcal{A}\}$ and in the last inequality we used the fact that $\left(\frac{e^{-aT}}{T} - \frac{e^{-a\theta_0}}{\theta_0}\right)^2$ is a decreasing function of $a$ for any $b, h, T$, and $\theta_0$. Since, the RHS in the equation above does not depend on $a$, it follows from the result in (59) and the definition of $L_n^{NV}(\theta, \theta_0)$ that

$$d_L^{NV}(\theta, \theta_0) \leq \left[\frac{\left(\frac{h}{\theta_0} - \frac{h}{T}\right)^2 + (b+h)^2 \left(\frac{e^{-aT}}{T} - \frac{e^{-a\theta_0}}{\theta_0}\right)^2}{d_H^2(T, \theta_0)}\right]^{1/2} d_H(\theta, \theta_0).$$

$\square$

**Lemma C.11.** *For any $\theta \in \Theta = [T, \infty)$, for sufficiently small $T > 0$, and $\theta_0$ lying in the interior of $\Theta$, we have*

$$d_H^2(\theta, \theta_0) = 1 - 2\frac{\sqrt{\theta\theta_0}}{\theta + \theta_0} \leq \left(\frac{\theta_0}{(T+\theta_0)^2}\left(\sqrt{\frac{\theta_0}{T}} - \sqrt{\frac{T}{\theta_0}}\right)\right)|\theta - \theta_0|.$$

*Proof.* Observe that

$$\frac{\partial d_H^2(\theta, \theta_0)}{\partial \theta} = -2\frac{(\theta+\theta_0)\frac{\sqrt{\theta_0}}{2\sqrt{\theta}} - \sqrt{\theta\theta_0}}{(\theta + \theta_0)^2} = \frac{\theta_0}{(\theta+\theta_0)^2}\left(\sqrt{\frac{\theta}{\theta_0}} - \sqrt{\frac{\theta_0}{\theta}}\right).$$

Observe that $\theta \to 0$, $\frac{\partial d_H^2(\theta, \theta_0)}{\partial \theta} \to \infty$. Since, $\theta \in \Theta = [T, \infty)$, therefore the $\sup_{\theta \in \Theta}\left|\frac{\partial d_H^2(\theta, \theta_0)}{\partial \theta}\right| < \infty$. In fact, for sufficiently small $T > 0$, $\sup_{\theta \in \Theta}\left|\frac{\partial d_H^2(\theta, \theta_0)}{\partial \theta}\right| = \left|\frac{\theta_0}{(T+\theta_0)^2}\left(\sqrt{\frac{T}{\theta_0}} - \sqrt{\frac{\theta_0}{T}}\right)\right| = \left(\frac{\theta_0}{(T+\theta_0)^2}\left(\sqrt{\frac{\theta_0}{T}} - \sqrt{\frac{T}{\theta_0}}\right)\right)$. Now the result follows immediately since the derivative of $d_H^2(\theta, \theta_0)$ is bounded on $\Theta$, which implies that $d_H^2(\theta, \theta_0)$ is Lipschitz on $\Theta$. $\square$

**Lemma C.12.** *For any $\theta \in \Theta = [T, \infty)$, and $a \in \mathcal{A}$,*

$$d_L^{NV}(\theta, \theta_0) \leq \frac{h}{T^2}|\theta - \theta_0|.$$

*Proof.* Recall,

$$R(a, \theta) = ha - \frac{h}{\theta} + (b+h)\frac{e^{-a\theta}}{\theta}.$$

First, observe that for any $a \in \mathcal{A}$,

$$\frac{\partial R(a, \theta)}{\partial \theta} = \frac{h}{\theta^2} - a(b+h)\frac{e^{-a\theta}}{\theta} - (b+h)\frac{e^{-a\theta}}{\theta^2} = \frac{1}{\theta^2}\left(h - (b+h)e^{-a\theta}(1 + a\theta)\right) \leq \frac{h}{\theta^2}. \quad (60)$$

The result follows immediately, since $\sup_{\theta \in \Theta}\frac{\partial R(a, \theta)}{\partial \theta} \leq \frac{h}{T^2}$. $\square$

*Proof.* Proof of Lemma B.1

It follows from Lemma C.10 that $d_L^{NV}(\theta, \theta_0)$ for any $\theta \in \Theta = [T, \infty)$ and $\theta_0$ lying the interior of $\Theta$, satisfies Assumption C.1 with

$$K_1 = \left[\frac{\left(\frac{h}{\theta_0} - \frac{h}{T}\right)^2 + (b+h)^2\left(\frac{e^{-aT}}{T} - \frac{e^{-a\theta_0}}{\theta_0}\right)^2}{d_H^2(T, \theta_0)}\right]^{1/2} := K_1^{NV}$$

. Similarly, it follows from Lemma and C.12 that for sufficiently small $T > 0$, $d_L^{NV}(\theta, \theta_0)$ satisfies Assumption C.2 with $K_2 = h/T^2 := K_2^{NV}$. Now using similar arguments as used in Lemma C.8 and Lemma 2.1, for a given $\epsilon_n > 0$ and every $\epsilon > \epsilon_n$, such that $n\epsilon_n^2 \geq 1$, it can be shown that , $L_n^{NV}(\theta, \theta_0) = n\left(\sup_{a \in \mathcal{A}}|R(a, \theta) - R(a, \theta_0)|\right)^2$ satisfies

$$\mathbb{E}_{P_0^n}[\phi_n] \le C_0 \exp(-Cn\epsilon^2), \tag{61}$$

$$\sup_{\{\theta \in \Theta : L_n^{NV}(\theta,\theta_0) \ge C_1 n\epsilon^2\}} \mathbb{E}_{P_\theta^n}[1 - \phi_n] \le \exp(-Cn\epsilon^2), \tag{62}$$

where $C_0 = 20$ and $C = \frac{C_1}{2(K_1^{NV})^2}$ for a constant $C_1 > 0$. $\qquad\square$

*Proof.* Proof of Lemma B.2:

First, we write the Rényi divergence between $P_0^n$ and $P_\theta^n$,

$$D_{1+\lambda}\left(P_0^n \| P_\theta^n\right) = \frac{1}{\lambda} \log \int \left(\frac{dP_0^n}{dP_\theta^n}\right)^\lambda dP_0^n = n\frac{1}{\lambda} \log \int \left(\frac{dP_0}{dP_\theta}\right)^\lambda dP_0$$

$$= n\left(\log \frac{\theta_0}{\theta} + \frac{1}{\lambda} \log \frac{\theta_0}{(\lambda+1)\theta_0 - \lambda\theta}\right),$$

when $((\lambda+1)\theta_0 - \lambda\theta) > 0$ and $D_{1+\lambda}\left(P_0^n \| P_\theta^n\right) = \infty$ otherwise. Also, observe that, $D_{1+\lambda}\left(P_0^n \| P_\theta^n\right)$ is non-decreasing in $\lambda$ (this also follows from non-decreasing property of the Rényi divergence with respect to $\lambda$). Therefore, observe that

$$\Pi(D_{1+\lambda}\left(P_0^n \| P_\theta^n\right) \le C_3 n\epsilon_n^2) \ge \Pi(D_\infty\left(P_0^n \| P_\theta^n\right) \le C_3 n\epsilon_n^2) = \Pi\left(0 \le \log \frac{\theta_0}{\theta} \le C_3 \epsilon_n^2\right)$$

$$= \Pi\left(\theta_0 e^{-C_3\epsilon_n^2} \le \theta \le \theta_0\right).$$

Now, recall that for a set $A \subseteq \Theta = [T, \infty)$, we define $\Pi(A) = \mathrm{Inv} - \Gamma(A \cap \Theta)/\mathrm{Inv} - \Gamma(\Theta)$. Now, observe that for sufficiently small $T$ and large enough $n$, we have

$$\Pi\left(\theta_0 e^{-C_3\epsilon_n^2} \le \theta \le \theta_0\right) \ge \mathrm{Inv} - \Gamma\left(\theta_0 e^{-C_3\epsilon_n^2} \le \theta \le \theta_0\right)$$

The cumulative distribution function of inverse-gamma distribution is $\mathrm{Inv} - \Gamma(\{\theta \in \Theta : \theta < t\}) := \frac{\Gamma\left(\alpha, \frac{\beta}{t}\right)}{\Gamma(\alpha)}$, where $\alpha(> 0)$ is the shape parameter, $\beta(> 0)$ is the scale parameter, $\Gamma(\cdot)$ is the Gamma function, and $\Gamma(\cdot, \cdot)$ is the incomplete Gamma function. Therefore, it follows for $\alpha > 1$ that

$$\mathrm{Inv} - \Gamma\left(\theta_0 e^{-C_3\epsilon_n^2} \le \theta \le \theta_0\right)$$

$$= \frac{\Gamma(\alpha, \beta/\theta_0) - \Gamma\left(\alpha, \beta/\theta_0 e^{C_3\epsilon_n^2}\right)}{\Gamma(\alpha)} = \frac{\int_{\beta/\theta_0}^{\beta/\theta_0 e^{C_3\epsilon_n^2}} e^{-x} x^{\alpha-1} dx}{\Gamma(\alpha)}$$

$$\ge \frac{e^{-\beta/\theta_0 e^{C_3\epsilon_n^2} + \alpha C_3\epsilon_n^2}}{\alpha\Gamma(\alpha)} \left(\frac{\beta}{\theta_0}\right)^\alpha \left[1 - e^{-\alpha C_3\epsilon_n^2}\right]$$

$$\ge \frac{e^{-\beta/\theta_0 e^{C_3}}}{\alpha\Gamma(\alpha)} \left(\frac{\beta}{\theta_0}\right)^\alpha \left[e^{-\alpha C_3 n\epsilon_n^2}\right]$$

where the penultimate inequality folows since $0 < \epsilon_n^2 < 1$ and the last inequality follows from the fact that, $1 - e^{-\alpha C_3\epsilon_n^2} \ge e^{-\alpha C_3 n\epsilon_n^2}$, for large enough $n$. Also note that, $1 - e^{-\alpha C_3\epsilon_n^2} \ge e^{-\alpha C_3 n\epsilon_n^2}$ can't hold true for $\epsilon_n^2 = 1/n$. However, for $\epsilon_n^2 = \frac{\log n}{n}$ it holds for any $n \ge 2$ when $\alpha C_3 > 2$. Therefore, for inverse-Gamma prior restricted to $\Theta$, $C_2 = \alpha C_3$ and any $\lambda > 1$ the result follows for sufficiently large $n$.

$\qquad\square$

*Proof.* Proof of Lemma B.3: Recall,

$$R(a, \theta) = ha - \frac{h}{\theta} + (b+h)\frac{e^{-a\theta}}{\theta}.$$

First, observe that for any $a \in \mathcal{A}$,

$$\frac{\partial R(a, \theta)}{\partial \theta} = \frac{h}{\theta^2} - a(b+h)\frac{e^{-a\theta}}{\theta} - (b+h)\frac{e^{-a\theta}}{\theta^2} = \frac{1}{\theta^2}\left(h - (b+h)e^{-a\theta}(1 + a\theta)\right). \tag{63}$$

Using the above equation the (finite) critical point $\theta^*$ must satisfy, $h - (b+h)e^{-a\theta^*}(1+a\theta^*) = 0$. Therefore,

$$R(a,\theta) \geq R(a,\theta^*) = h\left(a - \frac{1}{\theta^*} + \frac{1}{\theta^*(1+a\theta^*)}\right) = \frac{ha^2\theta^*}{(1+a\theta^*)}.$$

Since $h, b > 0$ and $a\theta^* > 0$, hence

$$R(a,\theta) \geq \frac{h\underline{a}^2\theta^*}{(1+a\theta^*)},$$

where $\underline{a} := \min\{a \in \mathcal{A}\}$ and $\underline{a} > 0$.

$\square$

*Proof.* Proof of Lemma B.4:

First, observe that $R(a,\theta)$ is bounded above in $\theta$ for a given $a \in \mathcal{A}$

$$R(a,\theta) = ha - \frac{h}{\theta} + (b+h)\frac{e^{-a\theta}}{\theta}$$

$$\leq ha + \frac{b}{\theta}.$$

Using the above fact and the Cauchy-Schwarz inequality, we obtain

$$\int_{\left\{e^{\gamma R(a,\theta)} > e^{C_4(\gamma)n\epsilon_n^2}\right\}} e^{\gamma R(a,\theta)}\pi(\theta)d\theta$$

$$\leq \left(\int e^{2\gamma R(a,\theta)}\pi(\theta)d\theta\right)^{1/2}\left(\int \mathbb{1}_{e^{\gamma R(a,\theta)} > e^{C_4(\gamma)n\epsilon_n^2}}\pi(\theta)d\theta\right)^{1/2}$$

$$\leq \left(\int e^{2\gamma\left(ha+\frac{b}{\theta}\right)}\pi(\theta)d\theta\right)^{1/2}\left(\int \mathbb{1}_{\left\{e^{\gamma\left(ha+\frac{b}{\theta}\right)} > e^{C_4(\gamma)n\epsilon_n^2}\right\}}\pi(\theta)d\theta\right)^{1/2}$$

$$\leq e^{-C_4(\gamma)n\epsilon_n^2}\left(\int e^{2\gamma\left(ha+\frac{b}{\theta}\right)}\pi(\theta)d\theta\right), \tag{64}$$

where the last inequality follows from using the Chebyshev's inequality.

Now using the definition of the prior distribution, which is an inverse gamma prior restricted to $\Theta = [T, \infty)$, we have

$$\int_{\left\{e^{\gamma R(a,\theta)} > e^{C_4(\gamma)n\epsilon_n^2}\right\}} e^{\gamma R(a,\theta)}\pi(\theta)d\theta \leq e^{-C_4(\gamma)n\epsilon_n^2}\left(\int e^{2\gamma\left(ha+\frac{b}{\theta}\right)}\pi(\theta)d\theta\right)$$

$$\leq e^{-C_4(\gamma)n\epsilon_n^2}e^{2\gamma\left(h\bar{a}+\frac{b}{T}\right)},$$

where $\bar{a} := \max\{a \in \mathcal{A}\}$ and $\bar{a} > 0$. Since $n\epsilon_n^2 \geq 1$, we must fix $C_4(\gamma)$ such that $e^{C_4(\gamma)} > e^{2\gamma\left(h\bar{a}+\frac{b}{T}\right)}$, that is $C_4(\gamma) > 2\gamma\left(h\bar{a} + \frac{b}{T}\right)$ and $C_5(\gamma) = C_4(\gamma) - 2\gamma\left(h\bar{a} + \frac{b}{T}\right)$.

$\square$

*Proof.* Proof of Lemma B.5: Since family $\mathcal{Q}$ contains all shifted-gamma distributions, observe that $\{q_n(\cdot) \in \mathcal{Q}\}\forall n \geq 1$. By definition, $q_n(\theta) = \frac{n^n}{\theta_0^n\Gamma(n)}(\theta - T)^{n-1}e^{-n\frac{(\theta-T)}{\theta_0}}$. Now consider the first term; using the definition of the KL divergence it follows that

$$\text{KL}(q_n(\theta)\|\pi(\theta)) = \int_T^\infty q_n(\theta)\log(q_n(\theta))d\theta - \int_T^\infty q_n(\theta)\log(\pi(\theta))d\theta. \tag{65}$$

Substituting $q_n(\theta)$ in the first term of the equation above and expanding the logarithm term, we obtain

$$\int_T^\infty q_n(\theta)\log(q_n(\theta))d\theta$$

$$= (n-1)\int_T^\infty \log(\theta - T)\frac{n^n}{\theta_0^n\Gamma(n)}(\theta - T)^{n-1}e^{-n\frac{\theta-T}{\theta_0}}d\theta - n + \log\left(\frac{n^n}{\theta_0^n\Gamma(n)}\right)$$

$$= -\log\theta_0 + (n-1)\int_T^\infty \log\frac{\theta - T}{\theta_0}\frac{n^n}{\theta_0^n\Gamma(n)}(\theta - T)^{n-1}e^{-n\frac{\theta-T}{\theta_0}}d\theta - n + \log\left(\frac{n^n}{\Gamma(n)}\right) \tag{66}$$

Now consider the second term in the equation above. Substitute $\theta = \frac{t\theta_0}{n} + T$ into the integral, we have

$$\int_T^\infty \log \frac{\theta - T}{\theta_0} \frac{n^n}{\theta_0^n \Gamma(n)} (\theta - T)^{n-1} e^{-n\frac{\theta-T}{\theta_0}} d\theta = \int_0^\infty \log \frac{t}{n} \frac{1}{\Gamma(n)} t^{n-1} e^{-t} dt$$

$$\leq \int \left(\frac{t}{n} - 1\right) \frac{1}{\Gamma(n)} t^{n-1} e^{-t} dt = 0. \qquad (67)$$

Substituting the above result into (66), we get

$$\int_T^\infty q_n(\theta) \log(q_n(\theta)) d\theta \leq -\log \theta_0 - n + \log \left(\frac{n^n}{\Gamma(n)}\right)$$

$$\leq -\log \theta_0 - n + \log \left(\frac{n^n}{\sqrt{2\pi n} n^{n-1} e^{-n}}\right)$$

$$= -\log \sqrt{2\pi} \theta_0 + \frac{1}{2} \log n, \qquad (68)$$

where the second inequality uses the fact that $\sqrt{2\pi n} n^n e^{-n} \leq n\Gamma(n)$. Recall $\pi(\theta) = \frac{\beta^\alpha}{\Gamma(\alpha)} \theta^{-\alpha-1} e^{-\frac{\beta}{\theta}}$. Now consider the second term in (65). Using the definition of inverse-gamma prior and expanding the logarithm function, we have

$$-\int_T^\infty q_n(\theta) \log(\pi(\theta)) d\theta$$

$$= -\log \left(\frac{\beta^\alpha}{\Gamma(\alpha)}\right) + (\alpha+1) \int_T^\infty \log \theta \frac{n^n}{\theta_0^n \Gamma(n)} (\theta - T)^{n-1} e^{-n\frac{\theta-T}{\theta_0}} d\theta + \beta \frac{n}{(n-1)\theta_0}$$

$$= -\log \left(\frac{\beta^\alpha}{\Gamma(\alpha)}\right) + \int_T^\infty \log \frac{\theta}{\theta_0} \frac{n^n}{\theta_0^n \Gamma(n)} (\theta - T)^{n-1} e^{-n\frac{\theta-T}{\theta_0}} d\theta$$

$$+ \beta \frac{n}{(n-1)\theta_0} + (\alpha+1) \log \theta_0$$

$$\leq -\log \left(\frac{\beta^\alpha}{\Gamma(\alpha)}\right) + \int_T^\infty \frac{\theta - T}{\theta_0} \frac{n^n}{\theta_0^n \Gamma(n)} (\theta - T)^{n-1} e^{-n\frac{\theta-T}{\theta_0}} d\theta$$

$$+ \beta \frac{n}{(n-1)\theta_0} + (\alpha+1) \log \theta_0$$

$$= -\log \left(\frac{\beta^\alpha}{\Gamma(\alpha)}\right) + \beta \frac{n}{(n-1)\theta_0} + (\alpha+1) \log \theta_0, \qquad (69)$$

where the first inequality is due to fact that $\mathbb{E}_{q_n}[\beta/\theta] \leq \mathbb{E}_{q_n}[\beta/(\theta - T)]$ for any $\theta > T$ and the penultimate inequality follows from the observation in (67) and the fact that $\log \frac{\theta}{\theta_0} \leq \frac{\theta}{\theta_0} - 1 \leq \frac{\theta}{\theta_0} - \frac{T}{\theta_0}$ for any $\theta_0 > T$. Substituting (69) and (68) into (65) and dividing either sides by $n$, we obtain

$$\frac{1}{n} \text{KL}(q_n(\theta) \| \pi(\theta))$$

$$\leq \frac{1}{n} \left(-\log \sqrt{2\pi} \theta_0 + \frac{1}{2} \log n - \log \left(\frac{\beta^\alpha}{\Gamma(\alpha)}\right) + \beta \frac{n}{(n-1)\theta_0} + (\alpha+1) \log \theta_0\right)$$

$$= \frac{1}{2} \frac{\log n}{n} + \beta \frac{1}{(n-1)\theta_0} + \frac{1}{n} \left(-\log \sqrt{2\pi} - \log \left(\frac{\beta^\alpha}{\Gamma(\alpha)}\right) + (\alpha) \log \theta_0\right). \qquad (70)$$

Now consider the second term in the assertion of the lemma. Since $\xi_i, i \in \{1, 2 \dots n\}$ are independent and identically distributed, we obtain

$$\frac{1}{n} \mathbb{E}_{q_n(\theta)} \left[\text{KL}\left(dP_0^n \| p(\tilde{X}_n | \theta)\right)\right] = \mathbb{E}_{q_n(\theta)} \left[\text{KL}\left(dP_0 \| p(\xi | \theta)\right)\right]$$

Now using the expression for KL divergence between the two exponential distributions, we have

$$\frac{1}{n} \mathbb{E}_{q_n(\theta)} \left[\text{KL}\left(dP_0^n \| p(\tilde{X}_n | \theta)\right)\right] = \int_T^\infty \left(\log \frac{\theta_0}{\theta} + \frac{\theta}{\theta_0} - 1\right) \frac{n^n}{\theta_0^n \Gamma(n)} (\theta - T)^{n-1} e^{-n\frac{\theta-T}{\theta_0}} d\theta$$

$$\leq \frac{n}{n-1} + 1 - 2 = \frac{1}{n-1}, \qquad (71)$$

where second inequality uses the fact that $\log x \leq x - 1 \leq x - \frac{T}{\theta_0}$ for $\theta_0 > T$. Combined together (71) and (70) for $n \geq 2$ implies that

$$\frac{1}{n}\left[\text{KL}\left(q_n(\theta)\|\pi(\theta)\right) + \mathbb{E}_{q_n(\theta)}\left[\text{KL}\left(dP_0^n\|p(\tilde{X}_n|\theta)\right)\right]\right]$$
$$\leq \frac{1}{2}\frac{\log n}{n} + \frac{1}{n}\left(2 + \frac{2\beta}{\theta_0} - \log\sqrt{2\pi} - \log\left(\frac{\beta^\alpha}{\Gamma(\alpha)}\right) + \alpha\log\theta_0\right) \leq C_9\frac{\log n}{n}. \qquad (72)$$

where $C_9 := \frac{1}{2} + \max\left(0, 2 + \frac{2\beta}{\theta_0} - \log\sqrt{2\pi} - \log\left(\frac{\beta^\alpha}{\Gamma(\alpha)}\right) + \alpha\log\theta_0\right)$ and the result follows. $\square$

*Proof.* Proof of Lemma B.5: Since family $\mathcal{Q}$ contains all gamma distributions, observe that $\{q_n(\cdot) \in \mathcal{Q}\}\forall n \geq 1$. By definition, $q_n(\theta) = \frac{n^n}{\theta_0^n\Gamma(n)}\theta^{n-1}e^{-n\frac{\theta}{\theta_0}}$. Now consider the first term; using the definition of the KL divergence it follows that

$$\text{KL}(q_n(\theta)\|\pi(\theta)) = \int q_n(\theta)\log(q_n(\theta))d\theta - \int q_n(\theta)\log(\pi(\theta))d\theta. \qquad (73)$$

Substituting $q_n(\theta)$ in the first term of the equation above and expanding the logarithm term, we obtain

$$\int q_n(\theta)\log(q_n(\theta))d\theta = (n-1)\int\log\theta\frac{n^n}{\theta_0^n\Gamma(n)}\theta^{n-1}e^{-n\frac{\theta}{\theta_0}}d\theta - n + \log\left(\frac{n^n}{\theta_0^n\Gamma(n)}\right)$$
$$= -\log\theta_0 + (n-1)\int\log\frac{\theta}{\theta_0}\frac{n^n}{\theta_0^n\Gamma(n)}\theta^{n-1}e^{-n\frac{\theta}{\theta_0}}d\theta - n + \log\left(\frac{n^n}{\Gamma(n)}\right) \qquad (74)$$

Now consider the second term in the equation above. Substitute $\theta = \frac{t\theta_0}{n}$ into the integral, we have

$$\int\log\frac{\theta}{\theta_0}\frac{n^n}{\theta_0^n\Gamma(n)}\theta^{n-1}e^{-n\frac{\theta}{\theta_0}}d\theta = \int\log\frac{t}{n}\frac{1}{\Gamma(n)}t^{n-1}e^{-t}dt$$
$$\leq \int\left(\frac{t}{n} - 1\right)\frac{1}{\Gamma(n)}t^{n-1}e^{-t}dt = 0. \qquad (75)$$

Substituting the above result into (74), we get

$$\int q_n(\theta)\log(q_n(\theta))d\theta \leq -\log\theta_0 - n + \log\left(\frac{n^n}{\Gamma(n)}\right)$$
$$\leq -\log\theta_0 - n + \log\left(\frac{n^n}{\sqrt{2\pi n}n^{n-1}e^{-n}}\right)$$
$$= -\log\sqrt{2\pi}\theta_0 + \frac{1}{2}\log n, \qquad (76)$$

where the second inequality uses the fact that $\sqrt{2\pi n}n^n e^{-n} \leq n\Gamma(n)$. Recall $\pi(\theta) = \frac{\beta^\alpha}{\Gamma(\alpha)}\theta^{-\alpha-1}e^{-\frac{\beta}{\theta}}$. Now consider the second term in (73). Using the definition of inverse-gamma prior and expanding the logarithm function, we have

$$-\int q_n(\theta)\log(\pi(\theta))d\theta$$
$$= -\log\left(\frac{\beta^\alpha}{\Gamma(\alpha)}\right) + (\alpha+1)\int\log\theta\frac{n^n}{\theta_0^n\Gamma(n)}\theta^{n-1}e^{-n\frac{\theta}{\theta_0}}d\theta + \beta\frac{n}{(n-1)\theta_0}$$
$$= -\log\left(\frac{\beta^\alpha}{\Gamma(\alpha)}\right) + (\alpha+1)\int\log\frac{\theta}{\theta_0}\frac{n^n}{\theta_0^n\Gamma(n)}\theta^{n-1}e^{-n\frac{\theta}{\theta_0}}d\theta$$
$$+ \beta\frac{n}{(n-1)\theta_0} + (\alpha+1)\log\theta_0$$
$$\leq -\log\left(\frac{\beta^\alpha}{\Gamma(\alpha)}\right) + \beta\frac{n}{(n-1)\theta_0} + (\alpha+1)\log\theta_0, \qquad (77)$$

where the last inequality follows from the observation in (75). Substituting (77) and (76) into (73) and dividing either sides by $n$, we obtain

$$
\frac{1}{n}\mathrm{KL}(q_n(\theta)\|\pi(\theta))
$$
$$
\leq \frac{1}{n}\left(-\log\sqrt{2\pi}\theta_0 + \frac{1}{2}\log n - \log\left(\frac{\beta^\alpha}{\Gamma(\alpha)}\right) + \beta\frac{n}{(n-1)\theta_0} + (\alpha+1)\log\theta_0\right)
$$
$$
= \frac{1}{2}\frac{\log n}{n} + \beta\frac{1}{(n-1)\theta_0} + \frac{1}{n}\left(-\log\sqrt{2\pi} - \log\left(\frac{\beta^\alpha}{\Gamma(\alpha)}\right) + (\alpha)\log\theta_0\right). \tag{78}
$$

Now, consider the second term in the assertion of the lemma. Since, $\xi_i, i \in \{1, 2 \ldots n\}$ are independent and identically distributed, we obtain

$$
\frac{1}{n}\mathbb{E}_{q(\theta)}\left[\mathrm{KL}\left(dP_0^n\|p(\tilde{X}_n|\theta)\right)\right] = \mathbb{E}_{q_n(\theta)}\left[\mathrm{KL}\left(dP_0\|p(\xi|\theta)\right)\right]
$$

Now using the expression for KL divergence between the two exponential distributions, we have

$$
\frac{1}{n}\mathbb{E}_{q(\theta)}\left[\mathrm{KL}\left(dP_0^n\|p(\tilde{X}_n|\theta)\right)\right] = \int\left(\log\frac{\theta_0}{\theta} + \frac{\theta}{\theta_0} - 1\right)\frac{n^n}{\theta_0^n\Gamma(n)}\theta^{n-1}e^{-n\frac{\theta}{\theta_0}}\,d\theta
$$
$$
\leq \frac{n}{n-1} + 1 - 2 = \frac{1}{n-1}, \tag{79}
$$

where second inequality uses the fact that $\log x \leq x - 1$. Combined together (79) and (78) for $n \geq 2$ implies that

$$
\frac{1}{n}\left[\mathrm{KL}\left(q(\theta)\|\pi(\theta)\right) + \mathbb{E}_{q(\theta)}\left[\mathrm{KL}\left(dP_0^n\|p(\tilde{X}_n|\theta)\right)\right]\right]
$$
$$
\leq \frac{1}{2}\frac{\log n}{n} + \frac{1}{n}\left(2 + \frac{2\beta}{\theta_0} - \log\sqrt{2\pi} - \log\left(\frac{\beta^\alpha}{\Gamma(\alpha)}\right) + \alpha\log\theta_0\right) \leq C_9\frac{\log n}{n}. \tag{80}
$$

where $C_9 := \frac{1}{2} + \max\left(0, 2 + \frac{2\beta}{\theta_0} - \log\sqrt{2\pi} - \log\left(\frac{\beta^\alpha}{\Gamma(\alpha)}\right) + \alpha\log\theta_0\right)$ and the result follows. $\qquad\square$

### C.7 Multi-product Newsvendor problem

In the multi-dimensional newsvendor problem, we fix $n^{-1/2}\sqrt{L_n^{MNV}(\theta,\theta_0)} = (\sup_{a\in\mathcal{A}}|R(a,\theta) - R(a,\theta_0)|)$, where $R(a,\theta) = \sum_{i=1}^d\left[(h_i + b_i)a_i\Phi(a_i) - b_ia_i + \theta_i(b_i - h_i)\right.$ $+\sigma_{ii}\left[h\frac{\phi((a_i-\theta_i)/\sigma_{ii})}{\Phi((a_i-\theta_i)/\sigma_{ii})} + b\frac{\phi((a_i-\theta_i)/\sigma_{ii})}{1-\Phi((a_i-\theta_i)/\sigma_{ii})}\right]\right]$.

For brevity, we denote $d_L^{MNV}(\theta,\theta_0) = n^{-1/2}\sqrt{L_n^{MNV}(\theta,\theta_0)}$. First, we show that

**Lemma C.13.** *For any compact decision space $\mathcal{A}$ and compact model space $\Theta$,*

$$
d_L^{MNV}(\theta,\theta_0) \leq K\|\theta - \theta_0\|,
$$

*for a constant $K$ depending on compact sets $\mathcal{A}$ and $\Theta$ and given $b, h$ and $\Sigma$.*

*Proof.* Observe that

$$
\partial_{\theta_i}R(a,\theta)
$$
$$
= (b_i - h_i) + (a_i - \theta_i)/\sigma_{ii}\phi((a_i - \theta_i)/\sigma_{ii})\left[\frac{h}{\Phi((a_i-\theta_i)/\sigma_{ii})} + \frac{b}{1-\Phi((a_i-\theta_i)/\sigma_{ii})}\right]
$$
$$
+ \sigma_{ii}\phi\left(\frac{(a_i-\theta_i)}{\sigma_{ii}}\right)\left[\frac{h\phi((a_i-\theta_i)/\sigma_{ii})}{\sigma_{ii}\Phi((a_i-\theta_i)/\sigma_{ii})^2} - \frac{b\phi((a_i-\theta_i)/\sigma_{ii})}{\sigma_{ii}(1-\Phi((a_i-\theta_i)/\sigma_{ii}))^2}\right]
$$
$$
= (b_i - h_i) + (a_i - \theta_i)/\sigma_{ii}\phi((a_i - \theta_i)/\sigma_{ii})\left[\frac{h}{\Phi((a_i-\theta_i)/\sigma_{ii})} + \frac{b}{1-\Phi((a_i-\theta_i)/\sigma_{ii})}\right]
$$
$$
+ \phi\left(\frac{(a_i-\theta_i)}{\sigma_{ii}}\right)\left[\frac{h\phi((a_i-\theta_i)/\sigma_{ii})}{\Phi((a_i-\theta_i)/\sigma_{ii})^2} - \frac{b\phi((a_i-\theta_i)/\sigma_{ii})}{(1-\Phi((a_i-\theta_i)/\sigma_{ii}))^2}\right]. \tag{81}
$$

Since, $\mathcal{A}$ and $\Theta$ are compact sets, therefore $\{(a_i - \theta_i)/\sigma_{ii}\}_{i=1}^d$ lie in a compact set. Consequently, $\phi((a_i - \theta_i)/\sigma_{ii})$ and $\Phi((a_i - \theta_i)/\sigma_{ii})$ also lie in bounded subset of $\mathbb{R}$ and thus $\sup_{\mathcal{A},\Theta} \|\partial_{\theta_i} R(a, \theta)\| \leq K$ for a given $b$, $h$ and $\Sigma$. Since , the norm of the derivative of $R(a, \theta)$ is bounded on $\Theta$ for any $a \in \mathcal{A}$, therefore, $d_L^{MNV}(\theta, \theta_0)$ is uniformly Lipschitz in $\mathcal{A}$ with Lipschitz constant $K$, that is

$$d_L^{MNV}(\theta, \theta_0) \leq K\|\theta - \theta_0\|.$$

$\square$

Next, we show that the $P_\theta$ satisfies Assumption 2.1, for distance function $L_n^{MNV}(\theta, \theta_0)$.

*Proof.* Proof of Lemma B.6:

First consider the following test function, constructed using $\tilde{X}_n = \{\xi_1, \xi_2, \ldots, \xi_n\}$.

$$\phi_{n,\epsilon} := \mathbb{1}_{\left\{\tilde{X}_n : \|\hat{\theta}_n - \theta_0\| > \sqrt{C\epsilon^2}\right\}},$$

where $\hat{\theta}_n = \frac{\sum_{i=1}^n \xi_i}{n}$. Note that $\hat{\theta}_n - \theta_0 \sim \mathcal{N}(\cdot|0, \frac{1}{n}\Sigma)$, where $\frac{1}{n}\Sigma$ is a symmetric positive definite matrix. Therefore it can be decomposed as $\Sigma = Q^T \Lambda Q$, where $Q$ is an orthogonal matrix and $\Lambda$ is a daigonal matrix consisting of respective eigen values and consequently $\hat{\theta}_n - \theta_0 \sim Q\mathcal{N}(\cdot|0, \frac{1}{n}\Lambda)$. So, we have $\|\hat{\theta}_n - \theta_0\|^2 \sim \|\mathcal{N}(\cdot|0, \frac{1}{n}\Lambda)\|^2$. Notice that $\|\mathcal{N}(\cdot|0, \frac{1}{n}\Lambda)\|^2$ is a linear combination of $d$ $\chi^2_{(1)}$ random variable weighted by elements of the diagonal matrix $\frac{1}{n}\Lambda$. Using this observation, we first verify that $\phi_{n,\epsilon}$ satisfies condition (*i*) of the Lemma. Observe that

$$\mathbb{E}_{P_0^n}[\phi_n] = P_0^n\left(\tilde{X}_n : \left\|\hat{\theta}_n - \theta_0\right\|^2 > C\epsilon^2\right) = P_0^n\left(\tilde{X}_n : \|\mathcal{N}(\cdot|0, \Lambda)\|^2 > Cn\epsilon^2\right).$$

Note that $\chi^2_{(1)}$ is $\Gamma$ distributed with shape $1/2$ and scale $2$, which implies $\chi^2_{(1)} - 1$ is a sub-gamma random variable with scale factor $2$ and variance factor $2$. Now observe that for $\hat{\Lambda} = \max_{i \in \{1,2,\ldots d\}} \Lambda_{ii}$,

$$\begin{aligned}
P_0^n\left(\tilde{X}_n : \|\mathcal{N}(\cdot|0, \Lambda)\|^2 > Cn\epsilon^2\right) &\leq P_0^n\left(\tilde{X}_n : \chi^2_{(1)} > \frac{1}{d\hat{\Lambda}}Cn\epsilon^2\right) \\
&\leq P_0^n\left(\tilde{X}_n : \chi^2_{(1)} > \frac{1}{d\hat{\Lambda}}Cn\epsilon^2\right) \\
&= P_0^n\left(\tilde{X}_n : \chi^2_{(1)} - 1 > \frac{1}{d\hat{\Lambda}}Cn\epsilon^2 - 1\right) \\
&\leq e^{-\frac{\left(\frac{1}{d\hat{\Lambda}}Cn\epsilon^2 - 1\right)^2}{2\left(2 + 2\left(\frac{1}{d\hat{\Lambda}}Cn\epsilon^2 - 1\right)\right)}} \\
&\leq e^{-1/8\frac{1}{d\hat{\Lambda}}Cn\epsilon^2 + 1/8} \leq e^{-1/8\left(\frac{C}{d\hat{\Lambda}} - 1\right)n\epsilon^2}, \quad (82)
\end{aligned}$$

where in the third inequality we used the well known tail bound for sub-gamma random variable (Lemma 3.12 [5]) assuming that $C$ is sufficiently large such that $\left(\frac{1}{d\hat{\Lambda}}Cn\epsilon^2 - 1\right) > 1$ and in the last inequality follows from the assumption that $n\epsilon^2 > n\epsilon_n^2 \geq 1$.

Now, we fix the alternate set to be $\{\theta \in \mathbb{R}^d : \|\theta - \theta_0\| \geq 2\sqrt{C\epsilon^2}\}$. Next, we verify that $\phi_{n,\epsilon}$ satisfies condition (*ii*) of the lemma. First, observe that

$$\mathbb{E}_{P_\theta^n}[1 - \phi_n] = P_\theta^n\left(\tilde{X}_n : \left\|\hat{\theta}_n - \theta_0\right\|^2 \leq C\epsilon^2\right) \leq P_\theta^n\left(\tilde{X}_n : \|\hat{\theta}_n - \theta\| \geq \|\theta - \theta_0\| - \sqrt{C\epsilon^2}\right), \tag{83}$$

where in the last inequality, we used the fact that $\|\theta - \theta_0\| \leq \|\hat{\theta}_n - \theta\| + \left\|\hat{\theta}_n - \theta_0\right\|$. Now on alternate set $\{\theta \in \mathbb{R}^d : \|\theta - \theta_0\| \geq 2\sqrt{C\epsilon^2}\}$,

$$\begin{aligned}
\mathbb{E}_{P_\theta^n}[1 - \phi_n] &\leq P_\theta^n\left(\tilde{X}_n : \|\hat{\theta}_n - \theta\| \geq \|\theta - \theta_0\| - \sqrt{C\epsilon^2}\right) \\
&\leq P_\theta^n\left(\tilde{X}_n : \|\hat{\theta}_n - \theta\| \geq \|\theta - \theta_0\| - \sqrt{C\epsilon^2}\right) \\
&\leq P_\theta^n\left(\tilde{X}_n : \|\hat{\theta}_n - \theta\| \geq \sqrt{C\epsilon^2}\right). \quad (84)
\end{aligned}$$

Now, it follows from (82) and $\Theta \subset \mathbb{R}^d$ that

$$\mathbb{E}_{P_0^n}[\phi_n] \leq e^{-1/8\left(\frac{C}{d\hat{\Lambda}} - 1\right)n\epsilon^2},$$

$$\sup_{\{\theta \in \Theta : \|\theta - \theta_0\| \geq 2\sqrt{C\epsilon^2}\}} \mathbb{E}_{P_\theta^n}[1 - \phi_n] \leq \sup_{\{\theta \in \mathbb{R}^d : \|\theta - \theta_0\| \geq 2\sqrt{C\epsilon^2}\}} \mathbb{E}_{P_\theta^n}[1 - \phi_n] \leq e^{-1/8\left(\frac{C}{d\hat{\Lambda}} - 1\right)n\epsilon^2}.$$

Using Lemma C.13, $\{\theta \in \Theta : n^{-1/2}\sqrt{L_n^{MNV}(\theta, \theta_0)} \geq 2K\sqrt{C\epsilon^2}\} = \{\theta \in \Theta : d_L^{MNV}(\theta, \theta_0) \geq 2K\sqrt{C\epsilon^2}\} \subseteq \{\theta \in \Theta : \|\theta - \theta_0\| \geq 2\sqrt{C\epsilon^2}\}$, which implies that

$$\sup_{\{\theta \in \Theta : L_n^{MNV}(\theta, \theta_0) \geq 4K^2 Cn\epsilon^2\}} \mathbb{E}_{P_\theta^n}[1 - \phi_n] \leq \sup_{\{\theta \in \Theta : \|\theta - \theta_0\| \geq 2\sqrt{C\epsilon^2}\}} \mathbb{E}_{P_\theta^n}[1 - \phi_n].$$

Therefore, $P_\theta$ for $\theta \in \Theta$, satisifes Assumptions 2.1 for $L_n(\theta, \theta_0) = L_n^{MNV}(\theta, \theta_0)$ for $C_0 = 1$, $C_1 = 4K^2 C$ and $C = 1/8\left(\frac{C}{d\hat{\Lambda}} - 1\right)$. $\qquad\square$

*Proof.* Proof of Lemma B.7:

First, we write the Rényi divergence between two multivariate Gaussian distribution with known $\Sigma$ as

$$D_{1+\lambda}(\mathcal{N}(\cdot|\theta_0)\|\mathcal{N}(\cdot|\theta)) = \frac{\lambda + 1}{2}(\theta - \theta_0)^T\Sigma(\theta - \theta_0), \tag{85}$$

and $D_{1+\lambda}(\mathcal{N}(\cdot|\theta)\|\mathcal{N}(\cdot|\theta_0)) < \infty$ if and only if $\Sigma^{-1}$ is positive definite [13].

Since, we assumed that the sequence of models are iid, therefore, $D_{1+\lambda}(P_0^n\|P_\theta^n) = \frac{1}{\lambda}\log\int\left(\frac{dP_0^n}{dP_\theta^n}\right)^\lambda dP_0^n = n\frac{1}{\lambda}\log\int\left(\frac{dP_0}{dP_\theta}\right)^\lambda dP_0 = n\left(\frac{\lambda+1}{2}(\theta - \theta_0)^T\Sigma(\theta - \theta_0)\right)$, when $\Sigma^{-1}$ is positive definite and $D_{1+\lambda}(P_0^n\|P_\theta^n) = \infty$ otherwise. Now observe that

$$\Pi(D_{1+\lambda}(P_0^n\|P_\theta^n) \leq nC_3\epsilon_n^2) = \Pi\left(\left((\theta - \theta_0)^T\Sigma(\theta - \theta_0)\right) \leq \frac{2}{\lambda + 1}C_3\epsilon_n^2\right)$$

$$= \Pi\left(\left([(\theta - \theta_0)Q]^T\Lambda[Q(\theta - \theta_0)]\right) \leq \frac{2}{\lambda + 1}C_3\epsilon_n^2\right)$$

$$\geq \Pi\left(\left([(\theta - \theta_0)Q]^T[Q(\theta - \theta_0)]\right) \leq \frac{2}{\hat{\Lambda}(\lambda + 1)}C_3\epsilon_n^2\right),$$

$$= \Pi\left(\left([(\theta - \theta_0)]^T[(\theta - \theta_0)]\right) \leq \frac{2}{\hat{\Lambda}(\lambda + 1)}C_3\epsilon_n^2\right), \tag{86}$$

where $\hat{\Lambda} = \max_{i \in \{1,2,\ldots d\}}\Lambda_{ii}$ and in the second equality we used eigen value decomposition of $\Sigma = Q^T\Lambda Q$. Next, observe that,

$$\Pi(D_{1+\lambda}(P_0^n\|P_\theta^n) \leq nC_3\epsilon_n^2) = \Pi\left(\left([(\theta - \theta_0)]^T[(\theta - \theta_0)]\right) \leq \frac{2}{\hat{\Lambda}(\lambda + 1)}C_3\epsilon_n^2\right)$$

$$= \Pi\left(\|(\theta - \theta_0)\| \leq \sqrt{\frac{2}{\hat{\Lambda}(\lambda + 1)}C_3\epsilon_n^2}\right)$$

$$\geq \Pi\left(\|(\theta - \theta_0)\|_\infty \leq \sqrt{\frac{2}{\hat{\Lambda}(\lambda + 1)}C_3\epsilon_n^2}\right)$$

$$= \prod_{i=1}^d \Pi_i\left(|(\theta_i - \theta_0^i)| \leq \sqrt{\frac{2}{\hat{\Lambda}(\lambda + 1)}C_3\epsilon_n^2}\right),$$

where in the last equality we used the fact that the prior distribution is uncorrelated. Now, the result follows immediately for sufficiently large $n$, if the prior distribution is uncorrelated and uniformly

distributed on the compact set $\Theta_i$, for each $i \in \{1, 2, \ldots, d\}$ . In particular observe that for large enough $n$, we have

$$\Pi(D_{1+\lambda}\left(P_0^n \| P_\theta^n\right) \leq nC_3\epsilon_n^2) \geq \prod_{i=1}^d \frac{\theta_0^i + \sqrt{\frac{2}{\hat{\Lambda}(\lambda+1)}C_3\epsilon_n^2} - \theta_0^i + \sqrt{\frac{2}{\hat{\Lambda}(\lambda+1)}C_3\epsilon_n^2}}{m(\Theta_i)}$$

$$= \frac{2^d \left(\frac{2}{\hat{\Lambda}(\lambda+1)}C_3\epsilon_n^2\right)^{d/2}}{\prod_{i=1}^d m(\Theta_i)} = \left(\frac{8}{(\hat{\Lambda}(\lambda+1))}\left(\prod_{i=1}^d m(\Theta_i)\right)^{-2/d}C_3\epsilon_n^2\right)^{d/2},$$

where $m(A)$ is the Lebesgue measure (volume) of any set $A \subset \mathbb{R}$. Now if $\epsilon_n^2 = \frac{\log n}{n}$, then for $\frac{8}{\hat{\Lambda}(\lambda+1)\left(\prod_{i=1}^d m(\Theta_i)\right)^{2/d}}C_3 > 2$, $\frac{8}{\hat{\Lambda}(\lambda+1)\left(\prod_{i=1}^d m(\Theta_i)\right)^{2/d}}C_3\epsilon_n^2 \geq e^{-\frac{8}{\hat{\Lambda}(\lambda+1)\left(\prod_{i=1}^d m(\Theta_i)\right)^{2/d}}C_3 n\epsilon_n^2}$ for all $n \geq 2$, therefore,

$$\Pi(D_{1+\lambda}\left(P_0^n \| P_\theta^n\right) \leq nC_3\epsilon_n^2) \geq e^{-\frac{4d}{\hat{\Lambda}(\lambda+1)\left(\prod_{i=1}^d m(\Theta_i)\right)^{2/d}}C_3 n\epsilon_n^2}.$$

$\square$

*Proof.* Proof of Lemma B.9: Since family $\mathcal{Q}$ contains all uncorrelated Gaussian distributions restricted to $\Theta$, observe that $\{q_n(\cdot) \in \mathcal{Q}\}\forall n \geq 1$. By definition, $q_n^i(\theta) \propto \frac{1}{\sqrt{2\pi\sigma_{i,n}^2}}e^{-\frac{1}{2\sigma_{i,n}^2}(\theta-\mu_{i,n})^2}\mathbb{1}_{\Theta_i} = \frac{\mathcal{N}(\theta_i|\mu_{i,n},\sigma_{i,n})\mathbb{1}_{\Theta_i}}{\mathcal{N}(\Theta_i|\mu_{i,n},\sigma_{i,n})}$ and fix $\sigma_{i,n} = 1/\sqrt{n}$ and $\theta_i = \theta_0^i$ for all $i \in \{1, 2, \ldots, d\}$. Now consider the first term; using the definition of the KL divergence it follows that

$$\mathrm{KL}(q_n(\theta)\|\pi(\theta)) = \int q_n(\theta) \log(q_n(\theta))d\theta - \int q_n(\theta) \log(\pi(\theta))d\theta. \tag{87}$$

Substituting $q_n(\theta)$ in the first term of the equation above and expanding the logarithm term, we obtain

$$\int q_n(\theta) \log(q_n(\theta))d\theta = \sum_{i=1}^d \int q_n^i(\theta_i) \log(q_n^i(\theta_i))d\theta_i$$

$$\leq \sum_{i=1}^d \int \mathcal{N}(\theta_i|\mu_{i,n},\sigma_{i,n}) \log \mathcal{N}(\theta_i|\mu_{i,n},\sigma_{i,n})d\theta_i$$

$$= -\sum_{i=1}^d [\log(\sqrt{2\pi e}) + \log \sigma_{i,n}], \tag{88}$$

where in the last equality, we used the well known expression for the differential entropy of Gaussian distributions. Recall $\pi(\theta) = \prod_{i=1}^d \frac{1}{m(\Theta_i)}$. Now consider the second term in (87). It is straightforward to observe that,

$$-\int q_n(\theta) \log(\pi(\theta))d\theta = \sum_{i=1}^d \log(m(\Theta_i)). \tag{89}$$

Substituting (89) and (88) into (87) and dividing either sides by $n$ and substituting $\sigma_{i,n}$, we obtain

$$\frac{1}{n}\mathrm{KL}(q_n(\theta)\|\pi(\theta)) \leq -\frac{1}{n}\sum_{i=1}^d [\log(\sqrt{2\pi e}) - \log(m(\Theta_i)) - \frac{1}{2}\log n]$$

$$= \frac{d}{2}\frac{\log n}{n} - \frac{1}{n}\sum_{i=1}^d [\log(\sqrt{2\pi e}) - \log(m(\Theta_i))]. \tag{90}$$

Now, consider the second term in the assertion of the lemma. Since $\xi_i, i \in \{1, 2 \ldots n\}$ are independent and identically distributed, we obtain

$$\frac{1}{n}\mathbb{E}_{q_n(\theta)}\left[\mathrm{KL}\left(dP_0^n\|p(\tilde{X}_n|\theta)\right)\right] = \mathbb{E}_{q_n(\theta)}\left[\mathrm{KL}\left(dP_0\|p(\xi|\theta)\right)\right]$$

Now using the expression for KL divergence between the two multivariate Gaussian distributions, we have

$$\frac{1}{n}\mathbb{E}_{q_n(\theta)}\left[\text{KL}\left(dP_0^n\|p(\tilde{X}_n|\theta)\right)\right] = \frac{1}{2}\mathbb{E}_{q_n(\theta)}\left[(\theta-\theta_0)^T\Sigma^{-1}(\theta-\theta_0)\right]$$

$$\leq \frac{\check{\Lambda}^{-1}}{2}\mathbb{E}_{q_n(\theta)}\left[(\theta-\theta_0)^T(\theta-\theta_0)\right]$$

$$\leq \frac{d}{n}\frac{\check{\Lambda}^{-1}}{2} \tag{91}$$

where $\check{\Lambda} = \min_{i\in\{1,2,\ldots d\}}\Lambda_{ii}$, and $\Sigma^{-1} = Q^T\Lambda^{-1}Q$, where $Q$ is an orthogonal matrix and $\Lambda$ is a daigonal matrix consisting of the respective eigen values of $\Sigma$. Combined together (91) and (90) implies that

$$\frac{1}{n}\left[\text{KL}\left(q_n(\theta)\|\pi(\theta)\right) + \mathbb{E}_{q_n(\theta)}\left[\text{KL}\left(dP_0^n\|p(\tilde{X}_n|\theta)\right)\right]\right]$$

$$\leq \frac{d}{2}\frac{\log n}{n} - \frac{1}{n}\sum_{i=1}^{d}[\log(\sqrt{2\pi e}) - \log(m(\Theta_i))] + \frac{d}{n}\frac{\check{\Lambda}^{-1}}{2} \leq C_9\frac{\log n}{n}. \tag{92}$$

where $C_9 := \frac{d}{2} + \max\left(0, -\sum_{i=1}^{d}[\log(\sqrt{2\pi e}) - \log(m(\Theta_i))] + \frac{d}{2}\check{\Lambda}^{-1}\right)$ and the result follows. $\qquad\square$

## C.8 Gaussian process classification

*Proof of Lemma B.11.* In view of Theorem 7.1 in [11], it suffices to show that

$$N\left(\epsilon, \Theta_n(\epsilon), d_{\text{TV}}\right) \leq e^{\bar{C}n\epsilon^2},$$

for some $\bar{C} > 0$. Now, first observe that

$$d_{\text{TV}}(P_{\theta(y)}, P_{\theta_0(y)}) = \frac{1}{2}\mathbb{E}_\nu\left(|\Psi_1(\theta(y)) - \Psi_1(\theta_0(y))| + |\Psi_{-1}(\theta(y)) - \Psi_{-1}(\theta_0(y))|\right)$$

$$= \mathbb{E}_\nu\left(|\Psi_1(\theta(y)) - \Psi_1(\theta_0(y))|\right)$$

$$\leq \mathbb{E}_\nu\left(|\theta(y) - \theta_0(y)|\right) \leq \|\theta(y) - \theta_0(y)\|_\infty, \tag{93}$$

where the second equality uses the definition of $\Psi_{-1}(\cdot)$. Since, total-variation distance above is bounded above by supremum norm, there exists a constant $0 < c' < 1/2$, such that

$$N\left(\epsilon, \Theta_n(\epsilon), d_{\text{TV}}\right) \leq N\left(c'\epsilon, \Theta_n(\epsilon), \|\cdot\|_\infty\right) \leq e^{\frac{2}{3}c'^2C_{10}n\epsilon^2}, \tag{94}$$

where the last inequality follows from (13) in Lemma B.10. Then if follows from Theorem 7.1 in [11] that for every $\epsilon > \epsilon_n$, there exists a test $\phi_n$ (depending on $\epsilon > 0$) such that, for every $j \geq 1$,

$$\mathbb{E}_{P_0^n}[\phi_n] \leq e^{\frac{2}{3}c'^2C_{10}n\epsilon^2}e^{-\frac{1}{2}n\epsilon^2}\frac{1}{1 - \exp\left(-\frac{1}{2}n\epsilon^2\right)}, \text{and}$$

$$\sup_{\{\theta\in\Theta_n(\epsilon):d_{TV}(P_\theta,P_{\theta_0})>j\epsilon\}}\mathbb{E}_{P_\theta^n}[1 - \phi_n] \leq \exp\left(-\frac{1}{2}n\epsilon^2 j\right).$$

Now for all $n$ such that $n\epsilon^2 > n\epsilon_n^2 > 2\log 2$ and $C_{10} = c'^{-2}/4 > 1$ and $j = 1$, we have

$$\mathbb{E}_{P_0^n}[\phi_n] \leq 2e^{-\frac{1}{3}n\epsilon^2}, \text{and} \tag{95}$$

$$\sup_{\{\theta\in\Theta_n(\epsilon):d_{TV}(P_\theta,P_{\theta_0})>\epsilon\}}\mathbb{E}_{P_\theta^n}[1 - \phi_n] \leq e^{-\frac{1}{2}n\epsilon^2} \leq e^{-\frac{1}{3}n\epsilon^2}. \tag{96}$$

Now observe that

$$\sup_{a\in\mathcal{A}}|G(a,\theta) - G(a,\theta_0)|$$

$$= \max\left(c_+|\mathbb{E}_\nu[\Psi_{-1}(\theta(y))] - \mathbb{E}_\nu[\Psi_{-1}(\theta_0(y))]|, c_-|\mathbb{E}_\nu[\Psi_1(\theta(y))] - \mathbb{E}_\nu[\Psi_1(\theta_0(y))]|\right)$$

$$= \max\left(c_+|\mathbb{E}_\nu[\Psi_1(\theta_0(y))] - \mathbb{E}_\nu[\Psi_1(\theta(y))]|, c_-|\mathbb{E}_\nu[\Psi_1(\theta(y))] - \mathbb{E}_\nu[\Psi_1(\theta_0(y))]|\right)$$

$$= \max(c_+, c_-)|\mathbb{E}_\nu[\Psi_1(\theta_0(y))] - \mathbb{E}_\nu[\Psi_1(\theta(y))]|$$

$$\leq \max(c_+, c_-)\mathbb{E}_\nu[|\Psi_1(\theta_0(y)) - \Psi_1(\theta(y))|]$$

$$\leq \max(c_+, c_-)d_{TV}(P_\theta, P_{\theta_0}) \tag{97}$$

where the second equality uses the fact that $\Psi_{-1}(\cdot) = 1 - \Psi_1(\cdot)$.

Consequently,

$$\{\theta \in \Theta_n(\epsilon) : \sup_{a \in \mathcal{A}} |G(a,\theta) - G(a,\theta_0)| > \max(c_+, c_-)\epsilon\} \subseteq \{\theta \in \Theta_n(\epsilon) : d_{TV}(P_\theta, P_{\theta_0}) > \epsilon\}$$

Therefore, it follows from (95) and (96) and the definition of $L_n(\theta, \theta_0)$ that

$$\mathbb{E}_{P_0^n}[\phi_n] \leq 2e^{-\frac{1}{3}n\epsilon^2}, \text{ and} \tag{98}$$

$$\sup_{\{\theta \in \Theta_n(\epsilon) : L_n(\theta,\theta_0) > (\max(c_+, c_-))^2 n\epsilon^2\}} \mathbb{E}_{P_\theta^n}[1 - \phi_n] \leq e^{-\frac{1}{2}n\epsilon^2} \leq e^{-\frac{1}{3}n\epsilon^2}. \tag{99}$$

Finally, the result follows for $C = 1/3$, $C_0 = 2$ and $C_1 = (\max(c_+, c_-))^2$.

$\square$

*Proof of Lemma B.12.* The Rényi divergence

$$D_{1+\lambda}(P_0^n \| P_\theta^n)$$
$$= n\frac{1}{\lambda} \ln \int \left( \Psi_1(\theta_0(y))^{1+\lambda} \Psi_1(\theta(y))^{-\lambda} + \Psi_{-1}(\theta_0(y))^{1+\lambda} \Psi_{-1}(\theta(y))^{-\lambda} \right) \nu(dy)$$
$$= n\frac{1}{\lambda} \ln \int e^{\lambda \frac{1}{\lambda} \ln\left( \Psi_1(\theta_0(y))^{1+\lambda} \Psi_1(\theta(y))^{-\lambda} + \Psi_{-1}(\theta_0(y))^{1+\lambda} \Psi_{-1}(\theta(y))^{-\lambda} \right)} \nu(dy). \tag{100}$$

Note that the derivative of the exponent in the integrand above with respect to $\theta(y)$ is

$$\frac{\left( -\lambda \Psi_1(\theta_0(y))^{1+\lambda} \Psi_1(\theta(y))^{-\lambda-1} \psi(\theta(y)) + \lambda \Psi_{-1}(\theta_0(y))^{1+\lambda} \Psi_{-1}(\theta(y))^{-\lambda-1} \psi(\theta(y)) \right)}{\left( \Psi_1(\theta_0(y))^{1+\lambda} \Psi_1(\theta(y))^{-\lambda} + \Psi_{-1}(\theta_0(y))^{1+\lambda} \Psi_{-1}(\theta(y))^{-\lambda} \right)}$$

$$= \lambda \psi(\theta(y)) \frac{\left( -\Psi_1(\theta_0(y))^{1+\lambda} \Psi_1(\theta(y))^{-\lambda-1} + \Psi_{-1}(\theta_0(y))^{1+\lambda} \Psi_{-1}(\theta(y))^{-\lambda-1} \right)}{\left( \Psi_1(\theta_0(y))^{1+\lambda} \Psi_1(\theta(y))^{-\lambda} + \Psi_{-1}(\theta_0(y))^{1+\lambda} \Psi_{-1}(\theta(y))^{-\lambda} \right)}$$

$$= \lambda \frac{\psi(\theta(y))}{\Psi_1(\theta(y))\Psi_{-1}(\theta(y))} \frac{\left( -\Psi_1(\theta_0(y))^{1+\lambda} \Psi_{-1}(\theta(y))^{\lambda+1} + \Psi_{-1}(\theta_0(y))^{1+\lambda} \Psi_1(\theta(y))^{\lambda+1} \right)}{\left( \Psi_1(\theta_0(y))^{1+\lambda} \Psi_{-1}(\theta(y))^{\lambda} + \Psi_{-1}(\theta_0(y))^{1+\lambda} \Psi_1(\theta(y))^{\lambda} \right)}$$

$$= \lambda \frac{\left( -\Psi_1(\theta_0(y))^{1+\lambda} \Psi_{-1}(\theta(y))^{\lambda+1} + \Psi_{-1}(\theta_0(y))^{1+\lambda} \Psi_1(\theta(y))^{\lambda+1} \right)}{\left( \Psi_1(\theta_0(y))^{1+\lambda} \Psi_{-1}(\theta(y))^{\lambda} + \Psi_{-1}(\theta_0(y))^{1+\lambda} \Psi_1(\theta(y))^{\lambda} \right)}$$

$$= \lambda \frac{\left( -e^{-(\lambda+1)\theta(y)} + e^{-(1+\lambda)\theta_0(y)} \right)}{\left( e^{-\lambda\theta(y)} + e^{-(\lambda+1)\theta_0(y)} \right) \left( 1 + e^{-\theta(y)} \right)}$$

$$= \lambda \frac{e^{-(1+\lambda)\theta_0(y)} \left( 1 - e^{-(\lambda+1)(\theta(y)-\theta_0(y))} \right)}{\left( e^{-\lambda\theta(y)} + e^{-(\lambda+1)\theta_0(y)} \right) \left( 1 + e^{-\theta(y)} \right)}$$

$$\leq \lambda \frac{(\lambda+1)(\theta(y) - \theta_0(y))}{\left( e^{-\lambda\theta(y)+(\lambda+1)\theta_0(y)} + 1 \right) \left( 1 + e^{-\theta(y)} \right)}$$

$$\leq \lambda(\lambda+1)|\theta(y) - \theta_0(y)|, \tag{101}$$

where in the fourth equality we used definition of the logistic function and the penultimate inequality follows from the well known inequality that $1 - e^{-x} \leq x$. Consequently, using Taylor's theorem it follows that the exponent in the integrand of the Rényi divergence in (100) is bounded above by $\lambda(\lambda+1)|\theta(y) - \theta_0(y)|^2$ and thus by $\lambda(\lambda+1)\|\theta(y) - \theta_0(y)\|_\infty^2$. Therefore,

$$D_{1+\lambda}(P_0^n \| P_\theta^n)$$
$$= n\frac{1}{\lambda} \ln \int \left( \Psi_1(\theta_0(y))^{1+\lambda} \Psi_1(\theta(y))^{-\lambda} + \Psi_{-1}(\theta_0(y))^{1+\lambda} \Psi_{-1}(\theta(y))^{-\lambda} \right) \nu(dy)$$
$$\leq n\frac{1}{\lambda} \ln \int e^{\lambda(\lambda+1)\|\theta(y)-\theta_0(y)\|_\infty^2} \nu(dy)$$
$$= n(\lambda+1)\|\theta(y) - \theta_0(y)\|_\infty^2.$$

Now using the inequality for $C_3 = 16(\lambda + 1)$ above observe that

$$
\begin{aligned}
\Pi(A_n) = \Pi(D_{1+\lambda}(P_0^n \| P_\theta^n) &\leq C_3 n \epsilon_n^2) \\
&\geq \Pi(n(\lambda + 1)\|\theta(y) - \theta_0(y)\|_\infty^2 \leq C_3 n \epsilon_n^2) \\
&= \Pi(\|\theta(y) - \theta_0(y)\|_\infty \leq 4\epsilon_n) \geq e^{-n\epsilon_n^2}
\end{aligned}
\tag{102}
$$

and the result follows from (15) of Lemma B.10.

$\square$

*Proof of Lemma B.13.* Let us first analyze the KL divergence between the prior distribution and variational family. Recall that two Gaussian measures on infinite dimensional spaces are either equivalent or singular. [27, Theorem 6.13] specify the condition required for the two Gaussian measures to be equivalent. In particular, note that $\theta_0^J(\cdot) \in \text{Im}(\mathcal{C}^{1/2})$. Now observe that the covariance operator of $Q_n$ has eigenvalues $\{\zeta_j^2\}_{j=1}^J{}_{k=1}^{2^{jd}}$, therefore operator $S$ in the definition of $\mathcal{C}_q$ has eigenvalues $\{1 - \zeta_j^2/\mu_j^2\}_{j=1}^J{}_{k=1}^{2^{jd}}$. For $\tau_j^2 = 2^{-2ja-jd}$ for any $a > 0$, $\sum_{j=1}^J 2^{jd}\left(\frac{n\epsilon_n^2 2^{-2ja-jd}}{1+n\epsilon_n^2 2^{-2ja-jd}}\right)^2 = \sum_{j=1}^J 2^{-jd}\left(\frac{n\epsilon_n^2 2^{-2ja}}{1+n\epsilon_n^2 2^{-2ja-jd}}\right)^2 < \infty$, therefore $S$ is an HS operator.

For any integer $J \leq J_\alpha$ define $\bar{\theta}_0^J = \int \theta_0^J(y)\nu(dy)$, where $\theta_0^J(\cdot) = \sum_{j=1}^J \sum_{k=1}^{2^{jd}} \theta_{0;j,k}\vartheta_{j,k}(\cdot)$. Since, $\theta_0^J(\cdot) \in \text{Im}(\mathcal{C}^{1/2})$ and $S$ is a symmetric and HS operator, we invoke Theorem 5 in [22], to write

$$
\begin{aligned}
\text{KL}(\mathcal{N}(\bar{\theta}_0^J, \mathcal{C}_q)\|\mathcal{N}(0, \mathcal{C})) &= \frac{1}{2}\|\mathcal{C}^{-1/2}\bar{\theta}_0^J\|^2 - \frac{1}{2}\log\det(I - S) + \frac{1}{2}tr(-S), \\
&= \frac{1}{2}\sum_{j=1}^J\sum_{k=1}^{2^{jd}}\frac{\theta_{0;k,j}^2}{\mu_j^2} - \frac{1}{2}\log\prod_{j=1}^J\prod_{k=1}^{2^{jd}}(1 - \kappa_j^2) - \frac{1}{2}\sum_{j=1}^J\sum_{k=1}^{2^{jd}}\kappa_j^2 \\
&= \frac{1}{2}\sum_{j=1}^J\sum_{k=1}^{2^{jd}}\frac{\theta_{0;k,j}^2}{\mu_j^2} - \frac{1}{2}\log\prod_{j=1}^J(1 - \kappa_j^2)^{2^{jd}} - \frac{1}{2}\sum_{j=1}^J 2^{jd}\kappa_j^2 \\
&= \frac{1}{2}\sum_{j=1}^J\sum_{k=1}^{2^{jd}}\frac{\theta_{0;k,j}^2}{\mu_j^2} - \frac{1}{2}\sum_{j=1}^J 2^{jd}\log(1 - \kappa_j^2) - \frac{1}{2}\sum_{j=1}^J 2^{jd}\kappa_j^2.
\end{aligned}
$$

Now for $\mu_j 2^{jd/2} = 2^{-ja}$, and using the definition of Besov norm of $\theta_0$ denoted as $\|\theta_0\|_{\beta,\infty,\infty}^2$, and denoting $1 - \kappa_j^2 = \frac{1}{1+n\epsilon_n^2\tau_j^2}$, we have

$$
\begin{aligned}
&\text{KL}(\mathcal{N}(\bar{\theta}_0^J, \mathcal{C}_q)\|\mathcal{N}(0, \mathcal{C})) \\
&\leq \frac{1}{2}\sum_{j=1}^J 2^{j(2a-2\beta+d)}\|\theta_0\|_{\beta,\infty,\infty}^2 - \frac{1}{2}\sum_{j=1}^J 2^{jd}\log(1 - \kappa_j^2) - \frac{1}{2}\sum_{j=1}^J 2^{jd}\kappa_j^2 \\
&= \frac{1}{2}\sum_{j=1}^J 2^{j(2a-2\beta+d)}\|\theta_0\|_{\beta,\infty,\infty}^2 - \frac{1}{2}\sum_{j=1}^J 2^{jd}\left(\log(1 - \kappa_j^2) + \kappa_j^2\right) \\
&= \frac{1}{2}\sum_{j=1}^J 2^{j(2a-2\beta+d)}\|\theta_0\|_{\beta,\infty,\infty}^2 + \frac{1}{2}\sum_{j=1}^J 2^{jd}\left(\log(1 + n\epsilon_n^2\tau_j^2) - \frac{n\epsilon_n^2\tau_j^2}{1 + n\epsilon_n^2\tau_j^2}\right) \\
&\leq \frac{1}{2}\sum_{j=1}^J 2^{j(2a-2\beta+d)}\|\theta_0\|_{\beta,\infty,\infty}^2 + \frac{1}{2}\sum_{j=1}^J 2^{jd}\left(n\epsilon_n^2\tau_j^2\right),
\end{aligned}
$$

where the last inequality follows from the fact that, $\log(1+x) - \frac{x}{1+x} \le \frac{x^2}{1+x} \le x$ for $x > 0$. Substituting $\tau_j^2 = 2^{-2ja-jd}$, we have

$$\frac{1}{n}\mathrm{KL}(\mathcal{N}(\bar{\theta}_0^J, \mathcal{C}_q)\|\mathcal{N}(0,\mathcal{C})) \le \frac{1}{2n}\sum_{j=1}^{J} 2^{j(2a-2\beta+d)}\|\theta_0\|_{\beta,\infty,\infty}^2 + \frac{\epsilon_n^2}{2}\sum_{j=1}^{J} 2^{-2ja}$$

$$\le \frac{\|\theta_0\|_{\beta,\infty,\infty}^2}{2n}\sum_{j=1}^{J} 2^{j(2a-2\beta+d)} + \frac{2^{-2a}}{2}\frac{1-2^{-2Ja}}{1-2^{-2a}}\epsilon_n^2.$$

The summation in the first term above is bounded by $\epsilon_n^2$ as derived in [30, Theorem 4.5]. Therefore,

$$\frac{1}{n}\mathrm{KL}(\mathcal{N}(\bar{\theta}_0^J, \mathcal{C}_q)\|\mathcal{N}(0,\mathcal{C})) \le \max\left(\|\theta_0\|_{\beta,\infty,\infty}^2, \frac{2^{-2a} - 2^{-2Ja-2a}}{1-2^{-2a}}\right)\epsilon_n^2. \tag{103}$$

Now consider the second term

$$\frac{1}{n}\mathbb{E}_{Q_n}\mathrm{KL}(P_0^n\|P_\theta^n)$$

$$= \mathbb{E}_{Q_n}\int\left(\Psi_1(\theta_0(y))\log\frac{\Psi_1(\theta_0(y))}{\Psi_1(\theta(y))} + \Psi_{-1}(\theta_0(y))\log\frac{\Psi_{-1}(\theta_0(y))}{\Psi_{-1}(\theta(y))}\right)\nu(dy)$$

$$\le \mathbb{E}_{Q_n}\int\langle\theta(y) - \theta_0(y), \theta(y) - \theta_0(y)\rangle\nu(dy)$$

$$= \mathbb{E}_{Q_n}\int\|\theta(y) - \theta_0^J(y) - (\theta_0(y) - \theta_0^J(y))\|_2^2\nu(dy)$$

$$= \mathbb{E}_{Q_n}\int\|\theta(y) - \theta_0^J(y)\|_2^2 + \|\theta_0(y) - \theta_0^J(y))\|_2^2 - 2\langle\theta(y) - \theta_0^J(y), \theta_0(y) - \theta_0^J(y)\rangle\nu(dy)$$

$$\le \mathbb{E}_{Q_n}\int\|\theta(y) - \theta_0^J(y)\|_2^2\nu(dy) + \|\theta_0(y) - \theta_0^J(y))\|_\infty^2$$

$$= \mathbb{E}_{Q_n}\int|\sum_{j=1}^{J}\sum_{k=1}^{2^{jd}}\zeta_j Z_{j,k}\vartheta_{j,k}(y)|^2\nu(dy) + \|\theta_0(y) - \theta_0^J(y))\|_\infty^2$$

$$\le \mathbb{E}_{Q_n}\sum_{j=1}^{J}\sum_{k=1}^{2^{jd}}\zeta_j^2 Z_{j,k}^2\int\vartheta_{j,k}(y)^2\nu(dy) + \|\theta_0(y) - \theta_0^J(y))\|_\infty^2$$

$$= \sum_{j=1}^{J}\sum_{k=1}^{2^{jd}}\zeta_j^2\mathbb{E}_{Q_n}[Z_{j,k}^2] + \|\theta_0(y) - \theta_0^J(y))\|_\infty^2$$

$$= \sum_{j=1}^{J}\sum_{k=1}^{2^{jd}}\mu_j^2(1 - \kappa_j^2) + \|\theta_0(y) - \theta_0^J(y))\|_\infty^2$$

$$= \sum_{j=1}^{J} 2^{jd}\frac{\mu_j^2}{1 + n\epsilon_n^2\tau_j^2} + \|\theta_0(y) - \theta_0^J(y))\|_\infty^2$$

$$\le \frac{1}{n\epsilon_n^2}\sum_{j=1}^{J}\frac{2^{-2ja}}{\tau_j^2} + \|\theta_0(y) - \theta_0^J(y))\|_\infty^2$$

$$= \frac{1}{n\epsilon_n^2}\sum_{j=1}^{J} 2^{jd} + \|\theta_0(y) - \theta_0^J(y))\|_\infty^2$$

$$= \frac{2^d}{n\epsilon_n^2}\frac{2^{dJ} - 1}{2^d - 1} + \|\theta_0(y) - \theta_0^J(y))\|_\infty^2$$

$$\le \frac{2^d/(2^d - 1)}{(\log n)^2} + C'\epsilon_n^2,$$

where in the second inequality, we used the second assertion of Lemma 3.2 [30] for logistic function, the fifth inequality uses the fact that $\theta(y) - \theta_0^J(y)$ is orthogonal to $\theta_0(y) - \theta_0^J(y)$. For any $a \leq \alpha$ fix $J = J_\alpha$ otherwise $J = J_a$, and then it is straightforward to check from the definition of $\epsilon_n$ given in the assertion of the theorem that $(2^{dJ-1}/n\epsilon_n^2) \leq (\log n)^{-2}$. The term $\|\theta_0(y) - \theta_0^J(y))\|_\infty^2$ is also bounded by $C'\epsilon_n^2$ as shown in the proof of Theorem 4.5 in [30]. Consequently, the term $\frac{1}{n}\mathbb{E}_{Q_n}\mathrm{KL}(P_0^n\|P_\theta^n)$ is bounded above by $\epsilon_n^2$ (upto a constant) for sufficiently large $n$ since $(\log n)^{-2} < \epsilon_n^2$ and the result follows. $\qquad\square$

*Proof of Theorem 4.2.* The proof is a direct consequence of Theorem 3.2, Lemmas B.11, B.12, B.13, and Proposition 3.2. $\qquad\square$

