# OpenReview forum: "On the Statistical Consistency of Risk-Sensitive Bayesian Decision-Making"
_NeurIPS.cc/2023/Conference — NeurIPS 2023 poster_

### Official Review · Reviewer_j4gK · 2023-07-03

**Soundness:** 3 good
**Presentation:** 3 good
**Contribution:** 3 good
**Rating:** 7
**Confidence:** 5

**Summary:**

The paper presents a study on data-driven decision-making problems within the Bayesian framework. It addresses situations where calculating the posterior distribution becomes intractable due to the data-generating models’ complexity or the datasets’ size. The authors propose a novel Risk-Sensitive Variational Bayesian (RSVB) framework to overcome these challenges. The RSVB framework leverages a dual representation of the entropic risk measure to approximate the risk-sensitive posterior distribution and derive the corresponding decision rule. The paper also investigates the impact of these computational approximations on the predictive performance of the inferred decision rules. Practical examples in parametric and nonparametric settings support the theoretical findings.

**Strengths:**

Innovative Approach: Introducing the Risk-Sensitive Variational Bayesian (RSVB) framework significantly contributes to the field. By replacing the expectation in the Bayes risk with a risk-sensitive entropic risk measure, the authors provide a novel solution for decision-making problems in the Bayesian framework where calculating the posterior distribution is intractable.

Practical Relevance: The paper addresses a common challenge faced in modern applications with large datasets and complex data-generating models. The proposed RSVB framework offers a promising approach to handling such situations, making it relevant and valuable for researchers and practitioners in various domains.

Theoretical Analysis: The authors provide a comprehensive theoretical analysis of the RSVB framework, including convergence rates of the approximate posterior and the corresponding optimal value. This analysis enhances the understanding of the proposed approach and its theoretical foundations.

Illustrative Examples: The inclusion of three examples, both parametric and nonparametric, helps to demonstrate the practical applicability of the RSVB framework. By applying the proposed approach to real-world scenarios, the authors strengthen their theoretical findings and showcase the potential benefits of the RSVB framework in different contexts.


**Weaknesses:**

Some areas for improvement are as follows:

Clarity of Exposition: While the paper touches upon important concepts and theoretical aspects, the overall clarity of the exposition could be improved. Some technical details may require further explanation or examples to enhance reader comprehension.

Comparison with Existing Methods: It would be beneficial to include a more detailed comparison with existing approaches in the field. Specifically, comparing the proposed RSVB framework with other methods for handling intractable posterior distributions would provide a better understanding of its advantages and limitations.

Empirical Evaluation: Although the paper presents illustrative examples, a more extensive empirical evaluation could further support the practical effectiveness of the RSVB framework. Comparing its performance against alternative methods on broader datasets and problem domains would add more robustness to the findings.

**Questions:**

NA

**Limitations:**

Yes / NA

---

> ### Author Rebuttal · Authors · 2023-08-09
>
> We thank you for reading through our manuscript and providing insightful comments. Please find our response to your questions below:
>
> 1. Clarity of Exposition....
>
> R. To enhance the clarity and exposition of our theoretical contributions, we have added a proof sketch to the main theorems and provided more explanation to Assumption 3.1 (as provided in our response to the first comment of reviewer 3wrT). We have also added more discussion on the other assumptions in the revised manuscript. We would be happy to include any additional clarification in the revised manuscript.
>
> 2. Comparison with Existing Methods...
>
> R. Thank you for pointing this out. In our simulation and applications, we do include the existing approaches for risk-sensitive Bayesian decision-making, namely the standard approach VB/VI [1,2] and LCVI [2,3]. We would also like to bring to your attention that our RSVB formulation includes LCVI (i.e. for $\gamma=1$). We will make these points clear in the revised manuscript.
>
> 3. Empirical Evaluation....
>
> R. Thank you for your comment. We would like to emphasize that the main focus of the current manuscript is to introduce the RSVB methodology and derive its theoretical properties. Nonetheless, our theoretical results establish statistical guarantees for a well-known risk-sensitive variational posterior approximation technique LCVI (or LCVB) [2,3], which is a special case of RSVB for $\gamma=1$.
> We also demonstrate the efficacy of RSVB compared to VI and LCVI with the help of some simple examples. As extracting the main message from more complicated problem domains with broader datasets is harder,  we present our methodology with these simple examples. We agree with you that evaluating its performance on broader datasets and problem domains would add more robustness to the findings; therefore, we are currently looking at more extensive applications of the proposed approach. However, with an objective to clearly convey the paper's main theoretical findings and be within the page limit, we will present a more detailed empirical evaluation of RSVB elsewhere.
>
>
> [1] D. M. Blei, A. Kucukelbir, and J. D. McAuliffe. Variational inference: A review for statisticians. JASA (2017) \
> [2] S. Lacoste-Julien, F. Huszár, and Z. Ghahramani. Approximate inference for the loss-calibrated Bayesian. AISTATS (2011)\
> [3] T. Ku ́smierczyk, J. Sakaya, and A. Klami. Variational Bayesian decision-making for continuous utilities. NeurIPS (2019)

---

> > ### Comment · Reviewer_j4gK · 2023-08-16
> >
> > I would like to thank the authors for providing detailed and satisfactory responses to my comments. Please try to incorporate as much details as possible in the revised version of the paper. I especially appreciate the addition of the proof sketch to the main theorems.
> >
> > I would like to note that the paper has 30+ pages of technical analysis, which is impossible to understand with a "light" pass of the paper, so proof sketches, intuitions, explanations would make the paper much more accessible (provided you would like to make your paper accessible!). Although I am not opposed to recommending acceptance of this paper -my score is a 7 already-, please try to provide such elements throughout the limited length of the main paper, as this is not a journal submission.

---

> > > ### Author Response · Authors · 2023-08-19
> > >
> > > Thank you for acknowledging our response and providing a valuable suggestion. We will add more intuition and explanation to the assumptions and our main theorems to make our manuscript more accessible.

---

### Official Review · Reviewer_r5cb · 2023-07-06

**Soundness:** 3 good
**Presentation:** 3 good
**Contribution:** 3 good
**Rating:** 7
**Confidence:** 3

**Summary:**

In machine learning we often want to infer an optimal course of action in an uncertain scenario from observations we've made. Bayesian decision theory provides a framework to do just that: observations are used to build a posterior, which is then used to build an objective to optimize to decide a course of action. Unfortunately, both building a posterior and optimizing the objective can be challenging. To address these issues, previous methods had relied on variational inference of the posterior: they built an objective to jointly optimize the action and the variational approximation. This paper sets out to generalize previous methods with a temperature parameter ($\gamma$ in eq (SO)), and theoretically assess the convergence of this procedure, in particular assessing the effect of the expressivity of the variational family. They establish a convergence rate (Thm 3.1 and Prop 3.1) that depends on the contraction rate of the posterior as well as the expressivity of the variational family; they also address the effect of their temperature parameter. Finally they apply their theory to derive convergence rates in 3 examples.

**Strengths:**

The theory the authors establish is based on a number of standard assumptions that are readily verified for most Bayesian settings, including nonparametric settings. The authors also demonstrate their theory by applying it in 3 examples. Theory, like that of the authors, that accounts for the expressivity of the variational class is also much needed in the variational inference literature.

**Weaknesses:**

The assumptions the authors make (2.1-2.4) are generalizations of the standard Bayesian non-parametric convergence rate assumptions from Ghosal et al 2000. It is not immediately clear the significance of their differences, and where each of the assumptions are used in the following theorems. It is a weakness that it requires one to read the proofs in the supplement carefully to understand why each assumption is made. A sketch of the proofs, statements of lemmas, such as establishing which assumptions are necessary for the convergence rate of the posterior, would strengthen the paper.

**Questions:**

$\eta_n$ seems to simply be a rescaled version of eq (RSVB) with an expectation. Could the authors elaborate on why the expectation must be taken before the infimum; it was not clear why this had to be the case in lemma 6.14.

Assumption 3.1 is also not a standard assumption; I understand that they are able to prove it for the distributions given in the examples, but could they discuss why that is the case, and what the assumption means in simple terms?

**Limitations:**

In the Lacoste-Julien et al paper, they switched the loss for a "gain" so their variational objective became a joint maximization problem rather than a min-max objective, the later of which are in principle more difficult to optimize. This seems to be the major limitation and it is addressed but he authors.

---

> ### Author Rebuttal · Authors · 2023-08-09
>
> Thank you for your constructive feedback and for asking insightful questions. We respond to each of them below:
>
> 1. The assumptions the authors ....  the paper.
>
> R. Thank you for pointing this out.  We would also like to note that our assumptions 2.1-2.3 are the same as that in [7], which are motivated by the seminal work in [8]. [7] prove concentration rates for variational posterior, which are not risk-sensitive. Since we are interested in studying risk-sensitive posterior approximation, we introduce two additional assumptions (2.4 and 2.5) that involve the risk function. In Assumption 2.4, the prior is assumed to place sufficiently less mass in the region of the parameter space where risk is large, while Assumption 2.5 just requires the risk to be lower bounded.  We also add a proof sketch of our main Theorem 3.1 in the Appendix of the revised manuscript, highlighting the necessity of the assumptions and where they are invoked.
>
> 2. $\eta_n$ seems to simply be a rescaled ...... lemma 6.14.
>
> R. Thank you for noting this. Yes, your observation is correct. We also agree that we can very well interchange the order of expectation and infimum. However, note that if we do so, we will be back to analyzing the RSVB optimizer, the very objective of the whole analysis.  Taking expectation before infimum enables us to derive a more interpretable and meaningful bound. This is mainly because it is easier to control the expectation of the RSVB objective than its infimum through Assumption 3.1.
>
> 3. Assumption 3.1 is also not a ..... means in simple terms?
>
> R. Thank you for your comment. We actually believe that Assumption 3.1 is quite easy to verify. The condition that the variational family contains Dirac delta distribution at each point in the parameter space is a mild and reasonable requirement for consistency. Further, the assumption requires that the variational family contains sequences of distribution that weakly converge to each Dirac delta distribution at a certain rate. This is easily satisfied if the variational family has no "holes". E.g. if it is the family of Gaussians with all means and variances, then we can always construct sequences converging to any Dirac delta at any rate. A similar assumption has also been made in [6,7], and is true for most exponential family distributions. For instance, in the newsvendor application, we fix the variational family to a class of shifted-Gamma distribution and choose a sequence of distribution with parameter sequence $\alpha=n$ and $\beta=\frac{n}{\theta_0}$ which implies that the sequence of distribution has mean $\theta_0$ and variance $\frac{\theta_0^2}{n}$, and, therefore, it converges to Dirac delta distribution at $\theta_0$.
>
> 4. In the Lacoste-Julien et al paper, they ..... limitation and it is addressed but he authors.
>
> R. Thank you for pointing this out. As we note in our discussion section, we agree that our RSVB formulation requires solving a stochastic minimax optimization problem, which can be difficult in high-dimension without further assumptions. To circumvent this issue [2,3] convert the problem to max-max by switching the risk function with gain, which is a much easier problem to solve computationally. Note that to convert risk to gain, one requires the risk to be bounded above. We emphasize that our theoretical results will continue to hold in the max-max setting, but since our emphasis was on the statistical aspects of this problem, we chose to present our results with minimum assumptions.
>
> [2] S. Lacoste-Julien, F. Huszár, and Z. Ghahramani. Approximate inference for the loss-calibrated Bayesian. AISTATS (2011)\
> [3] T. Ku ́smierczyk, J. Sakaya, and A. Klami. Variational Bayesian decision-making for continuous utilities. NeurIPS (2019) \
> [6] Y. Wang and D. M. Blei. Frequentist consistency of variational Bayes. JASA (2018)\
> [7] F. Zhang and C. Gao. Convergence rates of variational posterior distributions. Ann. Stat. (2020)\
> [8] S. Ghosal, J. K. Ghosh, and A. W. van der Vaart. Convergence rates of posterior distributions. Ann. Stat. (2000)

---

> > ### Comment · Reviewer_r5cb · 2023-08-11
> > **Response to authors**
> >
> > I thank the authors for adequately addressing my questions.
> >
> > On point 2, while I agree that it is easier to control the quantity what the authors have written, I am also sure that the authors recognize that there is potentially a large gap between the infimum of the expectation and the expectation of the infimum; in the interest of future uses of this theory I suggest that it may be helpful to report the stronger bound, or at least describe that the stronger bound also holds.
> >
> > I have also noticed a few extra brackets in the equation on lines 228 to 229.

---

> > > ### Author Response · Authors · 2023-08-11
> > >
> > > Thank you for acknowledging our response and your valuable suggestion.
> > >
> > >  __R1:__ Following your suggestion, in the revised manuscript, we will highlight this point in the discussion after Theorem 3.1 and also point out why we chose to work with the infimum of expectation.
> > >
> > > __R2:__ Thank you for pointing this out. We will fix this in the revised manuscript.

---

### Official Review · Reviewer_3wrT · 2023-07-06

**Soundness:** 4 excellent
**Presentation:** 3 good
**Contribution:** 3 good
**Rating:** 7
**Confidence:** 4

**Summary:**

This paper introduces risk-sensitivity parameter $\gamma$ to a decision-making problem where latent model parameter $\theta$ is considered via variational Bayesian inference.
The authors analyze how close the approximate posterior is to the true parameter (Theorem 3.1) and how good the chosen action is (Theorem 3.2).
The theory is also verified with experimental results.

**Strengths:**

This approach extends loss-calibrated variational Bayesian (LCVB) in that the risk sensitivity is controlled.
The objective of risk-sensitive variational Bayes (RSVB) has risk expection term and KL divergence between approximate and true posteiors.
This combination may distort the approximate posterior, but Theorem 3.1 proves the posterior approaches the true parameter and Theorem 3.2 claims the gap of risk is small (with accurately specified model parameter $\theta_0$).

**Weaknesses:**

It is nice to provide assumptions on variational family like Assumption 3.1, but how to check if it is met seems unclear in general.

**Questions:**

* What is the purpose of Eight-schools experiment in Section 4.3?
Is it meant to confirm any theoretical aspects? Or is this experiment designed to check how $\gamma$ affect a complex hierarchical model?
To show the actual loss as a function of data points (or epochs) with various $\gamma$ values may be informative to see the effect of $\gamma$.

* Figure 2 (b) is unclear visualization with many colors overlaid, as well as gives unclear message.
Is it obvious the variance of posterior can change with $\gamma$?

* Is Figure 1 (c) missing? This appears in the text (line 288), but I could not find it.

**Limitations:**

While the authors do not put a distinct Limitation section, the scope of method and its theory is given as assumptions.
I do not find potential societal impact regarding this paper.

---

> ### Author Rebuttal · Authors · 2023-08-09
>
> Thank you for your thoughtful comments and questions. We address each of them below:
>
> 1. It is nice to provide ....... unclear in general.
>
> R. Thank you for asking this question. We actually believe that Assumption 3.1 is quite easy to verify. The condition that the variational family contains Dirac delta distribution at each point in the parameter space is a mild and reasonable requirement for consistency. Further, the assumption requires that the variational family contains sequences of distribution that weakly converge to each Dirac delta distribution at a certain rate. This is easily satisfied if the variational family has no "holes". E.g. if it is the family of Gaussians with all means and variances, then we can always construct sequences converging to any Dirac delta at any rate. A similar assumption has also been made in [6,7], and is true for most exponential family distributions. For instance, in the newsvendor application,  we fix the variational family to a class of shifted-Gamma distribution and choose a sequence of distribution with parameter sequence $\alpha=n$ and $\beta=\frac{n}{\theta_0}$ which implies that the sequence of distribution has mean $\theta_0$ and variance $\frac{\theta_0^2}{n}$, and, therefore, it converges to Dirac delta distribution at $\theta_0$.
>
> 2. What is the purpose of Eight-schools ......... $\gamma$.
>
> R. Thank you for your comment. The primary purpose of the eight-schools experiment is to empirically study a real-world example and see the effect of $\gamma$ on the RSVB decision compared to the VI/VB approach.  In the eight-schools experiment, we do not have the 'true' decision, so we compare the performance of the RSVB approach to VB. We assume that by 'actual loss', you mean the average loss function evaluated at the actions generated by the algorithm, as opposed to the regret that we consider. The latter is just the 'actual loss' shifted by a constant with respect to $\gamma$, so that plotting the `actual loss' as a function of epochs is not going to change our conclusions.
>
> 3. Figure 2 (b) is unclear visualization with ..... $\gamma$?
>
> R. Thank you for point this out. We will make Figure 2(b) look better and clearer in the revised version (also uploaded in the rebuttal document). However, we would like to note that it is not obvious that the variance of the RSVB posterior will reduce as $\gamma$ increases. We believe that it depends a lot on the landscape of the expected risk function and the choice of the variational family. However, since we observed this phenomenon in our experiments, we thought it was worth reporting.  Intuitively, a possible explanation of this phenomenon can be provided using the equation (RSVB). Consider the RSVB formulation and note that $\text{KL}>0$,  therefore as $\gamma$ increases, there is more incentive to deviate from the true posterior and choose $Q \in \mathcal{Q}$ that maximizes expected risk for a given $a\in \mathcal{A}$. Note that to maximize the expected risk, a $Q$ that places more mass near the $\theta$ that maximizes the risk will be preferred over the one with more spread. We will include this clarification in the revised manuscript.
>
> 4. Is Figure 1 (c) missing.....
>
> R. Thank you for pointing out this typo. It should be 1(b). We corrected this in the revised manuscript.
>
>
> [6] Y. Wang and D. M. Blei. Frequentist consistency of variational Bayes. JASA (2018)\
> [7] F. Zhang and C. Gao. Convergence rates of variational posterior distributions. Ann. Stat. (2020)

---

> > ### Comment · Reviewer_3wrT · 2023-08-12
> >
> > Thank you for the responses and clarificatoins.
> > Now that my raised ponits are addressed, I recommend this work for acceptance.

---

### Official Review · Reviewer_FLVD · 2023-07-07

**Soundness:** 3 good
**Presentation:** 3 good
**Contribution:** 3 good
**Rating:** 6
**Confidence:** 2

**Summary:**

This paper studies risk-sensitive Bayesian decision-making, where the expectation in the Bayes risk is replaced by a risk-sensitive entropic risk measure with respect to the posterior distribution. In particular, this work focuses on problems where the posterior distribution is intractable, and leverages a dual representation of the entropic risk measure to introduce a new framework, namely, RSVB. The goal of the paper is not to simply approximate the posterior distribution, but to also make decisions when that posterior is intractable. Simulations are performed to validate the effectiveness of the proposed framework.

**Strengths:**

This paper introduces a novel risk-sensitive variational Bayesian (RSVB) framework to jointly compute a risk-sensitive posterior approximation and the corresponding decision rule by solving a minimax optimization problem, where the true posterior distribution is intractable. The objective being considered is a general functional that can be frequently encountered in various decision-making problems.

In particular, the authors derive the posterior convergence rate for RSVB, as well as the convergence rate for decision rules, and identify the regularity conditions on the prior, likelihood model and the risk function. The authors also study the asymptotic consistency and provides a theoretical understanding of the statistical properties LCVB posterior and the associated decision rule.

Different applications are discussed by applying the proposed framework.

This paper is well-motivated and technically solid, and makes a considerable contribution by providing a general framework with theoretical guarantees for a wide class of problems. The convergence rate could be of potential interest for future research in the community.

**Weaknesses:**

1. The current discussion of simulations and applications does not include a comparison with other risk-based methods or methods that solve non-risk-based objectives. Further discussion will be beneficial.

2. As a benchmark comparison, it is desirable to see the comparison with the exact decision rule in the case where the true posteriors are available.

3. The current method requires solving a stochastic minimax optimization to compute a decision rule, which can be computationally intractable in practice.

**Questions:**

Do you forsee the applicability of the proposed framework for sequential decision-making problems (e.g. bandits and reinforcement learning)? And whether the convergence rate analysis / decision rules derived using the proposed algorithm could benefit the further understanding of Thompson sampling based algorithms in the case where posterior distributions are intractable?

**Limitations:**

This is a theory-based work. No potential negative societal impact.

---

> ### Author Rebuttal · Authors · 2023-08-09
>
> Thank you for your thorough review and valuable feedback. Please find our response to your questions below:
>
> 1. The current ..... be beneficial.
>
> R. Thank you for pointing this out. In our simulation and applications, we do include the existing approaches for risk-sensitive Bayesian decision-making, namely the standard approach VB/VI [1, 2] and LCVI [2,3].  We would also like to bring to your attention that our RSVB formulation includes LCVI (i.e. for $\gamma=1$). We will make these points clear in the revised manuscript.
>
> 2. As a benchmark comparison, ..... posteriors are available.
>
> R. Thank you for your comment. In our applications, the optimality gap itself is defined relative to the true decision (which is available in the synthetic experiment). Therefore, our synthetic experiment compares the RSVB decision rule with the ground truth. In neither of our current experiments, the true posterior is available as the priors are non-conjugate, making the posterior computation intractable. Having said that, we agree that comparison with a conjugate prior benchmark would be useful, therefore, in the newsvendor experiment, we have added a comparison with a conjugate prior (where the posterior is tractable) in Figure 1. We also uploaded it to the one-page rebuttal document. We note that this is a slightly different model.  With this caveat we observed that the decision rule learned using the conjugate prior has a similar performance as the VI (or VB) approach. However, the variance of the true conjugate posterior is higher than those computed through the RSVB and VI approach, corresponding to the well-known fact that  the variational approximations underestimate the true posterior variance [4,5].
>
> 3. The current method requires solving a ..... computationally intractable in practice.
>
> R. Thank you for noting this limitation. As we note in our discussion section, we agree that our RSVB formulation requires solving a stochastic minimax optimization problem, which can be difficult in high-dimension without further assumptions. To circumvent this issue [2,3] convert the problem to max-max by switching the risk function with gain, which is a much easier problem to solve computationally. Note that to convert risk to gain, one requires the risk to be bounded above. We emphasize that our theoretical results will continue to hold in the max-max setting, but since our emphasis was on the statistical aspects of this problem, we chose to present our results with minimum assumptions.
>
> 4. Do you foresee the applicability of the ........ posterior distributions are intractable?
>
> R. Yes, we agree that our concentration analysis can be useful to study Thompson/ Posterior sampling techniques in the bandit and reinforcement learning literature. However, note that we are in an offline setting where we use the observed data and then make a decision, whereas, in these settings decisions are taken online while sequentially adding more data observed at the chosen decision points.  This will require extensions to our work and we agree that this is an interesting future direction.
>
>
> [1] D. M. Blei, A. Kucukelbir, and J. D. McAuliffe. Variational inference: A review for statisticians. JASA (2017) \
> [2] S. Lacoste-Julien, F. Huszár, and Z. Ghahramani. Approximate inference for the loss-calibrated Bayesian. AISTATS (2011)\
> [3] T. Ku ́smierczyk, J. Sakaya, and A. Klami. Variational Bayesian decision-making for continuous utilities. NeurIPS (2019) \
> [4] Y. Li and R. E. Turner. Rényi divergence variational inference. NeurIPS (2016) \
> [5] R. E. Turner and M. Sahani. Two problems with variational expectation maximisation for time-series models. Camb. Univ. P (2011)

---

> > ### Comment · Reviewer_FLVD · 2023-08-17
> >
> > Thank you for the detailed replies.
> >
> > Regarding Q1, adding a discussion in related works / intro to provide a thorough comparison can better help to build a clear roadmap, not just in the empirical studies and later sections.
> >
> > Regarding Q4, in online settings, there is existing literature on applying Bayesian inference (both MCMC and VI) to bandit settings (e.g. "On Thompson Sampling with Langevin Algorithms" from Mazumdar et al 2020). By adapting the current analysis from offline setting to online setting, what do you think will be the benefits / contributions of this approach compared to these existing works, which also aim at making decisions under intractable posteriors?
> >
> > Additionally, it is desirable to distill the key technical insights / novelties / sketch of the proof in the main text to highlight the most important ideas from the analysis.
> >
> > Overall, this paper is clearly structured. I hope you find the points raised helpful and I look forward to seeing the revised manuscript.

---

> > > ### Author Response · Authors · 2023-08-19
> > >
> > > Thank you for acknowledging our response and for your helpful comments.
> > >
> > > R1: Following your suggestions, we will add a discussion in the introduction comparing the RSVB approach with LCVB and VB on our experiments.
> > >
> > > R4: Thank you for sharing this interesting reference on Langevin TS. However, we are unaware of any theoretical work that computes regret bounds for TS with VI. We believe that a part of the analysis in TS using VI would require similar concentration results (as in this manuscript) for the joint posterior distribution (over reward parameters). Such concentration analysis would help account for the number of samples needed for each arm in order to ‘separate’ the best arm from the sub-optimal ones. However, we think it would be hard to come up with the contributions without actually doing the analysis.
> > >
> > > Q5:  *Additionally, it is de..........analysis.* \
> > > R. In response to another reviewer’s comment, we add a proof sketch of our main Theorem 3.1 in the Appendix. We will move it to the main text. In addition, we will also highlight important technical insights and novelties in the revised manuscript.

---

### Author Rebuttal · Authors · 2023-08-09

We thank all the reviewers for their insightful and constructive comments. We greatly appreciate the time and effort they have
dedicated to reviewing our work. We have addressed each of them below and accordingly revised the manuscript. We are also attaching a 1-page pdf containing updated figures.

---

### Decision · Program_Chairs · 2023-09-21

**Decision:**

Accept (poster)

**Comment:**

The paper presents a novel Risk-Sensitive Variational Bayesian (RSVB) framework for addressing data-driven decision-making problems where calculating the posterior distribution is intractable. The paper provides a comprehensive theoretical analysis of the RSVB framework, including convergence rates of the approximate posterior and the corresponding optimal value. It also includes illustrative examples to demonstrate practical applicability. There were some questions and concerns raised by reviewers including a more detailed comparison with existing approaches for handling intractable posterior distributions on broader datasets and problem domains. The paper would benefit from addressing these comments.